# An emerging view of neural geometry in motor cortex supports high-performance decoding

**Sean M Perkins[1,2], Elom A Amematsro[2,3], John Cunningham[2,4,5,6], Qi Wang[1], Mark M Churchland[2,3,6,7]***

[1]Department of Biomedical Engineering, Columbia University, New York, United States; [2]Zuckerman Institute, Columbia University, New York, United States; [3]Department of Neuroscience, Columbia University Medical Center, New York, United States; [4]Department of Statistics, Columbia University, New York, United States; [5]Center for Theoretical Neuroscience, Columbia University Medical Center, New York, United States; [6]Grossman Center for the Statistics of Mind, Columbia University, New York, United States; [7]Kavli Institute for Brain Science, Columbia University Medical Center, New York, United States

## eLife Assessment

This paper presents a new method called MINT that is effective at BCI-style decoding tasks. The authors show **convincing** evidence to support their claims regarding how MINT is a new method that produces excellent decoding performance relative to the state-of-the-art. This work is **important** and will be of broad interest to neuroscientists and neuroengineers.

*For correspondence:
mc3502@columbia.edu

**Abstract** Decoders for brain-computer interfaces (BCIs) assume constraints on neural activity, chosen to reflect scientific beliefs while yielding tractable computations. Recent scientific advances suggest that the true constraints on neural activity, especially its geometry, may be quite different from those assumed by most decoders. We designed a decoder, MINT, to embrace statistical constraints that are potentially more appropriate. If those constraints are accurate, MINT should outperform standard methods that explicitly make different assumptions. Additionally, MINT should be competitive with expressive machine learning methods that can implicitly learn constraints from data. MINT performed well across tasks, suggesting its assumptions are well-matched to the data. MINT outperformed other interpretable methods in every comparison we made. MINT outperformed expressive machine learning methods in 37 of 42 comparisons. MINT's computations are simple, scale favorably with increasing neuron counts, and yield interpretable quantities such as data likelihoods. MINT's performance and simplicity suggest it may be a strong candidate for many BCI applications.

## Introduction

Brain-computer interfaces (BCIs) seek to estimate target variables, in real time, from observations of neural spiking activity. Target variables may correspond to prosthetic motion (*Carmena et al., 2003*; *Collinger et al., 2013*; *Hochberg et al., 2012*; *Hochberg et al., 2006*; *Velliste et al., 2008*; *Wodlinger et al., 2015*), muscle activity (*Ajiboye et al., 2017*; *Bouton et al., 2016*; *Ethier et al., 2012*; *Moritz et al., 2008*), cursor control (*Gilja et al., 2012*; *Jarosiewicz et al., 2015*; *Nuyujukian et al., 2015*;

**Figure 1.** Two perspectives on the structure of neural activity during motor tasks. (**a**) Illustration of a traditional perspective. Each neural state (*colored dots*) is an *N*-dimensional vector of firing rates. States are limited to a manifold that can be usefully approximated by a subspace. In this illustration, neural states are largely limited to a two-dimensional subspace within the full three-dimensional firing-rate space. Within that restricted subspace, there exist neural dimensions where neural activity correlates with to-be-decoded variables. BCI decoding leverages knowledge of the subspace and of those correlations. If two tasks have similar motor outputs at two particular moments, the corresponding neural states will also be similar, aiding generalization. (**b**) An emerging perspective. Neural activity is summarized by neural factors (one per axis), with each neuron's firing rate being a simple function of the factors. Factors may be numerous (many dozens or even hundreds), resulting in a high-dimensional subspace of neural activity. Yet most of that space is unoccupied; factor-trajectories are heavily constrained by the underlying dynamics (*gray arrows*), such that most states are never visited. The order in which states are visited is also heavily sculpted by dynamics; e.g. trajectories rarely 'swim upstream'. The primary constraint is thus not a subspace, but a sparse manifold where activity largely flows in specific directions. Different tasks may often (but not always) employ different dynamics and thus different regions of the manifold. Because most aspects of neural trajectories fall in the null space of outgoing commands, there are no directions in neural state space that consistently correlate with kinematic variables, and distant states may correspond to similar motor outputs.

---

*Pandarinath et al., 2017*; *Serruya et al., 2002*; *Shanechi et al., 2017*; *Taylor et al., 2002*), navigation (*Libedinsky et al., 2016*; *Rajangam et al., 2016*; *Schroeder et al., 2022*), speech (*Anumanchipalli et al., 2019*; *Willett et al., 2023*; *Wilson et al., 2020*; *Card et al., 2024*; *Metzger et al., 2023*; *Wairagkar et al., 2023*), handwriting (*Willett et al., 2023*), or cognitive states (*Musallam et al., 2004*; *Provenza et al., 2019*; *Sani et al., 2018*; *Wallis, 2018*; *Yousefi et al., 2019*). Alternatively, one may estimate the state of the brain, a latent variable, for clinical monitoring (*Chari et al., 2020*), data visualization (*Cowley et al., 2013*), or closed-loop neurally contingent experiments (*Peixoto et al., 2021*). Because neural spiking is noisy, target variables must be estimated from patterns of spikes that were unobserved during training. Doing so requires assumptions.

*Figure 1a* illustrates a set of common assumptions. In this view, the primary objects are neural firing rates (which determine the probability of spiking) and a manifold (*Gallego et al., 2017*; *Gallego et al., 2018*; *Golub et al., 2018*; *Sadtler et al., 2014*) that constrains the possible values of the 'neural state': a vector containing the rate of every neuron. While that manifold is presumed to be nonlinear, a linear approximation – i.e. a subspace – may be acceptable for practical purposes (*Gallego et al., 2017*). Within this subspace, there exist neural dimensions where activity correlates with target variables such as hand velocity, providing a basis for estimating those variables. Generalization (e.g. *Gilja et al., 2012*; *Jarosiewicz et al., 2015*; *Weiss et al., 2019*) is thought to rely on two features. First, if distributions of behavioral variables overlap across tasks, neural-state distributions will also overlap (*Gallego et al., 2018*). Second, out-of-distribution generalization can leverage the reliable correlation between behavioral variables and neural activity.

Decoding methods frequently assume many or all of these features. For example, the classic population vector (*Georgopoulos et al., 1986*; *Schwartz, 1994*; *Taylor et al., 2002*) assumes a two- or three-dimensional manifold (for reaches in two or three physical dimensions) and leverages neural dimensions where activity correlates with hand velocity. Similarly, nearly all 'interpretable' methods

(those that make explicit assumptions rather than learning them from training data) assume neural dimensions where activity correlates – perhaps incidentally, but at least usefully – with target variables. Such methods often assume additional constraints on neural activity related to structure in behavior (*Brockwell et al., 2004*; *Kao et al., 2017*; *Kemere et al., 2003*; *Kemere et al., 2004a*; *Kemere et al., 2004b*; *Kemere et al., 2002*; *Pandarinath et al., 2017*; *Shanechi et al., 2017*; *Yu et al., 2007*). E.g. a Kalman filter can embody the assumption that velocity and position are dynamically linked (*Gilja et al., 2012*; *Wu et al., 2003*), while *Sani et al., 2021* leveraged the assumption that only a subspace within a low-dimensional neural manifold is relevant to behavior.

Although the perspective in *Figure 1a* has been useful, its assumptions may be true only locally (*Fortunato et al., 2024*) and perhaps not even then. Under an alternative perspective (*Figure 1b*), the primary objects are neural factors (which determine each neuron's instantaneous probability of spiking) and a flow-field (gray arrows) governing factor-state trajectories. Within the space of potential factor states, the flow-field ensures that few are visited. The resulting manifold is complex, and may or may not be locally flat in any useful sense. For example, when cycling at different speeds, the manifold is an (at least) seven-dimensional tube (as in *Figure 1b*, green) whose long-axis corresponds to cycling speed (*Saxena et al., 2022*). Many locations in factor-space are never visited, including the void within the tube. The manifold is thus sparse – a minority of factor-states are observable – even though individual-neuron responses are rarely sparse in the traditional sense (most neurons show time-varying activity during most movements). Different tasks (or subtasks) may require different computations, and thus employ different regions of factor space with different local flow-fields. This may result in a complex and extremely sparse manifold. Some factors may, in some regions of the manifold, correlate incidentally with external variables such as velocity. Yet this is not expected to be consistently true.

The assumptions in *Figure 1* accord with the hypothesis of factor-level dynamics (*Churchland and Shenoy, 2024*; *DePasquale et al., 2023*; *Vyas et al., 2020*) that sculpt computation-specific neural geometries (*Churchland et al., 2012*; *Mante et al., 2013*; *Saxena et al., 2022*; *Sussillo et al., 2015*; *Remington et al., 2018*; *Russo et al., 2018*; *Russo et al., 2020*; *Sohn et al., 2019*). As a simple example, *DePasquale et al., 2023* constructed a network of spiking neurons to generate muscle activity during the cycling task (*Russo et al., 2018*). Network dynamics produced a limit cycle in a 12-dimensional factor space, with any deviations being swiftly corrected by the flow-field. Thus, although the data occupy a sizable subspace, the manifold consists of only those states near the limit cycle. When the network was trained to both cycle and reach, it did so using task-specific dynamics in different neural dimensions, resulting in a sparse, complex manifold with task-specific sub-regions.

*Figure 1b* also accords with analyses that presume high-dimensional neural trajectories (*Chaudhuri et al., 2019*; *Goudar and Buonomano, 2018*; *Mishne et al., 2016*; *Stopfer et al., 2003*), with methods that assume strongly nonlinear mappings between neural activity and behavioral variables (*Schneider et al., 2023*; *Zhou and Wei, 2020*), and with the finding that locally linear relationships between activity and behavior are prominent during some tasks but not others (*Schroeder et al., 2022*). It agrees with the proposal that factors (though less numerous than neurons) are plentiful (*Churchland and Shenoy, 2007*; *Gao and Ganguli, 2015*; *Gao et al., 2017*; *Marshall et al., 2022*; *Seely et al., 2016*), with different tasks and task-epochs often using different factors (*Ames and Churchland, 2019*; *Heming et al., 2019*; *DePasquale et al., 2023*; *Elsayed et al., 2016*; *Miri et al., 2017*; *Schroeder et al., 2022*; *Warriner et al., 2022*; *Xing et al., 2022*) (the 'flexibility-via-subspace' hypothesis; *Churchland and Shenoy, 2024*), and with the ability of training to 'open up' previously unused degrees of freedom (*Oby et al., 2019*). The assumption of a strong flow-field agrees with empirical limitations on BCI-generated trajectories (*Athalye et al., 2023*; *Oby et al., 2024*) and with the finding that decoding is aided by assuming neural-state dynamics (*Kao et al., 2015*; *Pandarinath et al., 2018*). More broadly, low trajectory tangling (*Russo et al., 2018*; *Saxena et al., 2022*) – a feature of motor cortex activity in most tasks – constrains neural trajectories in ways that imply many of the features in *Figure 1b*, and thus argues for approximating the manifold using collections of trajectories rather than a subspace (*Brennan et al., 2023*; *Goudar and Buonomano, 2018*).

There is thus a potential mismatch between the structure of the data and the assumptions made by traditional interpretable decoders. This may be one reason why interpretable methods are often outperformed by expressive machine-learning methods (*Glaser et al., 2020*; *Schwemmer et al., 2018*; *Willett et al., 2023*). Expressive methods (*Ahmadi et al., 2021*; *Anumanchipalli et al., 2019*; *Card et al., 2024*; *Glaser et al., 2020*; *Makin et al., 2018*; *Metzger et al., 2023*; *Pandarinath et al.,*

*2018*; *Schwemmer et al., 2018*; *Sussillo et al., 2012*; *Sussillo et al., 2016*; *Tseng et al., 2019*; *Wairagkar et al., 2023*; *Willett et al., 2021*; *Willett et al., 2023*; *Ye and Pandarinath, 2021*) may be able to implicitly learn, during training, many of the constraints illustrated in *Figure 1b*. Motivated by these considerations, we constructed a novel algorithm, Mesh of Idealized Neural Trajectories (MINT), that explicitly leverages those constraints. MINT takes a trajectory-centric view, where a complicated manifold is approximated using previously observed trajectories and interpolations between them. MINT abandons any notion of neural dimensions that reliably correlate with behavioral variables. Instead, MINT creates a direct correspondence between neural and behavioral trajectories, allowing it to capture highly nonlinear relationships. These relationships can be task-specific if the data so argue. One might have expected that mathematical tractability would be compromised by embracing the unusual assumptions in *Figure 1b*. Yet the requisite computations are surprisingly simple. It also becomes straightforward to decode a variety of behavioral variables – even those that do not correlate with neural activity – and to estimate the neural state.

MINT's performance confirms that there are gains to be made by building decoders whose assumptions match a different, possibly more accurate view of population activity. At the same time, our results suggest fundamental limits on decoder generalization. Under the assumptions in *Figure 1b*, it will sometimes be difficult or impossible for decoders to generalize to not-yet-seen tasks. We found that this was true regardless of whether one uses MINT or a more traditional method. This finding has implications regarding when and how generalization should be attempted.

## Results

We applied MINT to a total of nine datasets, recorded from primates performing a variety of motor, sensory, and cognitive tasks. Each dataset consisted of simultaneous neural recordings and behavioral variables. We used MINT to decode, based on spiking observations, behavioral variables and the neural state. A central goal was to compare MINT's decoding performance with existing 'interpretable' methods (methods such as the Kalman filter that make explicit assumptions regarding data constraints) and with 'expressive' methods (e.g. neural networks that can learn constraints from training data). Current interpretable methods make assumptions largely aligned with *Figure 1a*. Thus, if *Figure 1b* better describes the data, MINT should consistently outperform other interpretable methods. Highly expressive methods can, given enough training data, implicitly learn a broad range of constraints, including those in *Figure 1a, b*, or some other yet-to-be imagined scenario. This provides a strong test of the assumptions in *Figure 1b*, because MINT should perform well if those assumptions are correct. Indeed, if *Figure 1b* is correct, MINT's performance should be similar to that of highly expressive methods, and may even be better in some cases because MINT can begin with good assumptions rather than having to learn them from data.

We also document properties of the data – directly and during decoding – that bear on the scientific question of whether *Figure 1a or b* makes better assumptions. A comprehensive characterization of neural trajectory geometry is beyond the scope of this study. Yet, whenever possible, we examine features of neural trajectories relevant to MINT's assumptions. To do so, we begin by focusing on one dataset (MC_Cycle) recorded during a task in which a primate grasps a hand-pedal and moves it cyclically forward or backward to navigate a virtual environment.

### Neural trajectories are locally stereotyped and sparsely distributed

During the cycling task, empirical neural trajectories are compatible with solutions found by neural networks trained to produce muscle commands as an output (*Russo et al., 2018*). Those trajectories have features that tend to argue for the assumptions in *Figure 1b*. Many of these features follow from the property of 'low trajectory tangling': during movement, it is never the case that very different factor-trajectory directions are associated with similar factor states. Trajectories are thus locally stereotyped: two trajectories that pass near one another are moving in similar directions. When trajectories travel in dissimilar directions, maintaining low tangling requires that they avoid one another, spreading out into additional dimensions and becoming more sparse in factor space. For example, the elliptical trajectories during forward and backward cycling (*Figure 2a*) appear to overlap in the dominant two dimensions. Yet despite this appearance, the corresponding neural trajectories never come close to crossing. For example, at the apparent crossing-point indicated by the gray disk, forward and

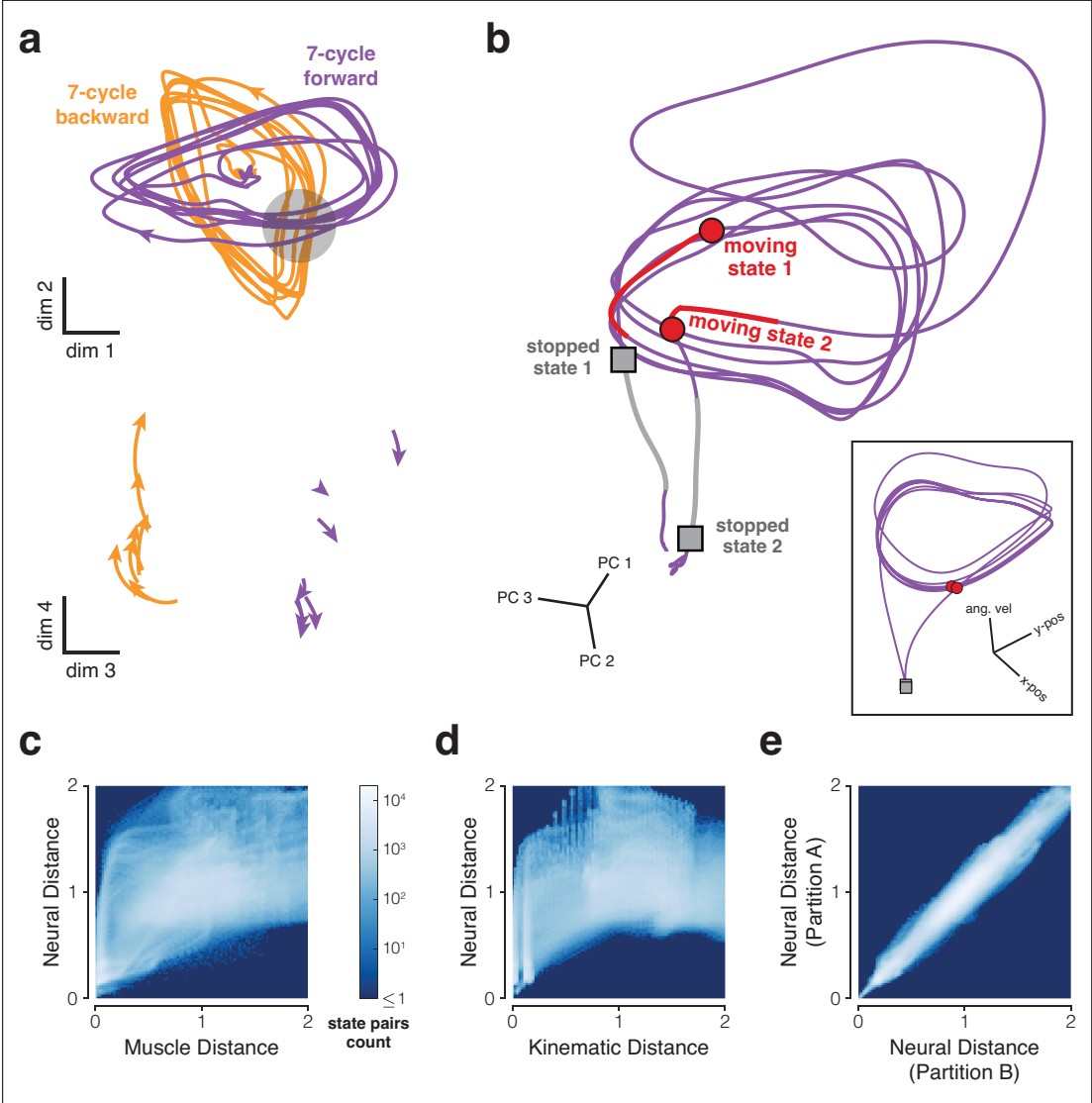

**Figure 2.** Properties of neural trajectories in motor cortex, illustrated for data recorded during the cycling task (MC_Cycle dataset). (**a**) Low tangling implies separation between trajectories that might otherwise be close. *Top.* Neural trajectories for forward (*purple*) and backward (*orange*) cycling. Trajectories begin 600 ms before movement onset and end 600 ms after movement offset. Trajectories are based on trial-averaged firing rates, projected onto two dimensions. Dimensions were selected to highlight apparent crossings of the cyclic trajectories during forward and backward cycling, while also capturing considerable variance (11.5%, comparable to the 11.6% captured by PCs 3 and 4). *Gray region* highlights one set of apparent crossings. *Bottom.* Trajectories during the restricted range of times in the gray region, but projected onto different dimensions. The same scale is used in top and bottom subpanels. (**b**) Examples of well-separated neural states (*main panel*) corresponding to similar behavioral states (*inset*). Colored trajectory tails indicate the previous 150 ms of states. Data from 7-cycle forward condition. (**c**) Joint distribution of pairwise distances for muscle and neural trajectories. Analysis considered all states across all conditions. For both muscle and neural trajectories, we computed all possible pairwise distances across all states. Each muscle state has a corresponding neural state, from the same time within the same condition. Thus, each pairwise muscle-state distance has a corresponding neural-state distance. The color of each pixel indicates how common it was to observe a particular combination of muscle-state distance and neural-state distance. Muscle trajectories are based on 7 z-scored intramuscular EMG recordings. Correspondence between neural and muscle state pairs included a 50 ms lag to account for physiological latency. Results were not sensitive to the presence or size of this lag. Neural and muscle distances were normalized (separately) by average pairwise distance. (**d**) Same analysis for neural and kinematic distances (based on phase and angular velocity). Correspondence between neural and kinematic state pairs included a 150 ms lag. Results were not sensitive to the presence or size of this lag. (**e**) Control analysis to assess the impact of sampling error. If two sets of trajectories (e.g. neural and kinematic) are isometric and can be estimated perfectly, their joint distribution should fall along the diagonal. To estimate the impact of sampling error, we repeated the above analysis comparing neural distances across two data partitions, each containing 15–18 trials/condition.

backward trajectories are well-separated in dimension 3 (all dimensions use the same scale). This is true at all apparent crossing points, leading to forward-cycling and backward-cycling trajectories that are well-separated. Additionally, both limit cycles contain a void in their center that is never occupied. One might expect that such voids would fill in as additional behaviors are considered. For example, perhaps the interior of the limit cycle becomes occupied when stationary, or at slower speeds? Yet when stationary, neural states are far from the limit-cycle center (*Figure 2b*). Cycling at different speeds involves neural trajectories that spread out in additional 'new' dimensions, and thus remain sparse (*Saxena et al., 2022*). This illustrates a key difference between the notion of a manifold in *Figure 1a and b*. In *Figure 1a*, if two distant neural states are both likely to be observed, then the state halfway between them is also reasonably likely to be observed. In *Figure 1b*, this will frequently not be true.

## Neural and behavioral trajectories are non-isometric

A potential consequence of low-tangled trajectories is that similar behavioral outputs can be associated with distant neural states. We found that this was common. As a simple example, there exist many moments when two-dimensional velocity is matched between forward and backward cycling: e.g. at a 45° phase when cycling forward versus 225° when cycling backward. Yet forward and backward trajectories remain distant at all moments. One might suspect that this is simply because angular position is different, even though velocity is matched. Yet neural states can be quite different even when both position and velocity are matched. This point is illustrated (*Figure 2b*) by the neural trajectory that unfolds when transitioning from stationary, to cycling forward seven times, to stationary again. This projection was chosen to highlight dimensions where the neural trajectory resembles the behavioral trajectory (inset). Nevertheless, neural and behavioral trajectories are far from isometric. Red circles indicate two neural states where the corresponding positions and angular velocities are nearly identical: cycling at ~1.7 Hz with the hand approaching the cycle's bottom. Despite this behavioral match, the neural states are distant. Gray squares indicate neural states that are distant even though, in both cases, the hand is stopped at the cycle's bottom.

These observations suggest a many-to-one mapping from a neural manifold with one geometry to a behavioral manifold with a very different geometry. To quantify, we leveraged the fact that each neural distance (between a pair of neural states) has a corresponding behavioral distance, obtained for the same pair of conditions and times (allowing for a latency shift). If behavioral and neural geometries are similar, behavioral and neural distances should be strongly correlated. If geometries differ, a given behavioral distance could be associated with a broad range of neural distances. To establish a baseline that incorporates sampling error, we compared neural distances across two partitions. As expected, the joint distribution was strongly diagonal (*Figure 2e*). In contrast, the joint distribution was non-diagonal when comparing neural versus muscle-population trajectories (*Figure 2c*) and neural versus kinematic trajectories (*Figure 2d*). Each behavioral distance (on the x-axis) was associated with a range of different neural distances (on the y-axis). This was true even when behavioral distances were small: for kinematic states with normalized distance <0.1, the corresponding neural states were as close as 0.02 and as far as 1.46. Thus, dissimilar neural states frequently correspond to similar behavioral states, as suggested by *Figure 1b* (see states indicated by arrows).

Importantly, the converse was not true. It was never the case that similar neural states corresponded to dissimilar behavioral states. Once past small values on the x-axis, the white regions in *Figure 2c, d* never extend all the way to the x-axis. For example, when kinematic distances were ~2, neural distances were always at least 0.53. Thus, accurate decoding should be possible as long as the decoder can handle the many-to-one mapping of neural states to behavioral variables. In principle this might be accomplished linearly. Suppose neural trajectories mirrored behavioral trajectories but with additional non-behavior-encoding dimensions. A linear decoder could simply ignore those additional dimensions. Yet, this will be suboptimal if the ignored dimensions capture the majority of response variance. The above observations thus argue that one may desire quite nonlinear decoding.

A potential concern regarding the analyses in *Figure 2c, d* is that they require explicit choices of behavioral variables: muscle population activity in *Figure 2c* and angular phase and velocity in *Figure 2d*. Perhaps these choices were misguided. Might neural and behavioral geometries become similar if one chooses 'the right' set of behavioral variables? This concern relates to the venerable search for movement parameters that are reliably encoded by motor cortex activity (*Churchland and*

*Shenoy, 2007*; *Fetz, 1992*; *Reimer and Hatsopoulos, 2009*; *Scott, 2008*; *Todorov, 2000*). If one chooses the wrong set of parameters (e.g. chooses muscle activity when one should have chosen joint angles) then of course neural and behavioral geometries will appear non-isometric. There are two reasons why this 'wrong parameter choice' explanation is unlikely to account for the results in *Figure 2c, d*. First, consider the implications of the left-hand side of *Figure 2d*. A small kinematic distance implies that angular pedal position and velocity are both nearly identical for the two moments being compared. Yet the corresponding pair of neural states can be quite distant. Under the concern above, this distance would be due to other encoded behavioral variables – perhaps joint angle and joint velocity – differing between those two moments. However, there are not enough degrees of freedom in this task to make this plausible. The shoulder remains at a fixed position (because the head is fixed) and the wrist has limited mobility due to the pedal design (*Russo et al., 2018*). Thus, shoulder and elbow angles are almost completely determined by cycle phase. More generally, 'external variables' (positions, angles, and their derivatives) are unlikely to differ more than slightly when pedal phase and angular velocity are matched. Muscle activity could be different because many muscles act on each joint, creating redundancy. However, as illustrated in *Figure 2c*, the key effect is just as clear when analyzing muscle activity. Thus, the above concern seems unlikely even if it can't be ruled out entirely. A broader reason to doubt the 'wrong parameter choice' proposition is that it provides a vague explanation for a phenomenon that already has a straightforward explanation. A lack of isometry between the neural population response and behavior is expected when neural-trajectory tangling is low and output-null factors are plentiful (*Churchland and Shenoy, 2024*; *Russo et al., 2018*). For example, in networks that generate muscle activity, neural and muscle-activity trajectories are far from isometric (*Russo et al., 2018*; *Saxena et al., 2022*; *Sussillo et al., 2015*). Given this straightforward explanation, and given repeated failures over decades to find the 'correct' parameters (muscle activity, movement direction, etc.) that create neural-behavior isometry, it seems reasonable to conclude that no such isometry exists.

We further explored the topic of isometry by considering pairs of distances. To do so, we chose two random neural states and computed their distance, yielding $d_{neural1}$. We repeated this process, yielding $d_{neural2}$. We then computed the corresponding pair of distances in muscle space ($d_{muscle1}$ and $d_{muscle2}$) and kinematic space ($d_{kin1}$ and $d_{kin2}$). We considered cases where $d_{neural1}$ was meaningfully larger than (or smaller than) $d_{neural2}$, and asked whether the behavioral variables had the same relationship; for example was $d_{muscle1}$ also larger than $d_{muscle2}$? For kinematics, this relationship was weak: across 100,000 comparisons, the sign of $d_{kin1} - d_{kin2}$ agreed with $d_{neural1} - d_{neural2}$ only 67.3% of the time (with 50% being chance). The relationship was much stronger for muscles: the sign of $d_{muscle1} - d_{muscle2}$ agreed with $d_{neural1} - d_{neural2}$ 79.2% of the time, which is far more than expected by chance yet also far from what is expected given isometry (e.g. the sign agrees 99.7% of the time for the truly isometric control data in *Figure 2e*). Indeed there were multiple moments during this task when $d_{neural1}$ was much larger than $d_{neural2}$, yet $d_{muscle1}$ was smaller than $d_{muscle2}$. These observations are consistent with the proposal that neural trajectories resemble muscle trajectories in some dimensions, but with additional output-null dimensions that break the isometry (*Russo et al., 2018*).

## Leveraging trajectories to estimate the manifold

The results above, along with other recent results, argue for the perspective in *Figure 1b*. In this view, the manifold of observable states is complicated and sparse. We thus designed MINT to approximate that manifold using the neural trajectories themselves, rather than their covariance matrix or corresponding subspace. Unlike a covariance matrix, neural trajectories indicate not only which states are likely, but also which state-derivatives are likely. If a neural state is near previously observed states, it should be moving in a similar direction. MINT leverages this directionality.

Training-set trajectories can take various forms, depending on what is convenient to collect. Most simply, training data might include one neural trajectory per condition, with each condition corresponding to a distinct movement type. Alternatively, one might employ one long trajectory spanning many movements. Another option is to employ many sub-trajectories, each briefer than a whole movement. The goal is simply for training-set trajectories to act as a scaffolding, outlining the manifold that might be occupied during decoding and the directions in which decoded trajectories are likely to be traveling.

Because training-set trajectories are unlikely to sample the manifold with sufficient density (e.g. one may train using eight reach directions but wish to decode any direction), we designed MINT to interpolate during decoding. We use the term 'mesh' to describe the scaffolding created by the training-set trajectories and the interpolated states that arise at runtime. The term mesh is apt because, if MINT's assumptions are correct, interpolation will almost always be local. If so, the set of decodable states will resemble a mesh, created by line segments connecting nearby training-set trajectories. However, this mesh-like structure is not enforced by MINT's operations. Interpolation could, in principle, create state-distributions that depart from the assumption of a sparse manifold. For example, interpolation could fill in the center of the green tube in *Figure 1b*, resulting in a solid manifold rather than a mesh around its outer surface. However, this would occur only if spiking observations argued for it. As will be documented below, we find that essentially all interpolation is local. This is a useful fact, because it implies that the estimated neural state (during decoding) will rarely be far from previously observed neural states for which the corresponding behavioral states are known. MINT finds those behavioral states using a direct (and highly nonlinear) mapping between neural and behavioral states. MINT then decodes behavior using local linear interpolation.

Although the mesh is formed of stereotyped trajectories, decoded trajectories can move along the mesh in non-stereotyped ways as long as they generally obey the flow-field implied by the training data. This flexibility supports many types of generalization, including generalization that is compositional in nature. Other types of generalization – for example from the green trajectories to the orange trajectories in *Figure 1b* – are unavailable when using MINT and are expected to be challenging for any method (as will be documented in a later section).

## Training and decoding using MINT

Training MINT requires computing a library of neural trajectories ($\Omega$) and a corresponding library of behavioral trajectories ($\Phi$). For ease of subsequent computation, neural trajectories are expressed in firing-rate space. MINT is agnostic regarding how neural trajectories are found. For example, one can estimate factor-trajectories, then convert to firing rates via a rectifying nonlinearity (*DePasquale et al., 2023*; *Pandarinath et al., 2018*). The simplest approach, when training data contain repeated trials per condition, is to compute each neuron's rate via trial-averaging. We use this approach to illustrate MINT's operations during a center-out reaching task. Neural and behavioral trajectories (*Figure 3a*) were learned from a training set containing repeated reaches (trials) to each target location. Each moment in that training data corresponds to a condition ($c$) and an index ($k$) indicating progress through the movement. For example, $c = 3$ corresponds to the purple reach condition and $k = 240$ indicates a moment 240 ms into that movement. There were many trials for this condition, and thus many measurements of neural activity and behavior for $c = 3$ and $k = 240$. By temporally filtering spikes and averaging across trials, one obtains a neural state $\overline{x}_k^c$ containing each neuron's estimated rate. If desired, firing-rate estimation can be further improved by a variety of means (see Methods). A similar process yields a behavioral state $\overline{z}_k^c$, which may contain position, velocity, or any other variables of interest.

As expected given results above, neural and behavioral trajectories differ during reaching: neural trajectories are stacked loops while position trajectories are arranged radially (velocity trajectories are also radial but return to center as the reach ends). These different neural and behavioral geometries might seem like an impediment to decoding, or to argue for a different choice of target behavioral variables. Yet because neural and behavioral trajectories share the same indices $c$ and $k$, decoding is straightforward regardless of which variables one wishes to decode.

The parameters of MINT are the libraries of neural trajectories ($\Omega$) and behavioral trajectories ($\Phi$). The trajectories in $\Omega$ form a scaffolding that outlines the expected neural manifold during decoding. $\Omega$ also specifies the directionality with which decoded states are expected to traverse that manifold. On long timescales, decoded trajectories may be very different from any library trajectory. Yet decoded trajectories will typically be composed of events resembling those from the library, reflecting the local stereotypy described above. Prior work has utilized stereotyped behavioral trajectories (*Kemere et al., 2004b*; *Kemere et al., 2002*) or a mixture of condition-specific behavioral trajectory models that capture trial-by-trial movement variability (*Yu et al., 2007*). For tractability, those methods assumed low-dimensional neural states that are roughly isometric to arm velocities. With MINT, neural trajectories can have any geometry, are learned empirically, and are related to behavioral trajectories via $c$ and $k$ rather than by an assumed geometric isometry.

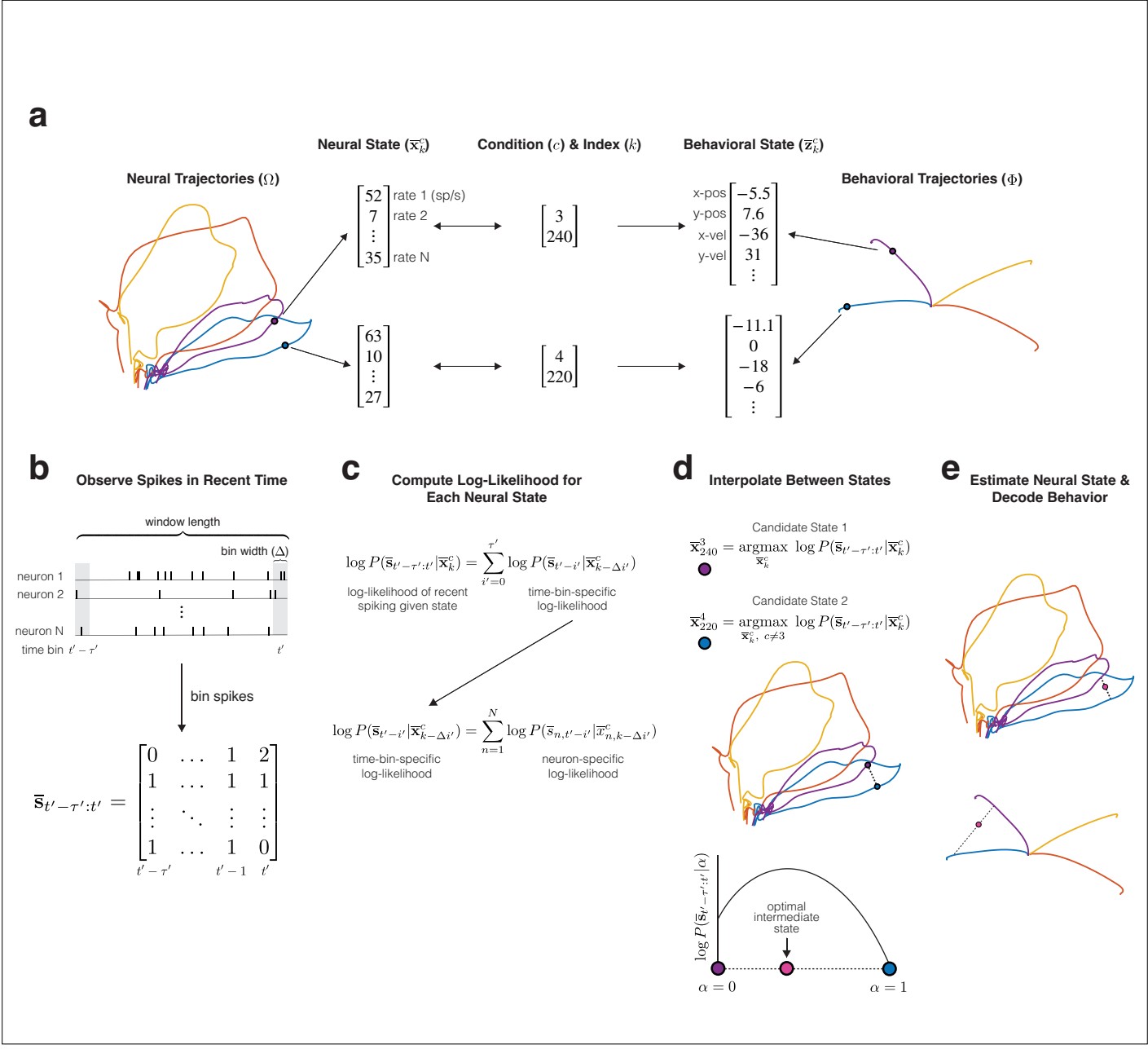

**Figure 3.** Example training (*top panel*) and decoding (*bottom panels*) procedures for MINT, illustrated using four conditions from a reaching task. (**a**) Libraries of neural and behavioral trajectories are learned such that each neural state $\overline{\mathbf{x}}_k^c$ corresponds to a behavioral state $\overline{\mathbf{z}}_k^c$. (**b**) Spiking observations are binned spike counts (20 ms bins for all main analyses). $\overline{\mathbf{s}}_{t'-\tau':t'}$ contains the spike count of each neuron for the present bin, $t'$, and $\tau'$ bins in the past. (**c**) At each time step during decoding, the log-likelihood of observing $\overline{\mathbf{s}}_{t'-\tau':t'}$ is computed for each and every state in the library of neural trajectories. Log-likelihoods decompose across neurons and time bins into Poisson log-likelihoods that can be queried from a precomputed lookup table. A recursive procedure (not depicted) further improves computational efficiency. (**d**) Two candidate neural states (*purple* and *blue*) are identified. The first is the state within the library of trajectories that maximizes the log-likelihood of the observed spikes. The second state similarly maximizes that log-likelihood, with the restriction that the second state must not come from the same trajectory as the first (i.e. must be from a different condition). Interpolation identifies an intermediate state that maximizes log-likelihood. (**e**) The optimal interpolation is applied to candidate neural states – yielding the final neural-state estimate – and their corresponding behavioral states – yielding decoded behavior. Despite utilizing binned spiking observations, neural and behavioral states can be updated at millisecond resolution (Methods).

During decoding, spike-based observations are binned and counted (*Figure 3c*, $\Delta$ = 20 ms for all analyses unless otherwise specified). Suppose we are at time-bin $t'$ within a decoding session. Recent spiking observations are denoted by $\bar{\mathbf{s}}_{t'-\tau':t'}$, where $\tau'$ specifies the number of previous time bins retained in memory. The foundation of MINT is the computation of the log-likelihood of the data, $\bar{\mathbf{s}}_{t'-\tau':t'}$, for a variety of neural states. The decoded state is chosen with the goal of maximizing that log-likelihood. Not all decoders involve data likelihood computations, but likelihoods are desirable when it is possible to obtain them. One typically wishes to decode the state that maximizes data likelihood, or maximizes an objective function that involves likelihoods across multiple states. Knowledge of likelihoods can also indicate the trustworthiness of decoding, as will be discussed in a subsequent section.

To decode, we first compute the log-likelihood that we would have observed $\bar{\mathbf{s}}_{t'-\tau':t'}$ for each neural state in the library (*Figure 3c*). Importantly, this computation is performed for relatively few states. Even with 1000 samples for each of the neural trajectories in *Figure 3*, there are only 4000 possible neural states for which log-likelihoods must be computed (in practice it is fewer still, see Methods). This is far fewer than if one were to naively consider all possible neural states in a typical rate- or factor-based subspace. It thus becomes tractable to compute log-likelihoods using a Poisson observation model. A Poisson observation model is usually considered desirable, yet can pose tractability challenges for methods that utilize a continuous model of neural states. For example, when using a Kalman filter, one is often restricted to assuming a Gaussian observation model to maintain computational tractability.

The library, $\Omega$, makes it simple to compute data likelihoods that take spike-history into account. Without the constraint of a flow-field, a great many past trajectories could have led to the present neural state. Computing the likelihood of $\bar{\mathbf{s}}_{t'-\tau':t'}$, given all these possible histories, may be impractical during real-time decoding. Yet given a strong flow-field, such histories will be similar on short timescales. For example, if we are presently 240 ms into the purple trajectory in *Figure 3a* (i.e. $c$ = 3 and $k$ = 240) then we know (approximately) the rate of every neuron over the last few hundred milliseconds. One can thus compute the log-likelihood of the observed pattern of spikes, over the last few hundred milliseconds, given that we are presently at $c$ = 3 and $k$ = 240. This computation is aided by the fact that, under a Poisson model, spike counts in non-overlapping bins are conditionally independent given knowledge of the underlying rate. Similarly, spike counts for different neurons are conditionally independent given knowledge of their rates. Thus, the log-likelihood of $\bar{\mathbf{s}}_{t'-\tau':t'}$, for a particular current neural state, is simply the sum of many individual log-likelihoods (one per neuron and time-bin). Each individual log-likelihood depends on only two numbers: the firing rate at that moment and the spike count in that bin. To simplify online computation, one can precompute the log-likelihood, under a Poisson model, for every plausible combination of rate and spike-count. For example, a lookup table of size 2001 × 21 is sufficient when considering rates that span 0-200 spikes/s in increments of 0.1 spikes/s, and considering 20 ms bins that contain at most 20 spikes (only one lookup table is ever needed, so long as its firing-rate range exceeds that of the most-active neuron at the most active moment in $\Omega$). Now suppose we are observing a population of 200 neurons, with a 200 ms history divided into ten 20 ms bins. For each library state, the log-likelihood of the observed spike-counts is simply the sum of 200 × 10 = 2000 individual log-likelihoods, each retrieved from the lookup table. In practice, computation is even simpler because many terms can be reused from the last time bin using a recursive solution (Methods). This procedure is lightweight and amenable to real-time applications. The assumption of locally stereotyped trajectories also enables neural states (and decoded behaviors) to be updated between time bins. While awaiting the next set of spiking observations, MINT simply assumes that each neural state advances deterministically according to the flow-field implied by the trajectories.

To decode stereotyped trajectories, one could simply obtain the maximum-likelihood neural state from the library, then render a behavioral decode based on the behavioral state with the same values of $c$ and $k$. This would be appropriate for applications in which conditions are categorical, such as typing or handwriting. Yet in most cases we wish for the trajectory library to serve not as an exhaustive set of possible states, but as a scaffolding for the mesh of possible states. MINT's operations are thus designed to estimate any neural trajectory – and any corresponding behavioral trajectory – that moves along the mesh in a manner generally consistent with the trajectories in $\Omega$. To illustrate, suppose the subject made a reach that split the difference between the purple and blue reach conditions in

*Figure 3a*. MINT identifies two candidate states with high likelihoods, along different neural trajectories (*Figure 3d*) in the library. MINT then interpolates between these high-likelihood states, and determines whether there exists a higher-likelihood state between them. MINT does so by considering a line segment of neural states parameterized by $\alpha$, ranging from 0 (at the purple state) to 1 (at the blue state). Because the log-likelihood of the observed spikes is a concave function of $\alpha$, the optimal neural state can be rapidly identified online using Newton's method. The decoded behavioral output is then produced by interpolation between the corresponding two behavioral states in $\Phi$, using the same $\alpha$ associated with the optimal neural state (*Figure 3e*).

The value of $\alpha$ may change with time. For example, this could occur if the true behavioral trajectory began close to the purple trajectory but became progressively more similar to the blue trajectory. The library trajectories that MINT interpolates between can also change with time. Thus, interpolation allows considerable flexibility. Not only is one not 'stuck' on a trajectory from $\Phi$, one is also not stuck on trajectories created by weighted averaging of trajectories in $\Phi$. For example, if cycling speed increases, the decoded neural state could move steadily up a scaffolding like that illustrated in *Figure 1b* (green). In such cases, the decoded trajectory might be very different in duration from any of the library trajectories. Thus, one should not think of the library as a set of possible trajectories that are selected from, but rather as providing a mesh-like scaffolding that defines where neural states are likely to live and the likely direction of their local motion. The decoded trajectory may differ considerably from any trajectory within $\Omega$. Nevertheless, individual decoded states are (empirically) close to states within $\Omega$, as will be discussed in a subsequent section. That proximity makes it reasonable to use the linear interpolation step when decoding behavior.

## Behavioral decoding when the ground-truth is known

We begin by assessing MINT's performance when applied to the activity of an artificial recurrent network of spiking neurons, trained to generate the empirical activity of the deltoid (*DePasquale et al., 2023*). Doing so provides a situation where the ground truth is known: we know the network's true output and that spiking is approximately Poisson, resembling that of cortical neurons (*DePasquale et al., 2023*). The network performed two tasks – reaching and cycling – governed by different local flow-fields in different regions of factor-space (as proposed by the flexibility-via-subspace hypothesis; *Churchland and Shenoy, 2024*). Indeed, the neural dimensions occupied during cycling are orthogonal to those occupied during reaching (*Figure 4b*). There are also moments when behavioral output – instantaneous muscle activity – is identical across tasks even though the underlying neural state is very different, in agreement with the empirical many-to-one mapping (*Figure 2c, d*). These features provide a test of whether MINT can handle geometry similar to that proposed in *Figure 1b*.

Decoded muscle activity (*Figure 4*) was virtually identical to network output (black). This was true during reaches (R3, R4, R6) and bouts of cycling (C). Decoding seamlessly transitioned between tasks. Decoding $R^2$ was .968 over ~7.1 min of test trials based on ~4.4 min of training data. This demonstrates that MINT performs well when data match its motivating assumptions, and also that it may be particularly useful in situations where decoding must automatically switch amongst tasks.

We leveraged this simulated data to compare MINT with a biomimetic decoder whose operations (unlike MINT) mimic the manner in which the true output is produced. The network's true output is simply a weighted sum of the spikes of 1200 neurons. We thus employed a linear readout (learned via ridge regression) leveraging all 1200 neurons. The resulting decode was very good: $R^2 = 0.982$ (and would have been unity if readout weights were inferred from connectivity rather than learned from activity). However, the decoding dimension – just like the actual output dimension – captured exceedingly little (~2%) of the total variance of network activity, a feature that is likely true of many real networks as well (*Russo et al., 2018*). Biomimetic decoding may therefore suffer when not all neurons are recorded. Indeed, when only using 5% of network neurons, performance of the biomimetic decoder dropped from $R^2 = 0.982$ to $R^2 = 0.773$. In contrast, MINT's performance declined only slightly from $R^2 = 0.968$ to $R^2 = 0.967$. This illustrates that, even when biomimetic decoding is possible, it may be outperformed by methods that leverage all neural dimensions rather than just output dimensions. MINT can also decode variables – e.g. kinematics – that are not literally encoded by network activity. These observations call into question an often-implicit assumption: that the activity-to-behavior 'model' used during decoding should attempt to approximate the true activity-to-behavior mapping. This assumption is presumably correct if a great many neurons can be recorded

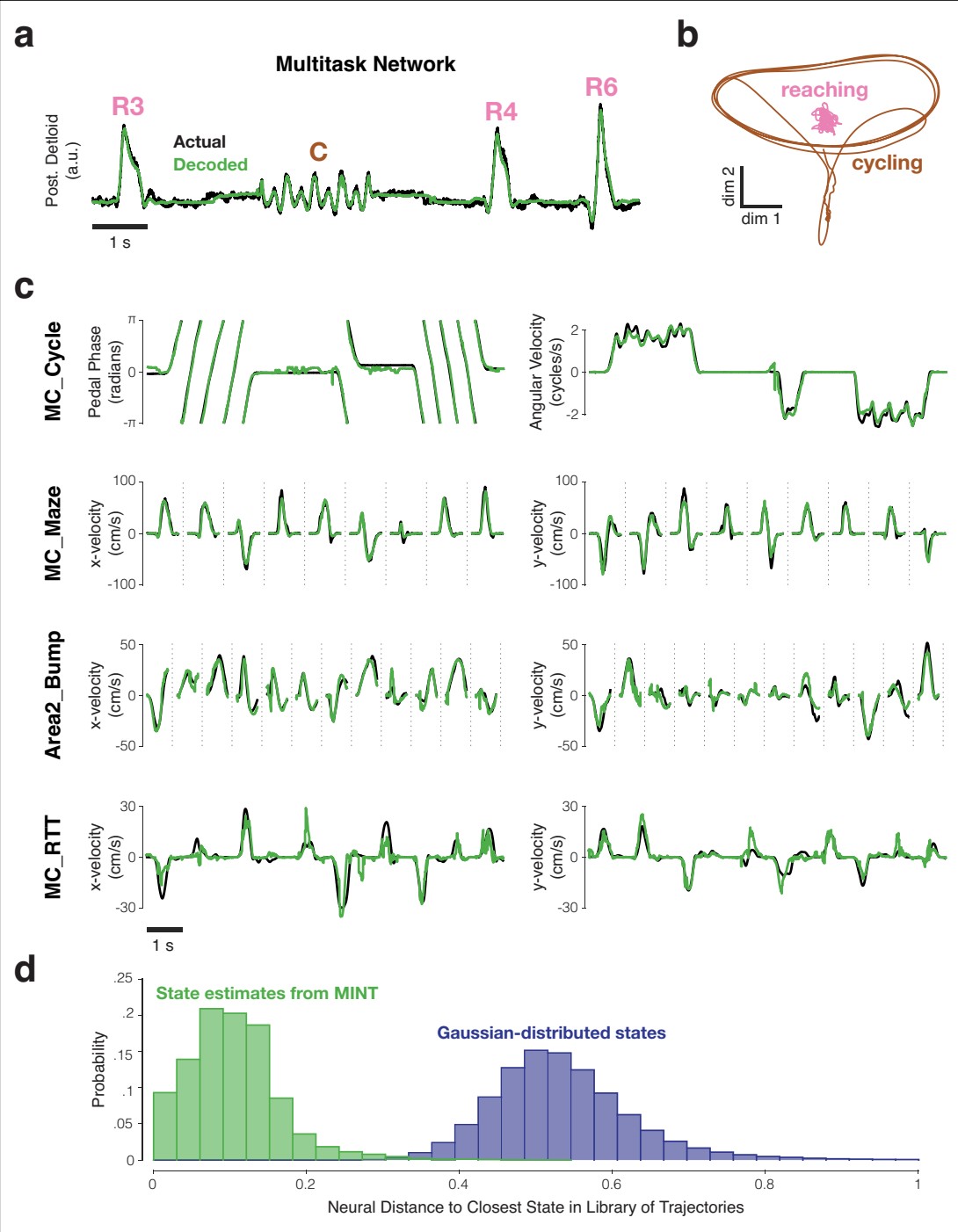

**Figure 4.** Examples of behavioral decoding provided by MINT for one simulated dataset and four empirically recorded datasets. All decoding is causal; only spikes from before the decoded moment are used. (**a**) MINT was applied to spiking data from an artificial spiking network. That network was trained to generate posterior deltoid activity and to switch between reaching and cycling tasks. Based on spiking observations, MINT approximately decoded the true network output at each moment. 'R3', 'R4', and 'R6' indicate three different reach conditions. 'C' indicates a cycling bout. MINT used no explicit task-switching, but simply tracked neural trajectories across tasks as if they were conditions within a task. (**b**) Illustration of the challenging nature, from a decoding perspective, of network trajectories. Trajectories are shown for two dimensions that are strongly occupied during cycling. Trajectories for the 8 reaching conditions (*pink*) are all nearly orthogonal to the trajectory for cycling (*brown*) and thus appear compressed in this projection. (**c**) Decoded behavioral variables (*green*) compared to actual behavioral variables (*black*) across four empirical datasets. MC_Cycle and MC_RTT show 10 seconds of continuous decoding. MC_Maze and Area2_Bump show randomly selected trials, demarcated by *vertical dashed lines*. (**d**) Distribution of distances between each decoded neural state and the nearest state in the library $\Omega$ (*green*), for the MC_Cycle dataset. To provide a reference, we drew neural states from a Gaussian distribution whose mean and covariance matched the library states, and computed distances to the library states (*purple*). This

*Figure 4 continued on next page*

*Figure 4 continued*

emulates what would occur if the manifold were defined by the data covariance and associated neural subspace. The *green* distribution mostly contains non-zero values, but is much closer to zero than the *purple* distribution. Thus, MINT rarely decodes a neural state that exactly matches a library state, but most decoded states are close to the scaffolding provided by the library states. Neural distances are normalized by the average pairwise distance between library states.

The online version of this article includes the following video for figure 4:

**Figure 4—video 1.** Video demonstrating causal neural state estimation and behavioral decoding from MINT on the MC_Cycle dataset.
https://elifesciences.org/articles/89421/figures#fig4video1

and the true output of the neural population is the quantity one wishes to decode – yet that is often not the case in real-world BCI applications.

## Behavioral decoding across multiple datasets

We investigated MINT's decoding performance across four datasets in which primates performed different motor tasks. All datasets included simultaneously collected spiking activity and behavioral measurements. The MC_Cycle dataset involved cycling a hand-held pedal and provided the motor cortex data in *Figure 2*. The MC_Maze dataset involved recordings from motor cortex during straight and curved reaches (*Churchland et al., 2010*). The Area2_Bump dataset involved recordings from somatosensory cortex (area 2) during active and passive center-out-reaches (*Chowdhury et al., 2020*). The MC_RTT dataset involved motor cortex recordings during point-to-point reaches with no condition structure (*Makin et al., 2018*). All decoding was offline, to facilitate comparison across methods using matched data. Many of these datasets are suspected to involve trajectories with properties matching those in *Figure 1b*. However, unlike for the network above, this is an empirical inference and is not known to be true for all datasets. It was thus unclear, a priori, how well MINT would perform. All decoding was causal; i.e. was based on spikes from before the decoded time. All decoding examples and performance quantifications utilized held-out test sets. Selection of MINT's (few) hyperparameters is documented in a later section.

For the MC_Cycle dataset (*Figure 4c*, top row), MINT's decode tracked pedal phase and angular velocity (also see *Figure 4—video 1*). When the pedal was stationary, erroneous deviations from non-zero angular velocity sometimes occurred (e.g. *Figure 4c*, just before the middle cycling bout) but were brief and rare: decoded angular speed was <0.1Hz for ~99% of non-moving times. Transitions between rest and cycling occurred with near-zero latency. During movement, angular velocity was accurately decoded and angular phase was decoded with effectively no net drift over time. This is noteworthy because angular velocity on test trials never perfectly matched any of the trajectories in $\Phi$. Thus, if decoding were restricted to a library trajectory, one would expect growing phase discrepancies. Yet decoded trajectories only need to locally (and approximately) follow the flow-field defined by the library trajectories. Based on incoming spiking observations, decoded trajectories speed up or slow down (within limits).

This decoding flexibility presumably relates to the fact that the decoded neural state is allowed to differ from the nearest state in $\Omega$. To explore, we computed, for each moment in the MC_Cycle dataset, the distance between the decoded neural state and the closest library state. Most distances were non-zero (*Figure 4d*, green distribution). At the same time, distances were much smaller than expected if decoded states obeyed a Gaussian with the empirical covariance (purple). These small distances are anticipated under the view that the empirical manifold (which produces the spiking observations that drive decoding) differs considerably in its shape from the distribution defined by the data's covariance matrix.

For the MC_Maze dataset (*Figure 4c*, second row), decoded velocity tracked actual reach velocity across a variety of straight and curved reaches (108 conditions). Decoded and actual velocities were highly correlated (*Figure 4c*, middle row, $R_x = 0.963 \pm 0.001$, $R_y = 0.950 \pm 0.002$; ranges indicate standard errors, computed by resampling test trials with replacement). For comparison with the other reach-like datasets below, we computed the mean absolute error (MAE) between decoded and actual velocity. Absolute errors were low ($MAE_x = 4.5 \pm 0.1$ cm/s, $MAE_y = 4.7 \pm 0.1$ cm/s), and constituted only ~2% of x- and y-velocity ranges (224.8 and 205.7 cm/s, respectively).

For the Area2_Bump dataset (*Figure 4c*, third row), correlations with horizontal and vertical hand velocity were also high: $R_x = 0.939 \pm 0.007$, $R_y = 0.895 \pm 0.015$. Mean absolute errors were small

(MAE$_x$ = 4.1 ± 0.2 cm/s, MAE$_y$ = 4.4 ± 0.2 cm/s) and similar to those for the MC_Maze dataset. In relative terms, errors were slightly larger for Area2_Bump versus MC_Maze because the range of velocities was smaller (95.8 and 96.8 cm/s for x- and y-velocity). Errors are thus slightly clearer in the traces plotted in *Figure 4c* (compare second and third rows).

Decoding was acceptable, but noticeably worse, for the MC_RTT dataset (*Figure 4c*, bottom row). In part this can be attributed to this dataset's slower reaches (86.6 and 61.3 cm/s x- and y-velocity ranges). When analyzing only movement periods (reach speed >1 cm/s), the median speed across all movement times was 38.1, 16.8, and 5.9 cm/s for MC_Maze, Area2_Bump, and MC_RTT, respectively. Thus, despite being modest in absolute terms (MAE$_x$ = 2.7 ± 0.1 cm/s, MAE$_y$ = 2.1 ± 0.1 cm/s), errors resulted in noticeably weakened correlations between decoder output and behavior: $R_x$ = 0.785 ± 0.013, $R_y$ = 0.843 ± 0.013. As will be discussed below, every decode method achieved its worst estimates of velocity for the MC_RTT dataset. In addition to the impact of slower reaches, MINT was likely impacted by training data that made it challenging to accurately estimate library trajectories. Due to the lack of repeated trials, MINT used AutoLFADS to estimate the neural state during training. In principle this should work well. In practice AutoLFADS may have been limited by having only ~10 min of training data. Because the random-target task involved more variable reaches, it may also have stressed the ability of all methods to generalize, perhaps for the reasons illustrated in *Figure 1b*. To aid interpolation amongst library trajectories (and possibly improve generalization), for this dataset we allowed MINT to consider multiple pairwise interpolations when finding the maximum-likelihood state. This aided performance, but only very modestly. We return to the issue of generalization in a later section.

## Comparison to other decoders

We compared MINT with four other decode algorithms: the Kalman filter, Wiener filter, feedforward neural network, and recurrent neural network (GRU). The Kalman filter and Wiener filter are historically popular BCI algorithms (*Carmena et al., 2003*; *Wu et al., 2003*) that are computationally simple and make linear assumptions. The feedforward neural network and GRU are expressive nonlinear function approximators that have been proposed for neural decoding (*Glaser et al., 2020*; *Wessberg et al., 2000*) and can be implemented causally. At each decoding time, both the feedforward neural network and the GRU leverage recent spiking history. The primary difference between the two is whether that history is processed all at once (feedforward network) or sequentially (GRU).

Each method was evaluated on each of the four datasets and was asked to decode multiple behavioral variables on held-out test sets (*Figure 5*). All decoders were provided with a trailing window of spiking observations, binned in 20 millisecond increments. The length of this trailing window was optimized separately for each decoder and dataset (and, like all hyperparameters, was chosen using validation data). For non-MINT decoders, hyperparameters were tuned, using Bayesian optimization, to maximize performance (*Snoek et al., 2012*). MINT's few hyperparameters were less extensively optimized. For example, window length was optimized just once for a given dataset for MINT, rather than separately for each set of behavioral variables as was done for most other decoders (excepting the Kalman filter). Minimal hyperparameter optimization embraces an unusual and useful aspect of MINT: there is no need to retrain if one wishes to decode a different behavioral variable. Once the neural state has been estimated, all behavioral variables are readily decodable. In principle, less-extensive optimization could have put MINT at a relative disadvantage. In practice, MINT is robust across reasonable hyperparameter choices (*Figure 5—figure supplement 1*); any disadvantage is thus likely very slight. By necessity and desire, all comparisons were made offline, enabling benchmarked performance across a variety of tasks and decoded variables, where each decoder had access to the exact same data and recording conditions.

There were 15 total 'behavioral groups' across the four datasets. For example, MC_Maze yielded two behavioral groups: position and velocity. We computed performance, for each group, by averaging across group members (e.g. across horizontal and vertical velocity). For every behavioral group, across all datasets, MINT's performance improved upon the 'interpretable' methods: the Kalman and Wiener filters. MINT's performance was typically comparable to, and often better than, that of the expressive GRU and feedforward neural network. MINT yielded the best performance, of all methods, for 11 of 15 behavioral groups. For 3 of the 4 behavioral groups where MINT's performance was not

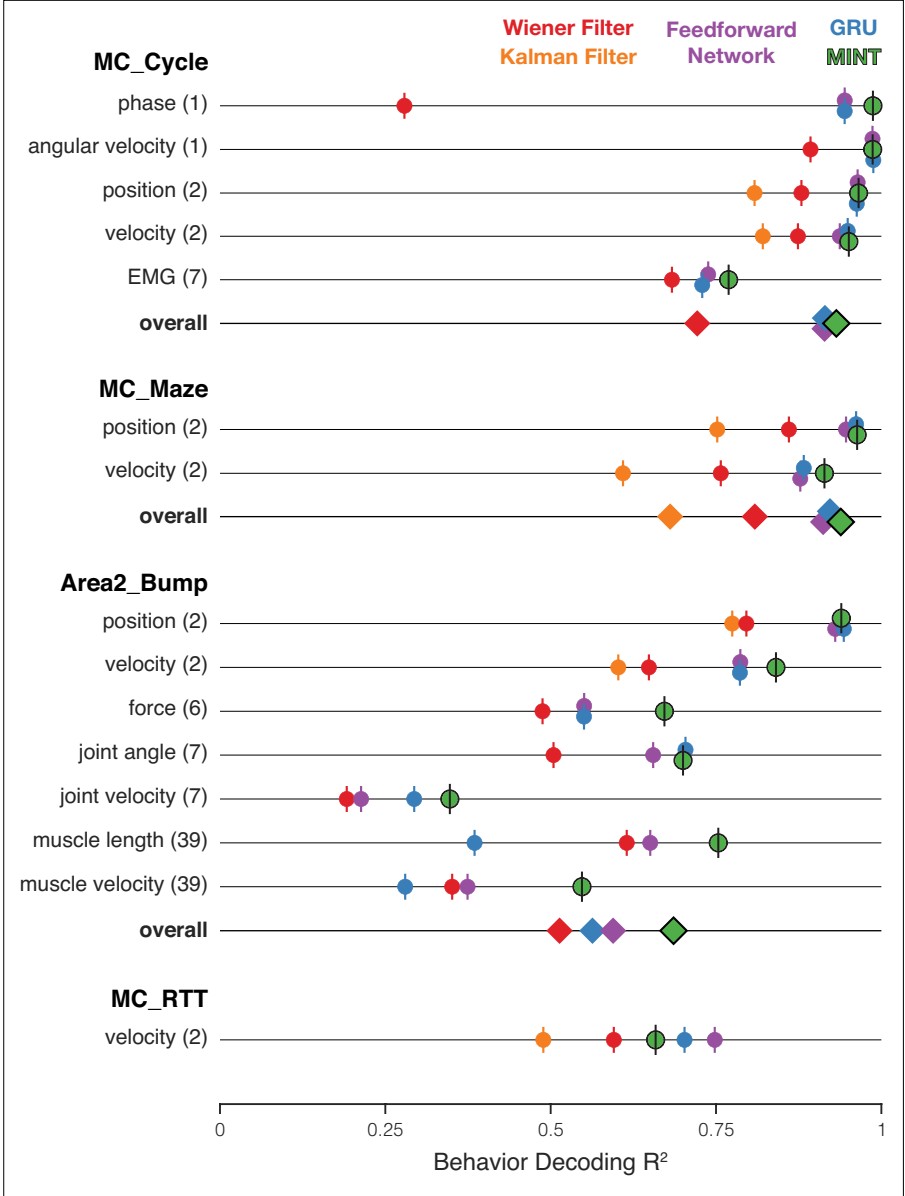

**Figure 5.** Comparison of decoding performance, for MINT and four additional algorithms, for four datasets and multiple decoded variables. On a given trial, MINT decodes all behavioral variables in a unified way based on the same inferred neural state. For non-MINT algorithms, separate decoders were trained for each behavioral group (with the exception of the Kalman filter, which used the same model to decode position and velocity). E.g. separate GRUs were trained to output position and velocity in MC_Maze. Parentheticals indicate the number of behavioral variables within a group. E.g. 'position (2)' has two components: x- and y-position. $R^2$ is averaged across behavioral variables within a group. 'Overall' plots performance averaged across all behavioral groups. $R^2$ values for feedforward networks and GRUs are additionally averaged across runs for 10 random seeds. The Kalman filter is traditionally utilized for position- and velocity-based decoding and was therefore only used to predict these behavioral groups. Accordingly, the 'overall' category excludes the Kalman filter for datasets in which the Kalman filter did not contribute predictions for every behavioral group. Results are based on the following numbers of training / test trials: MC_Cycle (174 train, 99 test), MC_Maze (1721 train, 574 test), Area2_Bump (272 train, 92 test), MC_RTT (810 train, 268 test). Vertical offsets and vertical ticks are used to increase visibility of data when symbols overlap.

The online version of this article includes the following figure supplement(s) for figure 5:

**Figure supplement 1.** MINT's decoding performance is robust to the choice of hyperparameters.

**Figure supplement 2.** Impact of different modeling and preprocessing choices on performance of MINT.

the highest, the deficit relative to the best method was negligible: $\Delta R^2 < 0.004$. Thus, in only 1 of 15 cases was MINT at a disadvantage relative to another decoder.

When decoding velocity, MINT provided a small but noticeable improvement over the expressive methods for two datasets (MC_Maze and Area2_Bump), was modestly worse for one dataset (MC_RTT), and was involved in essentially a three-way tie for one dataset (MC_Cycle). For all datasets where position was a behavioral group, MINT and both expressive methods provided extremely similar and very accurate performance, considerably better than that of the Wiener and Kalman filters. MINT tended to display the largest improvements, relative to the next-best decoder, when decoding muscle-related variables (EMG, force, muscle velocity, and muscle length) and phase. This is noteworthy because MINT used the same hyperparameters in all cases and was not re-optimized or re-trained across variables (unlike the other methods). Thus, the same MINT operations that produce a velocity decode naturally produce a position decode and a muscle-length decode.

For each dataset, we computed overall performance by averaging across behavioral groups. MINT had the best overall performance for three of four datasets: MC_Cycle, MC_Maze, and Area2_Bump. For two datasets (MC_Cycle and MC_Maze), MINT considerably outperformed the Kalman and Wiener filters, and had a tiny advantage over the expressive feedforward network and GRU. For the third (Area2_Bump), MINT's performance was noticeably better than all other methods.

The only dataset where MINT did not perform the best overall was the MC_RTT dataset, where it was outperformed by the feedforward network and GRU. As noted above, this may relate to the need for MINT to learn neural trajectories from training data that lacked repeated trials of the same movement (a design choice one might wish to avoid). Alternatively, the less-structured MC_RTT dataset may strain the capacity to generalize; all methods experienced a drop in velocity-decoding $R^2$ for this dataset compared to the others. MINT generalizes somewhat differently than other methods, and may have been at a modest disadvantage for this dataset. A strong version of this possibility is that perhaps the perspective in *Figure 1a* is correct, in which case MINT might struggle because it cannot use forms of generalization that are available to other methods (e.g. generalization based on neuron-velocity correlations). This strong version seems unlikely; MINT continued to significantly outperform the Wiener and Kalman filters, which make assumptions aligned with *Figure 1a*. Additionally, as will be described below, MINT's neural-state estimates for the MC_RTT dataset were not poor when compared with other methods (indeed, MINT excelled in this regard).

MINT's generally high level of decoding performance rests upon Bayesian computations: the estimation of data likelihoods and the selection of a decoded state that maximizes likelihood. A natural question is thus whether a simpler Bayesian decoder would have yielded similar results. We explored this possibility by testing a Naive Bayes regression decoder (*Glaser et al., 2020*) using the MC_Maze dataset. This decoder performed poorly, especially when decoding velocity ($R^2 = 0.688$ and $0.093$ for hand position and velocity, respectively), indicating that the modeling assumptions that differentiate MINT from a naive Bayesian decoder are important drivers of MINT's performance. We also investigated the possibility that MINT gained its performance advantage simply by having access to trial-averaged neural trajectories during training, while all other methods were trained on single-trial data. This difference arises from the fundamental requirements of the decoder architectures: MINT needs to estimate typical trajectories while other methods don't. Yet it might still be the case that other methods would benefit from including trial-averaged data in the training set, in addition to single-trial data. Alternatively, this might harm performance by creating a mismatch, between training and decoding, in the statistics of decoder inputs. We found that the latter was indeed the case: all non-MINT methods performed better when trained purely on single-trial data.

## How challenging is generalization?

The contrasting assumptions in *Figure 1a and b* yield different implications regarding generalization, illustrated in *Figure 6a and b*. In both views, generalization is possible when a newly observed neural trajectory is 'composed' of previously observed elements. However, the nature of those elements is very different. Under the view in *Figure 6a*, those elements are neural states within the manifold. New movements involve novel neural trajectories, but those trajectories are composed of states within the previously observed manifold. Because those neural states are assumed to correlate with behavioral variables (e.g. velocity) those variables can be decoded during novel movements.

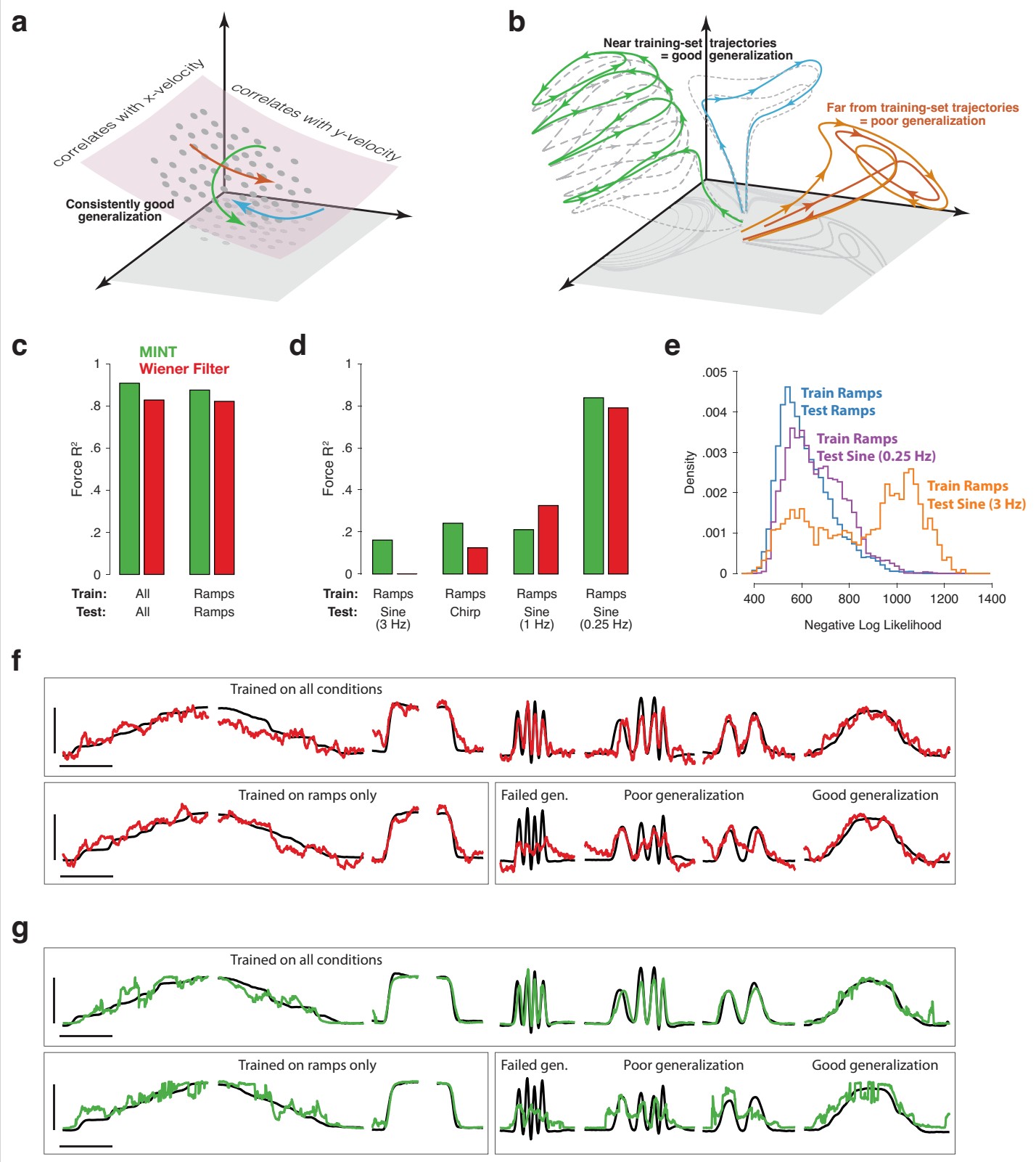

**Figure 6.** Implications of neural geometry for decoder generalization, in principle and in practice. (**a**) Under a traditional perspective, training that explores the full range of to-be-decoded variables will also explore the relevant neural manifold, allowing future generalization. *Gray dots* indicate neural states observed during training. *Colored traces* indicate newly observed trajectories. Because these trajectories traverse the previously defined manifold, they can be decoded by leveraging the established correlations between manifold dimensions and kinematics. (**b**) Under an emerging

*Figure 6 continued on next page*

*Figure 6 continued*

perspective, generalization is possible when new trajectories lie within a similar region of factor-space as training-set trajectories. *Gray-dashed traces* indicate training-set neural trajectories. *Colored traces* indicate newly observed trajectories. *Green* and *blue* trajectories largely follow the flow-field implied by training-set trajectories, allowing generalization. In contrast, the *orange* trajectory evolves far from any previously explored region, challenging generalization. This may occur even when behavioral outputs overlap with those observed during training. (**c**) Basic decoding performance on the MC_PacMan task for MINT (*green*) and a Wiener filter (*red*). Test sets used held-out trials from the same conditions as training sets. Force-decoding was excellent both when considering all conditions and when considering only ramps (fast and slow, increasing and decreasing). Results are based on the following numbers of training and test trials: All (234 train, 128 test), Ramps (91 train, 57 test). (**d**) Generalization when training employed the four ramp conditions and testing employed a sine or chirp. Results are based on 91 training trials (all cases) and the following numbers of test trials: Ramps (57), 0.25 Hz Sine (19), 1 Hz Sine (12), 3 Hz Sine (15), Chirp (25). (**e**) Data likelihoods indicate strained generalization. We computed the log-likelihood of the spiking observations for each neural state decoded by MINT. Rightward values indicate that those observations were unlikely even for the state selected as most likely to have generated those observations. Training employed the four ramp conditions. Distributions are shown when test data employed ramps, a 0.25 Hz sine, or a 3 Hz sine. (**f**) Examples of force decoding traces (for held-out trials) when decoding using the Wiener filter. *Top row:* all conditions were used during training. *Bottom row:* training employed the four ramp conditions. Horizontal and vertical scale bars correspond to 2 s and 16 Newtons, respectively. (**g**) As above, but when decoding using MINT. Trials and scale bars are matched across (**f**) and (**g**). Representative trials were selected as those that achieved median decoding performance by MINT within each condition.

The online version of this article includes the following figure supplement(s) for figure 6:

**Figure supplement 1.** Illustration of why decoding will typically fail to generalize across tasks when neural trajectories occupy orthogonal subspaces.

While this scenario would be advantageous, empirical observations question its assumptions. The constraint of low tangling implies that it is not possible to traverse previously observed states in entirely new ways. Indeed, even when encouraged to do so under BCI control, monkeys are incapable of producing new neural trajectories that conflict with the flow-field implied by previously seen trajectories (*Athalye et al., 2023*; *Oby et al., 2024*). This suggests that newly observed trajectories will fall into two categories. First, as illustrated by the green and blue trajectories in *Figure 6b*, new trajectories may be composed of trajectory segments that largely obey the flow-field implied by previously observed trajectories. For example, the green trajectory climbs the scaffolding of previously observed single-speed trajectories (dashed gray traces), mimicking what might occur in a new situation where cycling speed increases steadily (*Saxena et al., 2022*). Second, newly observed trajectories may sometimes evolve in previously unobserved portions of state-space (*Figure 6b*, orange trajectories) where the flow-field has never been estimated. This could occur if the internal computation must change considerably (*Trautmann et al., 2022*), a scenario that may be difficult to anticipate externally.

Under *Figure 6a*, MINT would be at a disadvantage relative to traditional decode methods. In particular, the standard Wiener filter should generalize well because it leverages neuron-behavior correlations and is not limited by any assumptions about a flow-field. In contrast, MINT might often generalize poorly because it cannot leverage neuron-behavior correlations and is constrained by previously observed trajectories. For example, if the green trajectory in *Figure 6a* were in the library, then MINT would be predisposed against decoding the blue trajectory during generalization because the two trajectories approach one another with opposing derivatives. The Wiener filter would have no such limitation.

If the scenario in *Figure 6b* holds, generalization will hinge less on the decoder itself and more on how new neural trajectories relate to the previously observed scaffolding. When newly observed trajectories hew close to that scaffolding (e.g. the green and blue trajectories) any good decode method should generalize well. When newly observed trajectories are far from the scaffolding (e.g. orange), all methods will struggle. MINT will struggle because it has no scaffolding in this region. Other methods will struggle because previously observed neuron-behavior correlations are no longer valid.

We desired a dataset where the predictions of *Figure 6a and b* are likely to be different. With the possible exception of MC_RTT, the datasets above lack this property; generalization to held-out trials is not particularly challenging and could be subserved by the features in *Figure 6a* or by decoding something like the green trajectory in *Figure 6b*. One way to challenge generalization, using existing data, is to restrict training to fewer conditions while ensuring that training still employs a broad range of values for each decoded variable. In the MC_Cycle dataset, the same set of x- and y-velocities occur during forward and backward cycling, just at different phases. If neural activity were dominated by correlations with velocity, traditional methods that leverage those correlations should generalize across cycling directions. Instead, we found that all tested methods failed to generalize when trained

on one cycling direction and tested on the other. Most $R^2$ values were close to zero or even negative. The likely reason is that neural trajectories for forward and backward cycling occupy nearly orthogonal subspaces (*Figure 6—figure supplement 1*), especially for this particular dataset (*Schroeder et al., 2022*). These observations are consistent with the view in *Figure 6b*. Of course, this did not have to be the case; the empirical neural trajectories during cycling could have obeyed the hypothesis in *Figure 6a* and thus supported generalization.

Whether one finds the above test conclusive depends upon which behavioral variables one considers primary. If only velocity (or only position) matters, then generalization should have been easy under *Figure 6a* because the same range of velocities (and positions) was present in training and test sets. Yet one may have different views regarding which variables are central. One may thus suspect that generalization failed because those variables (or their combinations) were insufficiently sampled during training. To address this concern, we explored generalization using a dataset where it is simple to confirm that the relevant behavioral variables are fully explored during training.

The MC_PacMan dataset involved Neuropixels-based recordings of motor cortex spiking activity and simultaneous measurements of isometric force applied to an immovable handle. All forces were generated in a forward direction (away from the body) and the arm did not move. Force is thus the central behavioral variable in this task; all other behavioral variables (including muscle activity and cursor height) mirror force or its derivative. Every condition explores a similar range of forces, yet involves very different temporal force profiles. Under the scenario in *Figure 6a*, generalization to new force profiles should be straightforward (for traditional methods) because neural states associated with each force level have already been observed. New force profiles are simply new trajectories through previously observed states. This scenario thus predicts that a simple traditional decoder, such as a Wiener filter, will generalize well but that MINT will not. In contrast, under the scenario in *Figure 6b*, some forms of generalization should be possible (for any method) while others will be intrinsically challenging (for any method).

To test these predictions, we trained both MINT and a Wiener filter to decode force. Training data included fast and slow increasing and decreasing ramps, and thus spanned the full range of forces used in this task. Generalization was tested using sinusoids (0.25, 2, and 3 Hz) and a slow-to-fast chirp. In agreement with *Figure 6b*, generalization performance was strongly condition-specific and only weakly decoder-specific (*Figure 6d, f, g*). Both MINT and the Wiener filter generalized well to the 0.25 Hz sinusoid: performance was almost as good as when training included all conditions. Yet both decoders generalized poorly at higher frequencies, especially the 3 Hz sinusoid. This was true even though the training data contained swiftly increasing and decreasing forces.

These empirical results make little sense under *Figure 6a*, because training data should have explored the relevant manifold. However, they do make sense under *Figure 6b*. Under this perspective, neural activity is not dominated by correlations with behavioral variables, but is shaped by the flow-field performing the underlying computation. Two conditions can involve different flow-fields, even if their behavioral outputs span similar ranges. Generalization will be strained in such situations, which may be difficult to anticipate 'from the outside'.

In instances where generalization is strained, and performance is poor overall, different decoders may experience performance drops of different, and perhaps difficult-to-predict, magnitudes. For example, MINT outperformed the Wiener filter when generalizing from ramps to the chirp, while the Wiener filter outperformed MINT when generalizing from ramps to the 1 Hz sine. This observation does not suggest a meaningful advantage for either decoder. These advantages were idiosyncratic and, more importantly, overall performance was so poor that either decoder would have been essentially unusable without retraining that included additional conditions. The MC_RTT task may also contain examples of this effect: the two expressive methods outperformed MINT but all decoders achieved their worst velocity-decode performance for this dataset (*Figure 5*, bottom), as did methods for estimating the neural state (see below).

For these reasons, one may wish to know, during decoding, if generalization is becoming strained. Usefully, MINT computes the log-likelihood of spiking observations for its estimated neural state (which is the most-probable state it can find). A prediction of *Figure 6b* is that, when generalization is strained, these likelihoods should become unusually low because the true neural state (which generated the spiking observations) will often be far from any state that MINT can estimate. To test

this prediction, we computed the distribution of negative log-likelihoods (*Figure 6e*) for all estimated neural states during the test of generalization described above. The distribution for 0.25 Hz sinusoids (purple) was similar to that for ramps (blue). This is consistent with the set of true neural states, during the 0.25 Hz sinusoids, being similar to states that are 'reachable' by MINT's interpolation step. In contrast, the distribution for 3 Hz sinusoids (orange) showed a second mode of spiking observations that were particularly unlikely. This is expected if the true neural state, at these moments, is far from any state that can be reached by MINT's interpolation (as would be true for the orange trace in *Figure 6b*). Data log-likelihoods could thus potentially be used to indicate – without having to ask the participant – when decoding is strained because neural states are far from those observed during training. Training could then be modified to include additional conditions.

## Modeling and preprocessing choices

MINT uses a direct mapping from neural states to behavioral states to instantiate highly nonlinear decoding, then uses locally linear interpolation to support decoding of behavioral states not observed during training. To determine the impact of these choices on performance, we ran a full factorial analysis that used neural state estimates generated by MINT but compared the direct neural-to-behavioral-state mapping to a linear neural-to-behavioral-state mapping (*Figure 5—figure supplement 2a*) and also assessed the impact of interpolation (*Figure 5—figure supplement 2b*). Both choices did indeed improve performance. We also assessed the impact of acausal versus causal decoding (*Figure 5— figure supplement 2c*). Causal decoding is important for real-time BCIs, and is thus employed for all our main decoding analyses. However, future applications could potentially introduce a small lag into online decoding, effectively allowing some 'acausal decoding' at the expense of longer-latency responses. As expected, acausal decoding provided modest performance increases, raising the possibility of an interesting tradeoff. Decoding can be zero-lag (causal), positive-lag (acausal), or even negative-lag. The latter would result in a very 'snappy' decoder that anticipated actions at the expense of some decoding accuracy. The lag hyperparameter wouldn't need to be specified prior to training and could be freely adjusted at any time.

All datasets were curated to contain sorted spikes. Yet during online performance, decoding must typically rely on unsorted threshold crossings. Prior studies have found that moving from sorted spikes to threshold crossings produces a modest but tolerable drop in performance (*Christie et al., 2015*). We similarly found that moving from sorted spikes to threshold crossings (selected to be from 'good' channels with reasonable signal-to-noise) produced only a small drop in performance (*Figure 5— figure supplement 2d*). The operations of MINT highlight why such robustness is expected. When spikes from two neurons are combined, the resulting 'unit' acts statistically as if it were a high firing-rate neuron. Spiking is still approximately Poisson, because the sum of Poisson processes is a Poisson process. Thus, MINT's statistical assumptions remain appropriate.

## Neural state estimation

MINT's success at behavioral decoding suggests underlying success in neural-state estimation. However, that would not necessarily have to be true. For example, the GRU also typically supported excellent behavioral decodes but doesn't provide neural state estimation. MINT does (by its very construction) but whether those estimates are accurate remains to be evaluated.

To evaluate neural-state estimates, we submitted firing-rate predictions to the Neural Latents Benchmark (*Pei et al., 2021*) an online benchmark for latent variable models that compares acausal neural-state estimates across methods and datasets. The four primary datasets for the benchmark are MC_Maze, Area2_Bump, MC_RTT, and an additional dataset, DMFC_RSG, that contains neural recordings from dorsomedial frontal cortex while a monkey performed a cognitive timing task (*Sohn et al., 2019*). Three secondary datasets (MC_Maze-L, MC_Maze-M, and MC_Maze-S) contain additional sessions of the maze task with progressively fewer training trials (500, 250, 100, respectively).

By submitting to the Neural Latents Benchmark, we can ask how MINT performs relative to a set of 'baseline' methods. One baseline method (smoothed spikes) is traditional and simple. Other baseline methods, Gaussian Process Factor Analysis (GPFA; *Yu et al., 2009*) and Switching Linear Dynamical System (SLDS; *Fox et al., 2008*), are modern yet well-established. Finally, two baseline methods are quite new and cutting-edge: Neural Data Transformers (NDT; *Ye and Pandarinath, 2021*) and

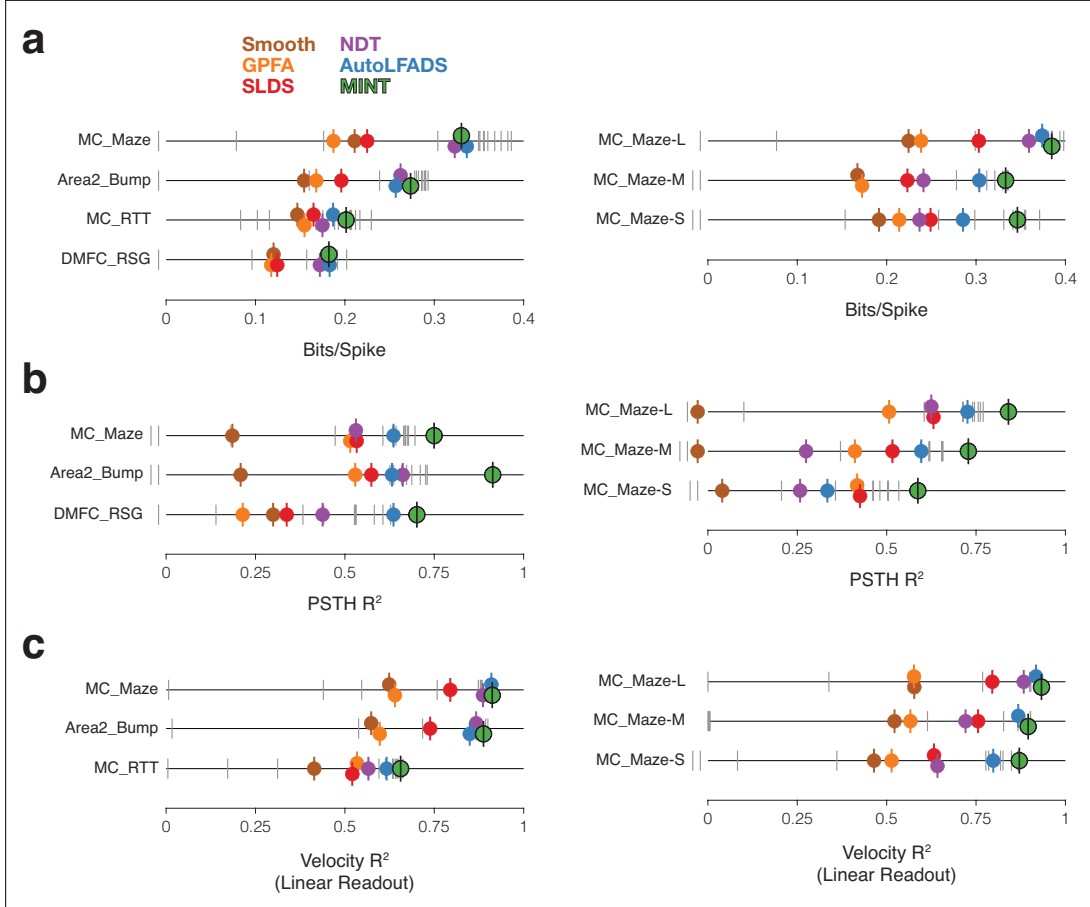

**Figure 7.** Evaluation of neural state estimates for seven datasets. (**a**) Performance quantified using bits per spike. The benchmark's baseline methods have colored markers. All other submissions have gray markers. Vertical offsets and vertical ticks are used to increase visibility of data when symbols are close due to similar values. Results with negative values are designated by markers to the left of zero, but their locations don't reflect the magnitude of the negative values. Data from Neural Latents Benchmark (https://neurallatents.github.io/). (**b**) Performance quantified using PSTH $R^2$. (**c**) Performance quantified using velocity $R^2$, after velocity was decoded linearly from the neural state estimate. A linear decode is used simply as a way of evaluating the quality of neural state estimates, especially in dimensions relevant to behavior.

AutoLFADS (*Keshtkaran et al., 2022*). These are expected to provide a high bar by which to judge the performance of other approaches.

Although the neural state is a well-defined quantity in spiking models (*DePasquale et al., 2023*), experimental data does not typically provide a ground-truth neural state against which estimates can be compared (unlike for behavioral-state decoding). Yet one can define attributes that an accurate neural-state decode should possess. A critical attribute is prediction of spiking activity. This attribute is assessed via bits per spike, which is closely related to the log-likelihood of spiking observations given a set of firing rates (assuming Poisson spiking). MINT performed similarly to AutoLFADS and was consistently slightly better than NDT on the four primary datasets (*Figure 7a*, left). MINT performed slightly better than AutoLFADS and NDT on the largest of the secondary datasets. That advantage grew as dataset-size shrank (*Figure 7a*, right). MINT outperformed the other three baseline methods by modest-to-sizeable margins for all seven datasets.

MINT was not designed to maximize bits per spike – good performance in this regard simply results from the primary goal of estimating a neural state containing firing rates. This matters because there exist spiking features – for example, synchrony beyond that due to correlated rates – that some methods may be able to leverage when predicting spike probabilities. Doing so is reasonable but separate from MINT's goal of estimating a rate-based neural state. It is thus worth considering other attributes expected of good neural-state estimates. The Neural Latents Benchmark includes two such attributes. First, if state estimation is accurate, trial-averaged neural state estimates should resemble

empirical trial-averaged firing rates (computed for test trials), an attribute assessed by PSTH $R^2$. MINT achieved higher PSTH $R^2$ values than every other method on every dataset (*Figure 7b*). This is in some ways unsurprising: MINT estimates neural states that tend to resemble (at least locally) trajectories 'built' from training-set-derived rates, which presumably resemble test-set rates. Yet strong performance is not a trivial consequence of MINT's design. MINT does not 'select' whole library trajectories; PSTH $R^2$ will be high only if condition ($c$), index ($k$), and the interpolation parameter ($\alpha$) are accurately estimated for most moments.

A second attribute is that, in sensorimotor areas during motor tasks, behavior should be decodable from the neural state. The benchmark thus measured $R^2$ for a linear decode of velocity from the neural state. MINT outperformed all baseline approaches (*Figure 7c*) for all datasets. (MINT could of course have achieved even better decoding using its typical direct association, but that would have undermined the utility of this comparison).

In addition to the five baseline methods, the Neural Latents Benchmark solicited submissions from other methods and many were submitted. These submissions spanned a broad range in terms of performance (gray vertical lines in *Figure 7*), highlighting the challenging nature of this problem. Some submissions involved ensemble approaches, some of which outperformed both MINT and the baseline methods in terms of bits per spike. For the primary datasets, the highest bits per spike overall was achieved by an ensemble of SpatioTemporal Neural Data Transformers (*Le and Shlizerman, 2022*), achieving 0.386 bits per spike on the MC_Maze dataset, compared to 0.330 for MINT. For the smaller secondary datasets, the very best submissions provided minimal-to-modest improvement over MINT in terms of bits per spike (*Figure 7a*, right), and the majority performed less well.

The ability of some expressive neural networks (and some ensembles of these networks) to improve the bits per spike metric indicates there exists variability in the neural recordings that MINT does not capture. That variability could arise from behaviorally irrelevant fluctuations in the neural state, from synchrony based effects, or from instabilities in the neural recordings. These are 'real' things and it is thus valid for a method to leverage them to predict spiking when that is the goal. Yet as noted above, they may often be incidental to one's goals in estimating the neural state, highlighting the need to consider additional attributes such as PSTH $R^2$ and velocity $R^2$. For PSTH $R^2$ (*Figure 7b*), MINT outperformed all other methods for all six datasets. For velocity $R^2$ (*Figure 7c*), MINT had the best performance for three out of six datasets and was very close to the highest $R^2$ values for the remaining three datasets (the largest deficit was $\Delta R^2 = 0.012$).

**Table 1.** MINT training times and average execution times (average time it took to decode a 20 ms bin of spiking observations). To be appropriate for real-time applications, execution times will ideally be shorter than the bin width. Note that the 20 ms bin width does not prevent MINT from decoding every millisecond; MINT updates the inferred neural state and associated behavioral state between bins, using neural and behavioral trajectories that are sampled every millisecond. For all table entries (except MC_RTT training time), means and standard deviations were computed across 10 train/test runs for each dataset. Training times exclude loading datasets into memory and any hyperparameter optimization. Timing measurements taken on a Macbook Pro (on CPU) with 32GB RAM and a 2.3 GHz 8-Core Intel Core i9 processor. Training and execution code used for timing measurements was written in MATLAB (with the core recursion implemented as a MEX file). For MC_RTT, training involved running AutoLFADS twice (and averaging the resulting rates) to generate neural trajectories. This training procedure utilized 10 GPUs and took ~1.6 hr per run. For AutoLFADS, hyperparameter optimization and model fitting procedures are intertwined. Thus, the training time reported includes time spent optimizing hyperparameters.

| Dataset | Training Time (s) | Execution Time (ms) |
| --- | --- | --- |
| Area2_Bump | 4.8 ± 0.2 | 0.31 ± 0.03 |
| MC_Cycle | 20.3 ± 2.9 | 0.55 ± 0.03 |
| MC_Maze | 42.8 ± 0.6 | 2.26 ± 0.09 |
| MC_Maze-L | 10.8 ± 0.4 | 0.83 ± 0.02 |
| MC_Maze-M | 8.0 ± 0.4 | 0.80 ± 0.04 |
| MC_Maze-S | 5.8 ± 0.2 | 0.59 ± 0.07 |
| MC_RTT | ~3.2 hr | 6.99 ± 0.28 |

In summary, MINT's strong decode performance (*Figure 5*) does indeed follow from good neural-state estimation. MINT provided estimates that were competitive with the best present methods. This includes existing well-documented methods, as well as a bevy of additional submissions. This implies that MINT could be used in situations where neural-state estimation is the goal, possibly including monitoring of the neural state for clinical purposes. MINT uses simple operations and is readily implemented causally – indeed this is typically how we would expect to see it used. Yet MINT's performance is typically similar to – and often better than – that of complex methods that would likely have to be limited to offline situations.

## Practical considerations

### Training and execution times

For MINT, 'training' simply means computation of standard quantities (e.g. firing rates) rather than parameter optimization. MINT is thus typically very fast to train (*Table 1*), on the order of seconds using generic hardware (no GPUs). This speed reflects the simple operations involved in constructing the library of neural-state trajectories: filtering of spikes and averaging across trials. At the same time we stress that MINT is a method for leveraging a trajectory library, not a method for constructing it. One may sometimes wish to use alternatives to trial-averaging, either of necessity or because they improve trajectory estimates. For example, for the MC_RTT task we used AutoLFADS to infer the library. Training was consequently much slower (hours rather than seconds) because of the time taken to estimate rates. Training time could be reduced back to seconds using a different approach – grouping into pseudo-conditions and averaging – but performance was reduced. Thus, training will typically be very fast, but one may choose computationally intensive methods when appropriate.

Execution times were well below the threshold for real-time applications (*Table 1*), even using a laptop computer. State estimation in MINT is $\mathcal{O}(NT_{total})$, where $T_{total}$ is the total number of times (across conditions) in the library of neural trajectories. This is $\mathcal{O}(NC)$ when making the simplifying assumption that all conditions are of equal length. Memory requirements are similarly $\mathcal{O}((N+M)C)$, where $M$ is the number of behavioral variables (a value that will typically remain quite small). Thus, both execution times and memory requirements grow only linearly with neurons or with conditions. Additionally, much of the estimation procedure in MINT is condition-specific and therefore parallelizable. This makes MINT a good candidate for decoding in BCI applications that involve large neuron counts and/or large behavioral repertoires.

## MINT performs well on small training sets

To investigate robustness to training with small trial-counts, we leveraged the fact that the four maze-reaching datasets – one primary (MC_Maze) and three secondary (MC_Maze-L, MC_Maze-M, and MC_Maze-S) – contained training sets of decreasing size. Because training involved computing trial-averaged rates, performance is expected to decline when fewer trials are available for averaging. For comparison, we chose the GRU and the feedforward neural network because of their generally good position and velocity decoding performance in the MC_Maze task. We assessed performance in terms of position and velocity $R^2$ (*Figure 8a*). MINT always performed at least as well as the two expressive methods. This gap was often (though not always) larger for smaller training sets. More broadly, MINT was quite robust to small training sets: an approximately fivefold reduction in training data led to a decline in $R^2$ of only 0.037 (position) and 0.050 (velocity).

## Performance when neurons are lost

It is desirable for a decoder to be robust to the unexpected loss of the ability to detect spikes from some neurons. Such loss might occur while decoding, without being immediately detected. Additionally, one desires robustness to a known loss of neurons / recording channels. For example, there may have been channels that were active one morning but are no longer active that afternoon. At least in principle, MINT makes it very easy to handle this second situation: there is no need to retrain the decoder, one simply ignores the lost neurons when computing likelihoods. This is in contrast to nearly all other methods, which require retraining because the loss of one neuron alters the optimal parameters associated with every other neuron.

To evaluate, we performed a neuron-dropping analysis using the MC_Maze-L dataset (*Figure 8b*). For comparison, we also evaluated a Wiener filter. The Wiener filter provides a useful comparison

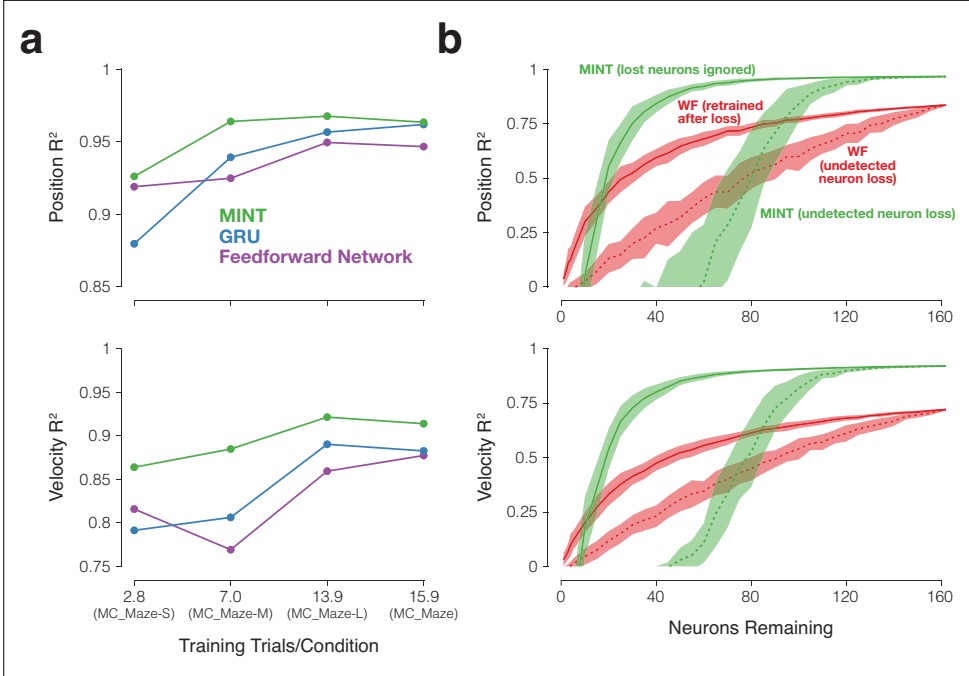

**Figure 8.** Decoding robustness in the face of small training sets and reduced neuron counts. (**a**) $R^2$ values for MINT and two neural network decoders (GRU and the feedforward network) when decoding position and velocity for four maze datasets with progressively fewer training trials per condition. Results are based on the following numbers of training and testing trials: MC_Maze-S (75 train, 25 test), MC_Maze-M (188 train, 62 test), MC_Maze-L (375 train, 125 test), MC_Maze (1721 train, 574 test). MC_Maze contains 108 conditions and the other maze datasets each contain 27 conditions. (**b**) Decoding performance in the face of undetected (*dashed traces*) and known (*solid traces*) loss of neurons. $R^2$ values are shown for MINT (*green*) and the Wiener filter (*red*) when decoding position and velocity from the MC_Maze-L dataset (same train/test trials as in **a**). To simulate undetected neuron loss, we chose $k$ neurons (randomly, with $k$ being the number of lost neurons out of 162 total) and eliminated their spikes without adjusting the decoders. This procedure was repeated 50 times for each value of $k$. Traces show the mean and standard deviation of the sampling distribution (equivalent to the standard error). To simulate known neuron loss, we followed the above procedure but altered the decoder to account for the known loss. For MINT, this simply involved ignoring the lost neurons when computing data likelihoods. In contrast, the Wiener filter had to be retrained using training data with those neurons eliminated (as would be true for most methods). This was done separately for each set of lost neurons.

The online version of this article includes the following figure supplement(s) for figure 8:

**Figure supplement 1.** The Kalman filter's relative performance would improve if neural data had different statistical properties.

**Figure supplement 2.** MINT is a modular algorithm amenable to a variety of modifications and extensions.

because (like MINT) it is easy to retrain quickly, with reliable results, across many training sets (which was critical in the present situation). To simulate undetected neuron loss, we chose a random subset of neurons and removed all their spikes, without making any adjustment to the decoders. This simulates a neuron (or channel) falling silent. This was repeated many times with different random draws. To simulate known neuron loss, we did the same but allowed the decoders to adjust: by ignoring the lost neurons in the case of MINT, and by retraining all parameters in the case of the Wiener filter. We evaluated both position (*Figure 8b*, top) and velocity (bottom) decoding.

For MINT, undetected loss of fewer than 20 (of 162) neurons caused essentially no performance deficit. Average position and velocity decoding $R^2$ remained above 95% of peak performance until fewer than 113 and 109 neurons remained, respectively. Large undetected losses – for example of half the neurons – caused major deficits in decoding. Yet decoding remained accurate if those losses were known, with the lost neurons ignored during decoding. MINT was extremely robust in this situation: MINT's average $R^2$ for position decoding remained above 95% of peak performance until fewer than 58 neurons (36%) remained. Results were similar for velocity decoding: performance was >95% of its

peak until fewer than 62 neurons (38%) remained. The Wiener filter was also fairly robust, but less so than MINT. To remain above 95% of peak performance, the retrained Wiener filter needed approximately twice as many remaining neurons: 118 for position and 126 for velocity. When not retrained, Wiener-filter performance dropped steadily without a clear plateau for modest neuron loss.

These results illustrate that MINT is reasonably robust to undetected neuron loss. In practice, the loss of many neurons would likely be detected. In such cases, MINT is very robust. Because there is no need for any true retraining, adjustment to known neuron-loss can occur on the fly with no need to pause decoding.

## Interpretability

One form of interpretability relates to whether an algorithm uses 'black box' optimization to learn a solution. For example, neural networks can be highly expressive and learn constraints directly from data. However, when constraints are known, one may prefer to rely more on explicit assumptions and less on expressivity. Interpretable methods, such as the Kalman filter, rely on explicit assumptions. Such assumptions may (if they match the data) improve performance. They may also provide other potential advantages. For example, the Kalman filter is a canonical 'interpretable' algorithm, with a probabilistic graphical model that makes an explicit Gaussian assumption regarding distributions of spike-count measurements. This explicitness may be helpful in understanding successes and failures (e.g. the Gaussian assumption can break down during low-rate periods).

The performance of interpretable methods should tend to reflect the extent to which their assumptions match the statistics of the data. To investigate, we considered the performance of both the Kalman filter and MINT on the MC_Maze reaching dataset. The Kalman filter produces a relatively poor decode ($R^2$ = 0.609 for hand velocity, versus 0.914 for MINT). This does not imply the Kalman filter is a poor algorithm, simply that its assumptions may not be well-aligned with the data. We confirmed this by generating synthetic data whose properties are closer to those assumed by the Kalman filter. The Kalman filter then performed quite competitively (*Figure 8—figure supplement 1*). This result underscores that MINT does not outperform the Kalman filter because it is intrinsically a better method. Instead, MINT performs well because its assumptions are presumably a better match to the underlying properties of the data. If *Figure 1a* were the correct perspective, the Kalman filter would have been expected to perform at least as well as MINT. Under *Figure 1b* the Kalman filter is predicted to perform less well, which is indeed what we found.

MINT is also interpretable regarding why each prediction is made. Suppose that, at a given moment during decoding, MINT selects neural state $\mathbf{x}$ as the most likely state and renders its decode using the associated behavioral state $\mathbf{z}$. As analyzed in *Figure 6e* above, selection of $\mathbf{x}$ comes with an associated likelihood that conveys information regarding how well the spiking data matched that neural state. Likelihoods are known not only for this 'selected' state, but for all non-selected states on all trajectories in the library. These likelihoods can be analyzed to understand why the selected state was chosen over the others, how close the algorithm was to selecting a different state, and which neurons contributed most strongly to state selection. More generally, because of its probability-based approach, MINT is readily modifiable and extensible in a variety of ways (*Figure 8—figure supplement 2*).

## Discussion

### Implications regarding neural geometry

Our results argue that the empirical structure of motor-cortex activity is closer to the hypothesis in *Figure 1b*. We began with the prediction that, under that hypothesis, MINT should consistently outperform existing interpretable methods such as the Kalman and Wiener filters. This was indeed the case: MINT always performed better than these two methods, often by sizable margins. In contrast, MINT and the Kalman filter performed comparably on simulated data that better approximated the assumptions in *Figure 1a*. Thus, MINT is not a 'better' algorithm – simply better aligned with the empirical properties of motor cortex data. This highlights an important caveat. Although MINT performs well when decoding from motor areas, its assumptions may be a poor match in other areas (e.g. the hippocampus). MINT performed well on two non-motor-cortex datasets – Area2_Bump (S1) and DMFC_RSG (dorsomedial frontal cortex) – yet there will presumably be other brain areas and/

or contexts where one would prefer a different method that makes assumptions appropriate for that area.

We also began with the prediction that, under the hypothesis in *Figure 1b*, MINT should be competitive with highly expressive approaches that can learn their constraints from data. MINT may even perform better in some cases because MINT can begin with appropriate constraints rather than having to learn them from training data. At the same time, MINT might sometimes be at a disadvantage if there are additional constraints or features that it does not take into account (a simple example would be drift in firing rates over time – one would desire a drift stabilization module to correct for this, as in *Figure 8—figure supplement 2*). Empirically, decoding performance was often exceedingly similar for MINT and both expressive methods (the feedforward network and GRU). Consider velocity, which is probably the most commonly decoded parameter for motor BCIs. Across the four datasets in *Figure 5*, the average $R^2$ for the velocity decode was 0.841 (MINT), 0.830 (GRU), and 0.837 (feedforward network). These values are all considerably higher than for the Kalman filter (0.630) and Wiener filter (0.719). There were multiple cases where MINT performed noticeably better than either expressive method, and one case (the MC_RTT dataset) where MINT performed noticeably worse. These results are consistent with the set of expectations outlined above: MINT should be highly competitive with expressive methods, often performing modestly better and sometimes performing modestly worse.

The hypothesis that MINT's assumptions are well-aligned with the data is further supported by its performance during neural-state estimation. Of the well-established baseline methods for state-estimation, the highest performing were AutoLFADS and a Neural Data Transformer (NDT). Across a variety of datasets and performance metrics, MINT either performed similarly to these methods or performed noticeably better. MINT consistently outperformed simple lightweight state estimators, despite being a simple lightweight estimator itself. We find it noteworthy that MINT naturally performs very well as a neural-state estimator, despite being designed as a decoder of behavior. MINT does so with no need for modification. This relates to the observation that MINT can readily switch from decoding one parameter (e.g. cycling phase) to a very different one (muscle activity) with no need for additional training, an option not available with any other standard method.

## Practical considerations for use of MINT as an online decoder

Across the four datasets we tested, the best decoders (MINT and the two expressive methods) were all very close in terms of performance. Given this, decoder-choice would likely be based on considerations beyond raw performance. Here MINT has many advantages. Because MINT's computations are so simple, it is typically fast to train and always very fast to run. Its interpretability supports principled extensions (discussed below) and the ability to investigate the source of any failures (as in *Figure 6e*). These considerations, combined with the fact that MINT often yielded the best behavioral decode of all methods, makes MINT a compelling candidate for online decoding in scientific or clinical applications. With that goal in mind, there exist three important practical considerations. First, some decode algorithms experience a performance drop when used online. One presumed reason is that, when decoding is imperfect, the participant alters their strategy which in turn alters the neural responses upon which decoding is based. Because MINT produces particularly accurate decoding, this effect may be minimized, but this cannot be known in advance. If a performance drop does occur, one could adapt the known solution of retraining using data collected during online decoding (*Gilja et al., 2012*). Another presumed reason (for a gap between offline and online decoding) is that offline decoders can overfit the temporal structure in training data (*Deo et al., 2023*). This concern is somewhat mitigated by MINT's use of a short spike-count history, but MINT may nevertheless benefit from data augmentation strategies such as including time-dilated versions of learned trajectories in the libraries.

A second practical consideration is that, for a participant with limited ability to move, training would involve asking subjects to observe and/or attempt movements. Both strategies engage neurons in motor cortex (*Vargas-Irwin et al., 2018*), but can also create uncertainty regarding the exact behavioral label for each moment (a challenge any decode method must face). MINT's decoding performance benefits from good estimates of $\Omega$ and $\Phi$, and will decline if those estimates are poor or misaligned. Fortunately, methods exist for temporally aligning neural data based on spiking activity alone (*Williams et al., 2020*) and the performance of such methods should only improve as more neurons are simultaneously recorded. Such methods would allow computation of trial-averaged rates

(or single-trial rates) that could then be aligned with a canonical movement trajectory (e.g. the usual bell-shaped velocity profile). Conveniently, any alterations to that alignment are trivial to accomplish, including at runtime. For example, to shift velocity earlier, one would simply shift MINT's library indices. More generally, MINT is amenable to a variety of methods – simple or sophisticated – for inferring the trajectory library. Possible future approaches include leveraging the typical neural geometry for a particular task. This could allow a participant-specific library to be learned using fewer trials. This might even be done in an unsupervised way (similar to *Dyer et al., 2017*): a trajectory library could be built without behavioral labels, then appropriately linked to a 'reference' behavioral library via prior knowledge of task-specific neural geometry. As a simple example, cycling across different speeds produces a tube-shaped manifold with different velocities at each end (*Saxena et al., 2022*). An empirically observed tube might be given behavioral labels based on this knowledge.

A final practical consideration concerns generalization. Because MINT uses a library of canonical trajectories, it is tempting to suppose that its decoded trajectories will simply be replicas of a limited set of stereotyped behaviors. By removing the interpolation step, MINT can indeed approximate this mode, which would be desirable in situations where the to-be-decoded behavior is itself stereotyped (e.g. producing keystrokes or handwritten characters). Yet one often wishes to decode behaviors beyond those observed in the training set. MINT can do so if the empirical neural trajectory mostly (and locally) obeys the flow-field implied by the library. This provides considerable, but not unlimited, flexibility. Might MINT thus be at a disadvantage, relative to other methods, with respect to generalization? This is predicted to be true under the hypothesis in *Figures 1a and 6a*. However, it is not expected to be true under the hypothesis in *Figures 1b and 6b*. Under that hypothesis, the forms of generalization that are unavailable to MINT will also be unavailable to standard methods. Consistent with that prediction, generalization success for the MC_PacMan dataset was condition-specific, not decoder-specific.

## Multi-task decoding

The fact that generalization may often be challenging does not imply that decoding across multiple tasks is a challenge. Indeed MINT is ideally suited for this situation. For example, MINT successfully decoded motor output for both tasks performed by the network in *Figure 4a*, and across all conditions in the MC_PacMan dataset (when trained on all conditions, *Figure 6c*). As long as trajectories relevant to both tasks are present in the library, 'task switching' occurs naturally and implicitly. Because MINT does not rely on correlations between neural activity and behavioral variables, task switching will not be impaired if those correlations change across tasks, as is anticipated under *Figures 1b and 6b*. As the number of situations decoders are expected to handle increases, this may be a useful property.

## Future avenues

MINT's probability-based framework is amenable to principled extensions. Rather than simply decode the state with the highest data likelihood, one could incorporate priors. Priors could be fixed or guided by incoming information (e.g. an eye tracker). Decoding could also focus on utility functions rather than probabilities. This would respect the fact that a participant may value avoiding certain types of errors more than others. For example, erroneous transitions from stationary to moving may be jarring or unsafe, and could be dissuaded via an appropriately designed utility function. These extensions are natural because MINT computes data likelihoods for a broad range of candidate states before choosing the maximum-likelihood state. Converting likelihoods to probabilities (based on a prior) or to an estimated cost can thus be done in a principled manner.

## Methods

### MINT (Part I): Estimating candidate states

#### Model of idealized trajectories

Let $\mathbf{s}_t \in \mathcal{S}^N$ where $\mathcal{S} = \{0, 1, ..., S\}$ be the measured spiking activity from $N$ neurons at time sample $t$. Spikes are associated with some underlying neural state $\mathbf{x}_t \in \mathbb{R}^N$ and a corresponding behavioral state $\mathbf{z}_t \in \mathbb{R}^M$ where $M$ is the number of behavioral variables (e.g. $x$- and $y$-velocity of the hand). We bin spiking activity in time such that $\bar{\mathbf{s}}_{i'} = \sum_{i=0}^{\Delta-1} \mathbf{s}_{\Delta i' - i}$. Given recent spiking history $\bar{\mathbf{s}}_{t' - \tau' : t'}$ (spike counts from the most recently completed bin $t' = \lfloor t/\Delta \rfloor$ and $\tau'$ previous bins), we are interested in inferring

posterior distributions over neural states $P(\mathbf{x}_t \mid \bar{\mathbf{s}}_{t'-\tau':t'})$ and over behavioral states $P(\mathbf{z}_t \mid \bar{\mathbf{s}}_{t'-\tau':t'})$. We can then perform *maximum* a posteriori estimation to generate candidate estimates for both the neural state and behavior. We introduce a model of the form:

$$\mathbf{x}_t \sim \text{Unif}(\Omega) \tag{1}$$

$$\mathbf{x}_{t-i-1} = g(\mathbf{x}_{t-i}), \ i \in \{0, 1, \dots, \tau - 1\} \tag{2}$$

$$\bar{\mathbf{s}}_{t'-i'} \sim \text{Pois}\left(\frac{1}{\Delta} \sum_{i=0}^{\Delta-1} \mathbf{x}_{\Delta(t'-i')-i}\right), \ i' \in \{0, 1, \dots, \tau'\} \tag{3}$$

$$\mathbf{z}_t = f(\mathbf{x}_t) \tag{4}$$

where $g : \Omega^+ \to \Omega^+$ is a state-transition lookup table, $f : \Omega \to \Phi$ is a lookup table that associates neural states with behavioral states, $\delta = t - \Delta t'$ is the number of samples elapsed since the last time bin completed, and $\tau = \delta + \Delta(\tau' + 1) - 1$ indicates the extent of the deterministic history for $\mathbf{x}_t$ (matching the extent of recent spiking history). This model has parameters $\Omega^+$ and $\Phi$. These parameters are described below, along with definitions for $g$ and $f$.

$\Omega^+$ is a set (library) of $C$ neural trajectories, with each trajectory consisting of an ordered set of neural states. Each neural state in the library is notated as $\bar{\mathbf{x}}_k^c$, where $c$ indexes trajectories and $k$ indexes locations along each trajectory (e.g. $c = 2$ and $k = 100$ indicates the 100th neural state along the second trajectory in the library). Each trajectory is presumed to contain the sequence of neural states that are traversed for a particular behavior. Thus, $k$ increasing along a neural trajectory corresponds to the passage of time during the execution of a behavior. *Equations 2 and 3* indicate that each $\mathbf{x}_t$ has a history of $\tau$ states that are responsible for generating recent spike count observations. However, the first $\tau$ states along each neural trajectory lack a complete state history. Thus, we learn the full $\Omega^+$, but in *Equation 1* we only consider a reduced set of neural states, $\Omega = \{\bar{\mathbf{x}}_k^c \in \Omega^+ \mid k > \tau\}$, because only these states have the necessary state histories. The state-transition lookup table is defined as follows,

$$g(\bar{\mathbf{x}}_k^c) = \bar{\mathbf{x}}_{k-1}^c \tag{5}$$

simply indicating that each neural state in the library has a recent history determined by the stereotyped ordering of neural states within each neural trajectory.

$\Phi$ is a set (library) of $C$ behavioral trajectories, where each trajectory consists of an ordered set of behavioral states. Each $\bar{\mathbf{x}}_k^c \in \Omega$ is associated with a behavioral state, $\bar{\mathbf{z}}_k^c$, via

$$f(\bar{\mathbf{x}}_k^c) = \bar{\mathbf{z}}_k^c \tag{6}$$

In other words, each neural state in the library is paired with a behavioral state for the same $c$ and $k$. Once the libraries of neural and behavioral trajectories have been learned, MINT's parameters are fully learned. $g$ will be determined by the ordering of states within each trajectory and $f$ will be determined by each state's $c$ and $k$ indices. There are a variety of methods one can employ for learning these trajectories. The methods used in this paper are described in the "Learning idealized trajectories" section. These trajectories can often be learned by averaging neural and behavioral data across repeated trials of the same movement, yielding $\bar{\mathbf{x}}_k^c$ and $\bar{\mathbf{z}}_k^c$ that are representative of the neural and behavioral states expected for a particular movement.

We assume $\bar{\mathbf{z}}_{k_1}^{c_1} = \bar{\mathbf{z}}_{k_2}^{c_2}$ if and only if $c_1 = c_2$ and $k_1 = k_2$. (We similarly assume $\bar{\mathbf{x}}_{k_1}^{c_1} = \bar{\mathbf{x}}_{k_2}^{c_2}$ if and only if $c_1 = c_2$ and $k_1 = k_2$.) This can be made trivially true by letting $c$ and $k$ be the first two behavioral variables. Thus, $f$ is invertible. This trick is mathematically convenient for subsequent derivations—it does not change the fact that multiple neural states can be associated with the same behavior as in *Figure 2*.

## Estimating candidate states

The prior in *Equation 1* ensures that $P(\mathbf{x}_t = \mathbf{x} \mid \bar{\mathbf{s}}_{t'-\tau':t'}) = 0$ if $\mathbf{x} \notin \Omega$. Every element of $\Omega$ can be written as $\bar{\mathbf{x}}_k^c$ for some $c$ and $k$, and the prior in *Equation 1* is uniform over the states in $\Omega$. *Equation 2* and *Equation 5* indicate that each $\bar{\mathbf{x}}_k^c$ has only one possible recent history, a history that is specified by the stereotyped ordering of neural states in the $c$-th trajectory. Thus, the posterior probabilities over neural states in $\Omega$ do not need to marginalize over multiple potential state histories. Rather, the probability associated with each $\bar{\mathbf{x}}_k^c$ is proportional to the likelihood of recently observed spikes, $\bar{\mathbf{s}}_{t'-i':t'}$,

given the unique history of neural states associated with $\overline{\mathbf{x}}_k^c$. Thus, the posterior probabilities over neural states in $\Omega$ can be written

$$P(\mathbf{x}_t = \overline{\mathbf{x}}_k^c \mid \overline{\mathbf{s}}_{t'-\tau':t'}) \propto \prod_{i'=0}^{\tau'} \prod_{n=1}^{N} h(\overline{s}_{n,t'-i'}, \lambda_{n,k-\delta-\Delta i'}^c) \tag{7}$$

$$\lambda_{n,j}^c = \frac{1}{\Delta} \sum_{i=0}^{\Delta-1} \overline{x}_{n,j-i}^c \tag{8}$$

where $h(s, \lambda) = \frac{\lambda^s e^{-\lambda}}{s!}$ (i.e. Poisson likelihoods, as dictated by *Equation 3*), and $\overline{s}_{n,i'}$ and $\overline{x}_{n,i}^c$ are the $n$-th components of $\overline{\mathbf{s}}_{i'}$ and $\overline{\mathbf{x}}_i^c$, respectively. Recall that $\delta$ is the number of time samples elapsed since the completion of the most recent time bin. A normalizing constant can convert *Equation 7* into full posterior probabilities. Computing this constant is straightforward because $\Omega$ is finite.

*Equations 4 and 6* allow the posterior over behavioral states to be written in terms of the posterior over neural states. $P(\mathbf{z}_t = \mathbf{z} \mid \overline{\mathbf{s}}_{t'-\tau':t'}) = 0$ if $\mathbf{z} \notin \Phi$, but for behaviors in $\Phi$ the posterior probabilities are

$$P(\mathbf{z}_t = \overline{\mathbf{z}}_k^c \mid \overline{\mathbf{s}}_{t'-\tau':t'}) = P(\mathbf{x}_t = f^{-1}(\overline{\mathbf{z}}_k^c) \mid \overline{\mathbf{s}}_{t'-\tau':t'}) \tag{9}$$

We can perform *maximum* a posteriori estimation on the log-transformed neural state posterior and read out the behavioral estimate directly.

$$\hat{\mathbf{x}}_t = \underset{\overline{\mathbf{x}}_k^c \in \Omega}{\operatorname{argmax}} \sum_{i'=0}^{\tau'} \sum_{n=1}^{N} \log h(\overline{s}_{n,t'-i'}, \lambda_{n,k-\delta-\Delta i'}^c) \tag{10}$$

$$\hat{\mathbf{z}}_t = f(\hat{\mathbf{x}}_t) \tag{11}$$

## Lookup table of log-likelihoods

One could proceed with querying *Equation 10* for all $\overline{\mathbf{x}}_k^c \in \Omega$, selecting the most probable neural state, then reading out the associated behavioral state with *Equation 11*. However, one may often wish to speed up this process, e.g. to deploy in a real-time application. In this section (and the following two sections), we describe a procedure for approximating *Equation 10* with considerably reduced computational burden.

Notice that the log-likelihood term in *Equation 10* only varies as a function of spike count and rate. Spike counts are finite and firing rates have limited dynamic ranges, which can be discretized, so it is possible to precompute and store these log-likelihoods in a lookup table in memory. Suppose the dynamic range of rates is given by $[\lambda_{min}, \lambda_{max}]$ and we sample this range with $V+1$ values such that $\tilde{\lambda}(v) = \lambda_{min} + \frac{v}{V}(\lambda_{max} - \lambda_{min})$ for $v \in \mathcal{V}$ where $\mathcal{V} = \{0, 1, \ldots, V\}$. Every rate $\lambda_{n,j}^c$ can now be approximated by $\tilde{\lambda}(v_{n,j}^c)$ for some $v_{n,j}^c \in \mathcal{V}$ that minimizes the approximation error. If we define a lookup table with entries $L(s, v) = \log h(s, \tilde{\lambda}(v))$, then *Equation 10* can be rewritten

$$\hat{\mathbf{x}}_t = \underset{\overline{\mathbf{x}}_k^c \in \Omega}{\operatorname{argmax}} \sum_{i'=0}^{\tau'} \sum_{n=1}^{N} L(\overline{s}_{n,t'-i'}, v_{n,k-\delta-\Delta i'}^c) \tag{12}$$

Thus, during training we can compute $L$ for all $s$ and $v$. Similarly, we can compute $v_{n,j}^c$ for all $c$, $n$, and $j$. This formulation ensures the costly process of computing log-likelihoods only needs to be performed once, during training.

## Recursive solution

The estimation procedure in *Equation 12* can be made faster still by exploiting recursive structure. Although the notation used in this section to carefully document this recursion is verbose to avoid any ambiguity, the concept is quite simple: the model of stereotyped trajectories reuses many log-likelihood terms across decoding timesteps that needn't be recomputed each time. Consider the following definitions.

$$q_{t,k}^c = \sum_{i'=0}^{\tau'} r_{t'-i',k-\delta-\Delta i'}^c \tag{13}$$

$$r_{j',j}^c = \begin{cases} \sum_{n=1}^{N} L(\bar{s}_{n,j'}, v_{n,j}^c), & j' > 0, j > 0 \\ 0, & \text{otherwise} \end{cases} \tag{14}$$

*Equation 13* states that $q_{t,k}^c$, the log-likelihood of the data given one is presently at moment $k$ in condition $c$, can be decomposed into a sum of time-bin-specific log-likelihoods (*Equation 14*). Those time-bin-specific log-likelihoods can again be decomposed into neuron-specific log-likelihoods that can be queried from a lookup table. The added case for non-positive $j'$ or $j$ is a minor implementation detail that covers edge cases (computing $q_{t,k}^c$ shortly after spiking data becomes available or for a $k$ near the beginning of a trajectory). *Equation 13* can be restated recursively,

$$q_{t,k}^c = \begin{cases} q_{t-\Delta,k-\Delta}^c + r_{t',k}^c - r_{t'-\tau'-1,k-\tau-1}^c, & \delta = 0 \\ q_{t-1,k-1}^c, & \delta > 0 \end{cases} \tag{15}$$

with initial conditions $q_{t,0}^c = q_{0,k}^c = 0$. In other words, when new spiking data arrives (corresponding to $\delta = 0$), each new log-likelihood $q_{t,k}^c$ can be computed by taking a previously computed log-likelihood associated with a nearby state, $q_{t-\Delta,k-\Delta}^c$, and updating it to remove a time-bin-specific log-likelihood associated with old data and add a time-bin-specific log-likelihood associated with new data. Given that $q_{t,k}^c$ is simply shifted by one index relative to the previous time step when $\delta > 0$ (i.e. no new spiking data is available, so state estimates are advanced deterministically using the model of stereotyped trajectories), the state estimate can be written

$$\hat{\mathbf{x}}_t = \begin{cases} \underset{\bar{\mathbf{x}}_k^c \in \Omega}{\operatorname{argmax}} \ q_{t,k}^c, & \delta = 0 \\ \bar{\mathbf{x}}_{\hat{k}+\delta}^{\hat{c}}, & \delta > 0 \end{cases} \tag{16}$$

where $\hat{c}$ and $\hat{k}$ are the indices such that $\hat{\mathbf{x}}_{t-\delta} = \bar{\mathbf{x}}_{\hat{k}}^{\hat{c}}$. Note that $\hat{\mathbf{x}}_t$ cannot be computed until $t > \tau$ (when sufficient spiking history has been collected).

## Querying fewer states

The approach described in *Equation 16* is efficient insofar as it renders a new estimate at each time $t$ while only requiring substantial computations every $\Delta$ time samples. Nevertheless, when $\delta = 0$, the recursion requires computing new $r_{t',k}^c$ and $r_{t'-\tau'-1,k-\tau-1}^c$ for every element in $\Omega^+$ to ensure that $q_{t,k}^c$ is defined for every $\bar{\mathbf{x}}_k^c \in \Omega$. However, we typically assume some smoothness in neural trajectories over time. Thus, for reasonably small $\Delta$, we can approximate the most likely neural state when $\delta = 0$ by only considering a subset of neural states defined by

$$\tilde{\Omega} = \left\{ \bar{\mathbf{x}}_k^c \in \Omega \ \middle| \ \frac{k}{\Delta} \in \mathbb{Z} \right\} \tag{17}$$

which is equivalent to downsampling the neural trajectories in $\Omega$ with $\Delta$ spacing between samples. Letting $k' = \frac{k}{\Delta}$, this choice yields a new expression for the neural state estimate,

$$\hat{\mathbf{x}}_t = \begin{cases} \underset{\bar{\mathbf{x}}_k^c \in \tilde{\Omega}}{\operatorname{argmax}} \ \tilde{q}_{t',k'}^c, & \delta = 0 \\ \bar{\mathbf{x}}_{\hat{k}+\delta}^{\hat{c}}, & \delta > 0 \end{cases} \tag{18}$$

where $\tilde{q}_{t',k'}^c = q_{\Delta t',\Delta k'}^c$. The recursion can now be performed over fewer states.

$$\tilde{q}_{t',k'}^c = \tilde{q}_{t'-1,k'-1}^c + r_{t',\Delta k'}^c - r_{t'-\tau'-1,\Delta k'-\tau-1}^c \tag{19}$$

This simplification reduces memory and execution time. Furthermore, by using interpolation (described in the next section) we can still estimate neural states outside of $\tilde{\Omega}$.

## Generating estimates at higher-than-bin-width resolution

*Equation 18* describes a procedure for generating new neural state estimates at every time sample. When $\delta = 0$, the estimate is updated using new spiking observations. When $\delta > 0$, the estimate is updated using the model of stereotyped trajectories, which specifies which neural state is likely to come next even in the absence of new spiking observations. However, to implement these equations in real time requires that the most time-consuming computations (those performed when $\delta = 0$) be performed within one time sample (e.g. 1 ms). MINT is very fast and in many cases will be capable of generating estimates in under a millisecond (*Table 1*). When this isn't possible, a small decoding lag can be introduced (e.g. 5 ms) such that the computations required when $\delta = 0$ can begin prior to needing the result.

## MINT (Part II): Interpolation

The algorithm described above leverages a library of neural trajectories composed of discrete sequences of neural states typically observed for a discrete set of conditions. These trajectories are presumed to sample an underlying neural manifold that is fundamentally continuous. The library of neural trajectories will more finely sample that manifold if one uses a large $C$ such that small variations in behavior each get their own neural and behavioral trajectories. Additionally, recall from the "Querying fewer states" section that, while decoding, only a subset of neural states (spaced apart by $\Delta$) are considered. The larger $\Delta$ is, the more likely it is that the neural state estimate will make a small error in time (estimating a neural state that is slightly ahead or slightly behind the true state along a trajectory). Thus, neural state estimation can be more accurate in time and reflect more behavioral variability if one uses a small $\Delta$ and a large $C$. However, the computational burden of the algorithm grows as $\Delta$ shrinks or $C$ grows. (The training data requirement also increases with $C$). This creates a potential trade-off between performance and computational burden. However, MINT side-steps this trade-off by using linear interpolation between states.

When the neural manifold is presumed to smoothly vary between neural states (across time or conditions) and there is also smooth variation between the corresponding behavioral states, it is unnecessary to use an overly small $\Delta$ and/or large $C$. Instead, one can identify candidate states along a small set of neural trajectories that coarsely sample the manifold, and then interpolate between those candidate states to find an intermediate state that is more consistent with the observed spikes. This two-step procedure (identify candidate states and then interpolate) allows the neural state estimate (and corresponding behavioral state estimate) to be accurate in time and capture variability between conditions (effectively as though a smaller $\Delta$ and larger $C$ had been used) while preserving the computational benefits of a reasonable $\Delta$ (e.g. 20 ms) and small $C$.

### Model of interpolated states

Suppose we identify two neural states, $\bar{\mathbf{x}}_{k_1}^{c_1}$ and $\bar{\mathbf{x}}_{k_2}^{c_2}$, and we'd like to consider the possibility that the actual neural state is somewhere between them. We can use a model of the following form:

$$\alpha \sim \text{Unif}(0, 1) \tag{20}$$

$$\mathbf{x}_{t-i} = (1 - \alpha)\bar{\mathbf{x}}_{k_1-i}^{c_1} + \alpha\bar{\mathbf{x}}_{k_2-i}^{c_2}, \; i \in \{0, 1, \ldots, \tau\} \tag{21}$$

$$\bar{\mathbf{s}}_{t'-i'} \sim \text{Pois}\left(\frac{1}{\Delta} \sum_{i=0}^{\Delta-1} \mathbf{x}_{\Delta(t'-i')-i}\right), \; i' \in \{0, 1, \ldots, \tau'\} \tag{22}$$

$$\mathbf{z}_t = (1 - \alpha)\bar{\mathbf{z}}_{k_1}^{c_1} + \alpha\bar{\mathbf{z}}_{k_2}^{c_2} \tag{23}$$

This model assumes that as the neural state smoothly varies from $\bar{\mathbf{x}}_{k_1}^{c_1}$ to $\bar{\mathbf{x}}_{k_2}^{c_2}$, the corresponding behavior smoothly varies from $\bar{\mathbf{z}}_{k_1}^{c_1}$ to $\bar{\mathbf{z}}_{k_2}^{c_2}$.

### Interpolating between states

Explicitly evaluating the probabilities associated with all $\mathbf{x}_t$ in this model would be computationally inefficient. Given that the prior on $\alpha$ is uniform over the range [0, 1], we can simply select the $\alpha$ that maximizes the log-likelihood of the observed spikes. The log-likelihood of the observed spikes is given by the equation:

$$\breve{q}_t(\alpha) = \log P(\bar{\mathbf{s}}_{t'-\tau':t'} \mid \alpha) = \sum_{i'=0}^{\tau'} \sum_{n=1}^{N} \log h\left(\bar{s}_{n,t'-i'}, (1-\alpha)\lambda_{n,k_1-\delta-\Delta i'}^{c_1} + \alpha\lambda_{n,k_2-\delta-\Delta i'}^{c_2}\right) \tag{24}$$

Differentiating $\breve{q}_t$ twice with respect to $\alpha$ yields

$$\frac{d^2\breve{q}_t}{d\alpha^2} = -\sum_{i'=0}^{\tau'} \sum_{n=1}^{N} \bar{s}_{n,t'-i'} \left(\frac{\lambda_{n,k_2-\delta-\Delta i'}^{c_2} - \lambda_{n,k_1-\delta-\Delta i'}^{c_1}}{(1-\alpha)\lambda_{n,k_1-\delta-\Delta i'}^{c_1} + \alpha\lambda_{n,k_2-\delta-\Delta i'}^{c_2}}\right)^2 \tag{25}$$

Notice that $\frac{d^2\breve{q}_t}{d\alpha^2} \leq 0$ always, $\breve{q}_t$ is a concave function of $\alpha$. Thus, we can rapidly compute $\hat{\alpha}$ using Newton's method and then estimate $\hat{\mathbf{x}}_t$ and $\hat{\mathbf{z}}_t$ via *Equations 21 and 23*. This procedure will yield the same $\hat{\alpha}$ for all $t$ associated with the same $t'$. Thus, the interpolation only needs to be performed once per time bin, when $\delta = 0$. In this paper, we used the following stopping criteria for Newton's method: (1) the estimate of $\alpha$ is within 0.01 of the estimate at the previous iteration, (2) the estimate of $\alpha$ saturates at 0 or 1, or (3) optimization runs for 10 iterations.

## Interpolating across indices

*Equation 17* restricts the neural states under consideration to be spaced apart by $\Delta$ indices. Thus, a straightforward use of the interpolation scheme described above is to interpolate between nearby neural states along the same trajectory to restore the precision in the estimate that was lost by this restriction. First, a candidate neural state, which we'll denote $\bar{\mathbf{x}}_{k_1}^{c_1}$, is identified using *Equation 18*. Then, a second candidate neural state is identified, which we'll denote $\bar{\mathbf{x}}_{k_2}^{c_1}$. This second state is whichever of the two adjacent states (either $\bar{\mathbf{x}}_{k_1+\Delta}^{c_1}$ or $\bar{\mathbf{x}}_{k_1-\Delta}^{c_1}$) has a larger log-likelihood of the observed spikes. Then, interpolation between $\bar{\mathbf{x}}_{k_1}^{c_1}$ and $\bar{\mathbf{x}}_{k_2}^{c_1}$ is performed as in the "Interpolating between states" section to yield neural and behavioral state estimates $\hat{\mathbf{x}}_t$ and $\hat{\mathbf{z}}_t$. This form of interpolation is less about generalization and more about enabling *Equation 17* to improve computational efficiency without loss of precision in the neural state estimates along each trajectory.

## Interpolating across conditions

To interpolate across conditions, we also identify candidate neural states. The first candidate neural state, $\bar{\mathbf{x}}_{k_1}^{c_1}$, is again identified using *Equation 18*. The second candidate neural state, $\bar{\mathbf{x}}_{k_2}^{c_2}$, is identified as the neural state that maximizes the log-likelihood of the observed spikes but is not a state along the same trajectory as $\bar{\mathbf{x}}_{k_1}^{c_1}$ (i.e. $c_2 \neq c_1$). Next, both $\bar{\mathbf{x}}_{k_1}^{c_1}$ and $\bar{\mathbf{x}}_{k_2}^{c_2}$ are each interpolated across indices with their adjacent states to yield improved candidate neural and behavioral state estimates. Then, an additional interpolation is performed between the improved candidate state estimates to yield the condition-interpolated neural and behavioral state estimates $\hat{\mathbf{x}}_t$ and $\hat{\mathbf{z}}_t$. This allows for generalization between similar behaviors.

## Other forms of interpolation

The exact manner in which interpolation is applied during MINT may vary by task or dataset. Often, interpolation across indices and conditions will be sufficient. However, in this section, we will review some alternative interpolation implementations that were used for various analyses throughout this paper.

First, a dataset like MC_RTT that lacks explicit condition structure will not necessarily have one trajectory per behavior. We used AutoLFADS to learn neural trajectories for MC_RTT, which would have yielded a single neural trajectory for the entire training set were it not for recording gaps that broke up that trajectory. Thus, different portions of the same trajectory frequently corresponded to similar movements, such that it made sense to consider interpolations between them. We therefore modified interpolation such that the second candidate state could be on the same trajectory as the first candidate state, but it had to be at least 1000 milliseconds away from that first state. Additionally, in a task with many degrees of behavioral freedom, there may be many trajectory fragments near a candidate neural state that could be worthwhile to interpolate to. Thus, for the MC_RTT dataset we expanded to multiple candidate neural states (6 for decoding analyses, 5 for the neural state estimation analysis; these values were optimized as hyperparameters). Interpolation occurred between all pairs of candidate neural states. Then, we proceeded to only use the interpolation that yielded the

highest spike count log-likelihoods. These modifications ensured that utilizing long trajectories spanning multiple behaviors in a reaching task with many degrees of behavioral freedom would not limit the ability of interpolation to generalize between similar behaviors.

Second, in DMFC_RSG we computed multiple libraries of neural trajectories, each library corresponding to a different section of the training data in which the overall firing rate statistics differed due to recording instabilities throughout the session. For this dataset, we wished to expand interpolation to occur across indices, conditions, *and* libraries. We simply extended the approach for interpolating across conditions one step further. The first candidate neural state was selected as the most likely neural state across all libraries. The second candidate neural state was selected as the most likely neural state in any library except the one in which the first candidate state resided. For each of these two candidate states, within-library interpolation across indices and conditions proceeded to improve the estimates. Then, a final interpolation occurred between the best estimate in the first library with the best estimate in the second library. This approach enabled MINT to robustly estimate the neural state even when fed test data from various sections of the dataset in which different recording instabilities were present.

Third, in MC_Cycle we decoded a circular variable: phase. To accommodate this circularity, phase interpolations were modified such that they always occurred across the lesser angle between the two phases. For example, if two candidate states corresponding to phases of 10 degrees and 350 degrees were identified, interpolation occurred across the 20 degree difference rather than the 340 degree difference.

## Datasets

We analyzed 9 empirical datasets (Area2_Bump, DMFC_RSG, MC_Cycle, MC_Maze, MC_Maze-L, MC_Maze-M, MC_Maze-S, MC_RTT, MC_PacMan), several simulated maze datasets, and an artificial multitask spiking network. For the MC_Cycle and MC_PacMan datasets, animal protocols were approved by the Columbia University Institutional Animal Care and Use Committee (Protocol number AC-AABE3550). For other datasets, protocols were approved by the relevant institutional committee for that institution. All empirical datasets contained spike times from offline or online sorted units. Detailed descriptions of all empirical datasets (except MC_Cycle and MC_PacMan) can be found in *Pei et al., 2021*. Here, we briefly describe each of the datasets.

### Area2_Bump

This dataset contains neural recordings (96-electrode Utah array) from Brodmann's area 2 of S1 (a proprioceptive area) while a monkey used a manipulandum to make center-out-reaches to one of eight targets. On some trials, the monkey volitionally reached to the targets ('active' trials). On other trials, the manipulandum perturbed the monkey's arm out toward one of the targets ('passive' trials). Thus, right after movement onset the 'active' trials involved predictable sensory feedback (the monkey planned the movement) and the 'passive' trials involved unexpected sensory feedback (the monkey was not planning to move). Between the eight targets and two trial types ('active' and 'passive'), there were a total of 16 conditions. The behavioral data for this dataset includes: hand position (x- and y-components), hand velocity (x- and y-components), forces and torques applied to the manipulandum (x- y- and z-forces; x- y- and z-torques), 7 estimated joint angles, 7 estimated joint velocities, 39 estimated muscle lengths, and 39 estimated muscle velocities. Position data was zeroed at movement onset for each trial to ensure position data was relative to a pre-movement baseline. More information on the task and data can be found in *Chowdhury et al., 2020*.

### DMFC_RSG

This dataset contains neural recordings (three Plexon probes) from dorsomedial frontal cortex while a monkey performed a time-interval reproduction task ('Ready-Set-Go'). In this task, the monkey received two visual cues, 'Ready' and 'Set', separated in time by a sample interval. The monkey was required to estimate the sample interval and report this estimate by performing an action ('Go') that followed the 'Set' cue by a production interval. The monkey was rewarded depending on how close the production interval was to the sample interval on a given trial. The sampling interval was selected on each trial from one of two prior distributions. The short prior distribution contained sampling intervals of 480, 560, 640, 720, and 800 ms. The long prior distribution contained sampling intervals of 800,

900, 1000, 1100, and 1200 ms. Trials were selected from prior distributions in blocks (i.e. many consecutive trials were drawn from the same prior distribution). Depending on the trial, the action could be reported in one of four ways: a joystick movement or an eye saccade to either the left or right. Between the two prior distributions, five sampling intervals per prior distribution, and four actions, there were a total of 40 conditions. More information on the task can be found in *Sohn et al., 2019*.

## MC_Cycle

This dataset contains neural recordings (two 96-electrode Utah arrays) from motor cortex (M1 and PMd) while a monkey performed a cycling task to navigate a virtual environment. Offline sorting of spike waveforms (using Kilosort: *Pachitariu et al., 2024*) yielded single-neuron and high-quality multi-neuron isolations. The dataset also includes threshold crossings that were detected online. On each trial, the monkey grasped a hand-pedal and moved it cyclically forward or backward to navigate the virtual environment. The color of the virtual landscape cued the monkey as to whether forward or backward cycling would advance their location in the virtual environment ('green' landscape: forward cycling advanced their location; 'tan' landscape: backward cycling advanced their location). Cycling distance was cued by a virtual target that appeared a fixed distance away in the virtual environment. The number of complete cycles of pedaling required to reach the target was either 1, 2, 4, or 7 cycles. Between the two cycling directions and four cycling distances, there were a total of 8 conditions. The behavioral data for this dataset includes: pedal phase, angular velocity of the pedal, x- and y- pedal position, x- and y- pedal velocity, and 7 intramuscular EMG recordings. The dataset includes both sorted spikes and threshold crossings. More information on the task can be found in *Schroeder et al., 2022*.

## MC_Maze

This dataset contains neural recordings (two 96-electrode Utah arrays) from motor cortex (M1 and PMd) while a monkey performed straight and curved reaches. The monkey was presented with a maze on a vertical screen with a cursor that tracked the motion of the monkey's hand. Targets appeared on the screen and the monkey had to make reaches that would move the cursor onto the target. Virtual barriers were presented that the cursor could not pass through, requiring curved reaches that avoided the barriers. Each maze configuration had three variants. The first variant had a single target with no barriers (straight reach). The second variant had a single target with barriers (curved reach). The third variant had a target with barriers (curved reach) as well as two unreachable distractor targets elsewhere in the maze. There were 36 maze configurations for a total of 108 conditions. The behavioral

**Table 2.** Neuron, condition, and trial counts for each dataset.

For some datasets, there are additional trials that are excluded from these counts. These trials are excluded because they were only usable for Neural Latents Benchmark submissions due to hidden behavioral data and partially hidden spiking data (see "Neural Latents Benchmark" section).

| Dataset | Neurons | Conditions | Trials |
|---|---|---|---|
| Area2_Bump | 65 | 16 | 364 |
| DMFC_RSG | 54 | 40 | 1006 |
| MC_Cycle | 112 | 8 | 273 |
| MC_Maze | 182 | 108 | 2295 |
| MC_Maze-L | 162 | 27 | 500 |
| MC_Maze-M | 152 | 27 | 250 |
| MC_Maze-S | 142 | 27 | 100 |
| MC_RTT | 130 | N/A | 1080 |
| MC_PacMan | 128 | 8 | 362 |
| Maze Simulations | 182 | 108 | 2295 |
| Multitask Network | 1200 | 9 | 270 |

data for this dataset includes x- and y-components of hand position and velocity. Position data was zeroed at movement onset for each trial to ensure position data was relative to a pre-movement baseline. More information on the task can be found in *Churchland et al., 2010*.

### MC_Maze-(L,M,S)

These three datasets were collected from the same monkey as in MC_Maze, performing the same task, but on different days with different conditions. The same arrays were used to collect neural recordings (though different numbers of sorted units were identified in each session) and the same behavioral data was collected. These datasets differ from MC_Maze primarily in that they have fewer conditions (nine maze configurations with three variants, for a total of 27 conditions) and fewer trials.

### MC_RTT

This dataset contains neural recordings (96-electrode Utah array) from motor cortex while a monkey made reaches in a horizontal plane to targets in an 8 x 8 grid. The task had no explicit trial structure nor did it require that individual movements be repeated throughout the session. Rather, a target would appear in one of the 64 grid locations and the monkey would reach to that target. Then, a new target would appear at a different random grid location and the monkey would reach there. This process continued throughout the session, leading to a collection of reaches that varied in terms of reach direction, reach distance, and reach speed. Despite lacking meaningful trial structure, the session was nevertheless broken up into 600 ms 'trial' segments (hence the trial count listed in *Table 2*). The behavioral data for this dataset includes x- and y-components of finger position and velocity. Due to the structure of the training data, neural trajectories were learned in long stretches spanning multiple movements. Thus, the zeroing of position data at movement onset that was employed for other datasets would have led to discontinuities in the behavioral trajectories for this task. We could have decoded position with no zeroing – however, similar movements (with similar corresponding neural activity) could have very different starting and ending positions in this dataset. Thus, we decided to focus decoding analyses on velocity only. More information on the task can be found in *Makin et al., 2018*.

### MC_PacMan

This dataset contains neural recordings (128 electrodes recorded simultaneously from a passive pre-production version of the Neuropixels 1.0-NHP probe) from motor cortex while a monkey generated forward isometric force against an immovable handle. The monkey was presented with a PacMan icon on a screen. PacMan's vertical position corresponded to the force applied to the handle. A path of scrolling dots moved horizontally across the screen and the monkey was rewarded when PacMan intercepted the dots. The pattern of scrolling dots was used to cue various dynamic force profiles. The task contained eight conditions that all spanned approximately the same range of forces: four ramping force profiles (fast and slow, ascending and descending), three sinusoidal force profiles (0.25 Hz, 1 Hz, 3 Hz), and a chirp profile in which the frequency steadily increased throughout the trial from 0-3 Hz. More information on the task can be found in *Marshall et al., 2022*.

### Maze task simulations

Several synthetic datasets were generated based on the maze task. These simulations were based on the MC_Maze dataset and matched the neuron, condition, and trial counts from MC_Maze. In all cases, the actual behavior from MC_Maze was used — spiking activity was the only component that was simulated.

To simulate firing rates, we first z-scored hand position, velocity, and acceleration (x- and y-components). Then, we simulated rates for each neuron as a random weighted sum of these z-scored kinematics with a constant offset. The weights and offsets were scaled such that the means and standard deviations of firing rates for the simulated neurons matched the means and standard deviations of the empirical trial-averaged rates in MC_Maze. In some simulations, we increased simulated firing rates by a factor of 5 or 10.

Simulated spiking activity was generated from simulated rates in one of two ways. In one simulation, spiking activity was simulated as a Poisson process, which corresponds to exponential-interval

spiking. In other simulations, spiking activity was simulated with gamma-interval spiking. More specifically, we let the interval between spikes be drawn from a gamma distribution with a shape parameter $\alpha = 2$ and a rate parameter $\beta = 2\lambda$, where $\lambda$ is the simulated firing rate. For reference, exponential-interval spiking corresponds to a gamma distribution with $\alpha = 1$ and $\beta = \lambda$. The overall rate of spiking was matched regardless of whether exponential-interval or gamma-interval spiking was simulated, but spiking occurred more regularly given the same rate for the gamma-interval simulations.

## Multitask spiking network

A simulated multitask dataset was generated using the method described in *DePasquale et al., 2023*. A spiking network was trained to generate posterior deltoid activity during reaching (eight reach conditions) and cycling (one cycling condition). In the network, there were 39 latent factors related to reaching and 12 latent factors related to cycling. The factors, and therefore the neural trajectories, for reaching and cycling were roughly orthogonal to one another. The network simulated 500 consecutive trials. A 135-trial stretch of this simulated dataset was designated as the test set (~7.1 min of data). The training set was constructed by randomly selecting 15 trials per condition (135 training trials total) from the remaining trials.

## Learning idealized trajectories

When training data includes repeated trials of the same behavior, neural and behavioral trajectories can be learned via standard trial-averaging. This procedure begins with filtering spikes to yield single-trial rates at millisecond resolution. Rates and behavioral variables are then aligned to behaviorally relevant events (e.g. movement onset). Single-trial rates are averaged across trials of the same condition to yield firing rates. The vector of firing rates at a given time defines a neural state. The collection of neural states within each condition defines a neural trajectory. Single-trial behavioral variables are similarly averaged across trials to yield behavioral states and trajectories.

**Table 3.** Hyperparameters for learning neural trajectories via standard trial-averaging.
$\sigma$ is the standard deviation of the Gaussian kernel used to temporally filter spikes. Trial-averaging Type I and Type II procedures are described in the "Averaging across trials" section. $D_{neural}$ and $D_{condition}$ are the neural- and condition-dimensionalities described in the "Smoothing across neurons and/or conditions" section. 'Full' means that no dimensionality reduction was performed. Condition smoothing could not be performed for MC_Cycle or the multitask network because different conditions in these datasets are of different lengths (i.e. $K_c$ is not the same for all $c$). (C) and (R) refer to the cycling and reaching trajectories, respectively, in the multitask network. $t_{move}$, $t_{stop}$, and $t_{go}$ correspond to movement onset, movement offset, and the 'go' time in the ready-set-go task, respectively. In MC_PacMan, each force profile is padded with static target forces at the beginning and end. $t_{prof\_start}$ and $t_{prof\_end}$ mark the beginning and end of the non-padded force profile on each trial.

| Dataset | Trajectory Start | Trajectory End | $\sigma$ | Trial Averaging | $D_{neural}$ | $D_{condition}$ |
|---|---|---|---|---|---|---|
| Area2_Bump | $t_{move} - 350$ | $t_{move} + 750$ | 25 | Type II | Full | Full |
| DMFC_RSG | $t_{go} - 1950$ | $t_{go} + 750$ | 55 | Type I | 49 | 17 |
| MC_Cycle | $t_{move} - 600$ | $t_{stop} + 600$ | 30 | Type I | Full | N/A |
| MC_Maze | $t_{move} - 500$ | $t_{move} + 700$ | 30 | Type II | Full | 21 |
| MC_Maze-L | $t_{move} - 500$ | $t_{move} + 700$ | 35 | Type II | Full | 16 |
| MC_Maze-M | $t_{move} - 500$ | $t_{move} + 700$ | 60 | Type II | Full | 20 |
| MC_Maze-S | $t_{move} - 500$ | $t_{move} + 700$ | 85 | Type II | 120 | 10 |
| MC_PacMan | $t_{prof\_start} - 1000$ | $t_{prof\_end} + 1000$ | 35 | Type II | Full | Full |
| Maze Simulations | $t_{move} - 500$ | $t_{move} + 700$ | 30 | Type II | Full | 21 |
| Multitask Network | (C) $t_{move} - 1000$<br>(R) $t_{move} - 800$ | (C) $t_{stop} + 1000$<br>(R) $t_{move} + 1000$ | 15 | Type I | Full | N/A |

While the above process is standard – it commonly forms the basis of scientific analyses – it may also require dataset-specific operations that reflect task structure and timing (e.g. one may need to align movements of slightly different durations). There also exist pre-processing and post-processing steps that can improve rate-estimates and have been used for this purpose in scientific analyses. These steps (which involve smoothing across conditions and neurons) can also improve decoding, and we thus employed them as appropriate. The sections below give complete dataset-specific details regarding how we computed firing rates. The broader point is that one should, for a given task-type, estimate rates in the way that makes the most sense given how the training data was collected, much as one would when analyzing the data scientifically.

## Filtering, extracting, and temporally aligning data on each trial

Spiking activity for each neuron on each trial was temporally filtered with a Gaussian to yield single-trial rates. *Table 3* reports the Gaussian standard deviations $\sigma$ (in milliseconds) used for each dataset. The optimal $\sigma$ for a dataset may depend on neuron count, typical spike rate, number of trials, and the timescale with which rates change in any given task or brain area. Given these factors vary across datasets, the optimal $\sigma$ is expected to vary as well. Thus, $\sigma$ was treated as a hyperparameter and optimized per-dataset.

Next, single-trial rates and behavioral data were extracted on each trial relative to key trial events. For example, in the maze datasets, data was extracted beginning 500 ms prior to movement onset and ending 700 ms after movement onset. These extraction boundaries (listed in *Table 3*) determine when the neural trajectories begin and end. These boundaries should be set to ensure that the behaviors one is interested in decoding are represented in the library of trajectories. It is also important to have extra data at the beginning of each trajectory because the first $\tau$ indices on each trajectory cannot be selected as candidate states (due to lack of sufficient state history).

In the maze datasets, Area2_Bump, and MC_PacMan, there was no need to warp time – all movements of the same condition unfolded over similar timescales relative to movement onset. However, in the cycling dataset this was not the case. The duration between movement onset and offset was variable across trials of the same condition (e.g. a mere 5% difference in cycling speed can lead to a ~180 ms discrepancy in the time it takes to complete 7 cycles). Thus, data in MC_Cycle was time-warped using the procedure described in *Russo et al., 2018*. In the DMFC_RSG dataset, the duration between the 'set' cue and 'go' action was variable across trials. Thus, uniform time-warping to match the average duration for each condition was used. More generally, one may find the time-warping technique in *Williams et al., 2020* to be a useful tool. It is important that warping take place *after* filtering spikes, because warping raw spiking activity would change the neurons' rates.

The result of filtering, extracting, and warping is that, for each condition $c$, single-trial rates can be formatted into a tensor $\mathbf{X}^c \in \mathbb{R}^{N \times K_c \times R}$ where $N$ is the number of neurons, $K_c$ is the number of extracted time points on each trial, and $R$ is the number of trials. Similarly, behavioral data can be formatted into tensors $\mathbf{Z}^c \in \mathbb{R}^{M \times K_c \times R}$ where $M$ is the number of behavioral variables.

## Averaging across trials

It is straightforward to average each $\mathbf{X}^c$ across trials yielding matrices $\overline{\mathbf{X}}^c \in \mathbb{R}^{N \times K_c}$. We refer to this as Type I averaging. However, we found that trial averages (and decoding) were often improved if we first used a preprocessing step that minimized the contribution of outliers. We refer to this as Type II averaging.

For Type II averaging, all single-trial rates ($\mathbf{X}^c$ for all $c$) are mean-centered and soft-normalized as described in *Russo et al., 2018*. The offset and scaling factor for this step are computed based on Type I averaged rates. These rates are formatted into matrices $\mathbf{X}_{in}^c \in \mathbb{R}^{R \times NK_c}$. PCA is used to reduce the trial-dimensionality of the rates to 1 for each $c$ via

$$\mathbf{X}_{out}^c = W_c W_c^\top \mathbf{X}_{in}^c \tag{26}$$

where $W_c \in \mathbb{R}^R$ is a column-vector corresponding to the first principal component of $\mathbf{X}_{in}^c$. Then, we reverse the soft-normalization and mean-centering operations on $\mathbf{X}_{out}^c$ and average across trials to get $\overline{\mathbf{X}}^c$ (Type II). Average rates were very similar regardless of how they were computed, but Type II rates have the potential advantage that the preprocessing step tends to minimize the contribution

of outliers, such as a single neuron having an unusually high rate on a single trial, at a time when all other neurons had typical rates. *Table 3* indicates for each dataset which trial-averaging procedure was used.

We average each $\mathbf{Z}^c$ across trials yielding matrices $\overline{\mathbf{Z}}^c \in \mathbb{R}^{M \times K_c}$. Typically, no modified trial-averaging procedure is required for behavioral data. However, there are cases where a behavioral variable should not be averaged without some pre- and/or post-processing. For example, in MC_Cycle, phase is a circular variable and therefore can't be linearly averaged. To accommodate this, we unwrapped phase, averaged across trials, then re-wrapped it.

## Smoothing across neurons and/or conditions

Scientific analyses sometimes improve estimates of firing rates by using dimensionality reduction to smooth across neurons or across conditions. Doing so reduces the impact of idiosyncratic events that are specific to one neuron or one condition. Smoothing across conditions can be particularly useful because it mitigates the typical tradeoff between collecting more conditions versus collecting more trials per condition.

To smooth across neurons, trial-averaged rates are mean-centered and soft-normalized, then concatenated across conditions into a matrix $\overline{\mathbf{X}}_{in} \in \mathbb{R}^{N \times K}$ where $K = \sum_c K_c$. PCA is used to reduce the neural-dimensionality of the rates via

$$\overline{\mathbf{X}}_{out} = WW^\top \overline{\mathbf{X}}_{in} \tag{27}$$

where $W \in \mathbb{R}^{N \times D_{neural}}$ is the top $D_{neural}$ principal components of $\overline{\mathbf{X}}_{in}$.

To smooth across conditions, $\overline{\mathbf{X}}_{out}$ is reformatted into a new $\overline{\mathbf{X}}_{in} \in \mathbb{R}^{C \times NK_c}$ where $K_c$ is the same for all $c$ (this form of smoothing cannot be applied when this is not the case). PCA is used to reduce the condition- dimensionality of the rates via

$$\overline{\mathbf{X}}_{out} = WW^\top \overline{\mathbf{X}}_{in} \tag{28}$$

where $W \in \mathbb{R}^{C \times D_{condition}}$ is the top $D_{condition}$ principal components of $\overline{\mathbf{X}}_{in}$.

Following the above, we reverse the soft-normalization and mean-centering operations on $\overline{\mathbf{X}}_{out}$, rectify to ensure non-negative rates, and reformat the smoothed rates back into matrices $\tilde{\overline{\mathbf{X}}}^c \in \mathbb{R}^{N \times K_c}$. Not all datasets benefited from neuron smoothing and/or condition smoothing. The smoothing dimensionalities used for each dataset are reported in *Table 3*.

After completing the previous steps (some combination of smoothing rates over time, trials, neurons, and/or conditions), there will be matrices of firing rates $\tilde{\overline{\mathbf{X}}}^c$ and behavioral data $\overline{\mathbf{Z}}^c$ for each $c$. The neural and behavioral states comprising the idealized neural and behavioral trajectories are then directly defined by these matrices.

$$\overline{\mathbf{x}}_k^c = \tilde{\overline{\mathbf{X}}}_{:,k}^c \tag{29}$$

$$\overline{\mathbf{z}}_k^c = \overline{\mathbf{Z}}_{:,k}^c \tag{30}$$

## Learning trajectories for MC_RTT

Neural trajectories were learned for MC_RTT using AutoLFADS (*Keshtkaran et al., 2022*). Following the procedure described in *Keshtkaran et al., 2022*, spiking activity in the training data was formatted into 600 ms segments with spike counts binned every 5 ms. Each segment overlapped with the previous segment by 200 ms and with the subsequent segment by 200 ms. AutoLFADS was then run twice on this formatted data (using 80/20 training and validation splits), yielding two sets of rate estimates that we then averaged across. Each AutoLFADS run yields slightly different results. Thus, we reasoned that averaging across runs would improve performance by reducing the impact of idiosyncratic rate variability that doesn't show up consistently across runs. We chose not to average more than two runs due to the computational burden this would accrue for training.

After averaging across runs, the rates from each segment were stitched back together (via the weighted average described in *Keshtkaran et al., 2022*) into long neural trajectories spanning many movements. The trajectories were then upsampled to 1 kHz via linear interpolation. If there had been no recording gaps, this procedure would have yielded a single neural trajectory. For the decoding

analyses, there were 2 recording gaps and therefore 3 neural trajectories. For the neural state estimation analysis (which utilized a larger training set), there were 3 recording gaps and therefore 4 neural trajectories. In all cases, each trajectory contained ~2.7 minutes of data. Although one might prefer trajectory boundaries to begin and end at behaviorally relevant moments (e.g. a stationary state), rather than at recording gaps, the exact boundary points are unlikely to be consequential for trajectories of this length that span multiple movements. If MINT estimates a state near the end of a long trajectory, its estimate will simply jump to another likely state on a different trajectory (or earlier along the same trajectory) in subsequent moments. Clipping the end of each trajectory to an earlier behaviorally-relevant moment would only remove potentially useful states from the libraries.

When running AutoLFADS for the neural state estimation analysis, one of the runs returned a highly oscillatory set of rates (~25-40 Hz oscillations) that did not by eye resemble the output of the other AutoLFADS runs. Although we confirmed this set of rates performed similarly for neural state estimation (0.200 bits/spike using the oscillatory rates alone, compared to 0.201 bits/spike when averaging across two non-oscillatory runs), we nevertheless chose to discard this run and average across two non-oscillatory AutoLFADS runs because the oscillatory solution was not one that AutoLFADS consistently returns for this particular training set.

See the "Other forms of interpolation" section for details on how these long neural trajectories are used during interpolation to improve neural state estimation and decoding.

## Learning trajectories for DMFC_RSG

Learning neural trajectories for the DMFC_RSG dataset involved two challenges related to the specifics of this dataset and the nature of the comparisons being made amongst methods. First, we needed to accommodate that each trial contained multiple trial events requiring separate alignment. Second, we desired a procedure for creating multiple libraries of neural trajectories corresponding to different portions of the session in which different recording instabilities were present. The solutions we developed for each of these challenges are described below in full detail for completeness. The solution for addressing recording instabilities described in this section was sufficient for facilitating fair comparison between MINT and other methods in the Neural Latents Benchmark that can implicitly model recording instabilities. However, the approach used here is rather specific to this desired comparison, and to the DMFC_RSG dataset. If MINT were deployed for online decoding, a different method to correct for recording instabilities would likely be desired (*Figure 8—figure supplement 2*).

Each trial contained five main trial events: the 'fixation' time when the monkey began looking at the center of the screen, the 'target onset' time when a white target appeared in one of two locations on the screen, the 'ready' cue, the 'set' cue, and the 'go' time at which the monkey initiated a joystick movement or saccade (depending on the condition). Thus, when learning neural trajectories, we needed to compute rates for five trial epochs: fixation to target onset, target onset to ready cue, ready cue to set cue, set cue to go time, and go time to the end of the trial. The first three epochs could not be averaged across trials in the standard way (because the durations varied from one trial to the next), but it was also not appropriate to warp them. Warping should be performed when the monkey is executing a computation or behavior at a slightly variable speed. In this case, the monkey was simply waiting for the next trial event without knowledge of when it would arrive. Thus, we chose to align to the initial event in each epoch, average across trials (ignoring missing data associated with the variable epoch durations), and then trim the averaged rates to the median epoch duration. In this task, certain conditions are not disambiguated for the monkey until the set cue arrives (e.g. the monkey doesn't know if they are in an 800 ms or 900 ms interval trial until the set cue arrives). Thus, averages for a given condition in these first three epochs included all trials from the given condition plus all trials from other conditions that could not be distinguished from the given condition at this stage of the trial. In the set cue to go time epoch, rates were uniformly warped to match the average duration within each condition and then averaged across trials. All data after the go time were aligned to the go time and averaged. After computing these epoch-specific rate averages, the rates were concatenated across epochs to yield neural trajectories for each condition. They were then trimmed according to the trajectory start and end times listed in *Table 3*.

With access to a very large dataset, computing multiple libraries of neural trajectories corresponding to different session epochs (e.g. beginning of session vs. end of session) is straightforward: simply break the training data up in time into different sections and separately compute a library

of trajectories for each. These libraries can then reflect how rates vary throughout a session due to recording instabilities. However, given limited training data, we sought to implement a similar strategy while not limiting the trajectories in each section to be based only on the small amount of spiking data available in that section. Thus, we first learned a library of neural trajectories based on the entire session (following the procedure described above) and smoothed across neurons and conditions to improve this estimate (yielding $\tilde{\tilde{\mathbf{X}}}^c$ for each condition $c$). Then, we broke up the session into 6 consecutive sections and proceeded to learn a complete set of neural trajectories for each section with no smoothing across neurons or conditions (yielding $\overline{\mathbf{Y}}^{c,d}$ for each condition $c$ and section $d$). Finally, we learned a linear transformation of $\tilde{\tilde{\mathbf{X}}}^c$ for each section $d$ that would better match the transformed firing rates for each section to those in $\overline{\mathbf{Y}}^{c,d}$ while preserving some of the neural geometry learned at the level of the session. This occurred via the equation

$$\tilde{\tilde{\mathbf{Y}}}^{c,d} = (W^{(d)}\Lambda + I)\tilde{\tilde{\mathbf{X}}}^c + b^{(d)} \tag{31}$$

where $W^{(d)} \in \mathbb{R}^{N \times N}$, $b^{(d)} \in \mathbb{R}^N$, and $\Lambda \in \mathbb{R}^{N \times N}$. Here, $\Lambda$ is simply a diagonal matrix whose action on $\tilde{\tilde{\mathbf{X}}}^c$ is to soft-normalize. The values in $\Lambda$ are learned from $\tilde{\tilde{\mathbf{X}}}^c$ via the same soft-normalization procedure described in *Russo et al., 2018* and used previously in this paper. The linear transformation was formulated in *Equation 31* to ensure that all regularization when learning the weights only applied to the deviation between the session-wide rates and the section-specific transformed rates. $W^{(d)}$ and $b^{(d)}$ were learned via an L2-regularized weighted least squares regression,

$$W^{(d)} = Q^{(d)}SP^\top(PSP^\top + \lambda I)^{-1} \tag{32}$$

$$b^{(d)} = \boldsymbol{\mu}_1^{(d)} - W^{(d)}\boldsymbol{\mu}_2 \tag{33}$$

$$Q^{(d)} = \overline{\mathbf{Y}}^d - \tilde{\tilde{\mathbf{X}}} - \boldsymbol{\mu}_1^{(d)} \tag{34}$$

$$P = \Lambda\tilde{\tilde{\mathbf{X}}} - \boldsymbol{\mu}_2 \tag{35}$$

$$\overline{\mathbf{Y}}^d = \begin{bmatrix} \overline{\mathbf{Y}}^{1,d} & \overline{\mathbf{Y}}^{2,d} & \dots & \overline{\mathbf{Y}}^{C,d} \end{bmatrix} \tag{36}$$

$$\tilde{\tilde{\mathbf{X}}} = \begin{bmatrix} \tilde{\tilde{\mathbf{X}}}^1 & \tilde{\tilde{\mathbf{X}}}^2 & \dots & \tilde{\tilde{\mathbf{X}}}^C \end{bmatrix} \tag{37}$$

where $S$ is a diagonal matrix of observation weights, $\boldsymbol{\mu}_1^{(d)}$ is the mean of $\overline{\mathbf{Y}}^d - \tilde{\tilde{\mathbf{X}}}$ across columns, and $\boldsymbol{\mu}_2$ is the mean of $\Lambda\tilde{\tilde{\mathbf{X}}}$ across columns. In *Equations 34 and 35*, the subtractions of $\boldsymbol{\mu}_1^{(d)}$ and $\boldsymbol{\mu}_2$ are applied to each column of the matrices they subtract from. The diagonal entries of $S$ weight how much each observation should matter when learning $W^{(d)}$. Given that the epoch between the set cue and the go time is when the monkey is performing an internal computation (keeping track of elapsed time internally) and this epoch is a large fraction of the evaluated trial period in the Neural Latents Benchmark, we decided $W^{(b)}$ should prioritize fitting this epoch well. Thus, diagonal entries of $S$ corresponding to samples in the set-go epoch were given a weight $s_{set-go}$ whereas all other diagonal entries of $S$ were set to 1. Both the set-go epoch weight and the L2 regularization term were treated as hyperparameters and optimized on a validation set, yielding $s_{set-go} = 4$ and $\lambda = 100$. To summarize, $W^{(d)}$ transforms the mean-centered, soft-normalized session-wide rates into a set of rate residuals that can be added to the session-wide rates to better match the firing rate statistics in each section of the data. The result of this procedure is a complete library of neural trajectories for each section of the training data $d$.

$$\overline{\mathbf{x}}_k^{c,d} = \tilde{\tilde{\mathbf{Y}}}_{:,k}^{c,d} \tag{38}$$

See the "Other forms of interpolation" section for details on how these multiple libraries are used during interpolation to improve the neural state estimate. There are no behavioral trajectories for this dataset (it was only used for neural state estimation analyses).

## Visualizing neural trajectories

Neural trajectories have dimensionality matching the number of recorded neurons (or multi-units), but it is often useful to visualize them in a 2D state space. Throughout this paper, PCA is used to project neural trajectories into a low-dimensional subspace. When this is done, firing rates are preprocessed

with mean-centering and soft-normalization (as in *Russo et al., 2018*). In some cases, it is useful to rotate the data within the top PCs to find a perspective that highlights certain properties of the trajectories. When this is done, we report the neural variance captured by the plotted dimensions (along with the neural variance captured by the top PCs) to contextualize the scale of the neural dimensions. Percent neural variance captured was computed in the standard way, as described in *Schroeder et al., 2022*.

## Distance analyses

In *Figure 2c-e*, several distance metrics were used to analyze the MC_Cycle dataset. Neural distance is defined as the Euclidean distance between two neural states in the full-dimensional space. Muscle distance is defined by z-scoring the 'muscle state' (the portion of the behavioral state corresponding to muscle activity), then computing the Euclidean distance between two normalized muscle states. The muscle state consisted of seven intramuscular EMG recordings (long head of the biceps, anterior deltoid, lateral deltoid, posterior deltoid, trapezius, and lateral and long heads of the triceps). Kinematic distance is defined using the phases and angular velocities associated with two behavioral states. Both phase and angular velocity are z-scored (for phase, this utilizes the circular standard deviation). Then, the phase distance ($d_{ph}$) is computed as the circular distance between the two normalized phases. The angular velocity distance ($d_{av}$) is computed as the absolute difference between the two normalized angular velocities. Then, the kinematic distance is computed as $d_{kin} = \sqrt{d_{ph}^2 + d_{av}^2}$. All trajectories (neural and behavioral) were binned at 10 ms resolution prior to computing pairwise distances to keep the analysis computationally manageable. In *Figure 2e*, data was partitioned into two trial sets ('Partition A' and 'Partition B') for a control analysis. To keep the number of trials used to compute trajectories matched across all analyses in *Figure 2c-e*, only trials from 'Partition A' were utilized in the neural distance vs. muscle distance and neural distance vs. kinematic distance analyses. When plotting pairwise distances, all distances were normalized (separately for each plotting axis) such that 1 corresponded to the average pairwise distance.

**Table 4.** Details for decoding analyses.
The number of training and testing trials for each dataset are provided along with the evaluation period over which performance was computed. Generalization analyses used subsets of these training and testing trials. The window lengths refer to the amount of spiking history MINT used for decoding (e.g. when $\tau' = 14$ and $\Delta = 20$ the window length is $(\tau' + 1)\Delta = 300$ ms). $t_{move}$ corresponds to movement onset, $t_{stop}$ corresponds to movement offset, $t_{start}$ refers to the beginning of a trial, and $t_{end}$ refers to the end of a trial. There is no defined condition structure for MC_RTT to use for defining trial boundaries. Thus, each trial is simply a 600 ms segment of data, with no alignment to movement. Although 270 of these segments were available for testing, the first 2 segments lacked sufficient spiking history for all decoders to be evaluated and were therefore excluded, leaving 268 test trials. In MC_PacMan, each force profile is padded with static target forces at the beginning and end. $t_{prof\_start}$ and $t_{prof\_end}$ mark the beginning and end of the non-padded force profile on each trial. For the multitask network, performance was evaluated on a continuous stretch of 135 trials spanning ~7.1 minutes.

| Dataset | Training Trials | Test Trials | Evaluation Start | Evaluation End | Window Length (ms) |
|---|---|---|---|---|---|
| Area2_Bump | 272 | 92 | $t_{move} - 100$ | $t_{move} + 500$ | 240 |
| MC_Cycle | 174 | 99 | $t_{move} - 250$ | $t_{stop} + 250$ | 200 |
| MC_Maze | 1721 | 574 | $t_{move} - 250$ | $t_{move} + 450$ | 300 |
| MC_Maze-L | 375 | 125 | $t_{move} - 250$ | $t_{move} + 450$ | 300 |
| MC_Maze-M | 188 | 62 | $t_{move} - 250$ | $t_{move} + 450$ | 300 |
| MC_Maze-S | 75 | 25 | $t_{move} - 250$ | $t_{move} + 450$ | 340 |
| MC_RTT | 810 | 268 | $t_{start}$ | $t_{start} + 599$ | 480 |
| MC_PacMan | 234 | 128 | $t_{prof\_start} - 500$ | $t_{prof\_end} + 1000$ | 300 |
| Maze Simulations | 1721 | 574 | $t_{move} - 250$ | $t_{move} + 450$ | 300 |
| Multitask Network | 135 | 135 | $t_{start}$ (first trial) | $t_{end}$ (last trial) | 300 |

The neural distances reported in *Figure 4d* are on the same scale as those in *Figure 2c-e*, i.e. they are Euclidean distances between neural states normalized by the average pairwise distance between neural states in the library of trajectories.

## Decoding analyses

All decoding analyses utilized the train-test trial splits listed in *Table 4*. Neural and behavioral trajectories were learned from the training set as described in the "Learning idealized trajectories" section. Then, MINT was provided with spiking activity on test trials with which to decode behavior. The trial epochs over which performance was evaluated were set to match the evaluation epochs from the Neural Latents Benchmark (for datasets that were used in the benchmark) and are listed in *Table 4*.

MINT used a bin size of $\Delta = 20$ ms. The amount of spiking history provided to MINT at each decoding time step varied by dataset and is listed in *Table 4*. Decoding was always causal (i.e. only utilizing spiking history prior to the decoded moment), with one exception described below in the "MINT variants" section. MINT utilized interpolation for all datasets. The lookup table of log-likelihoods utilized a minimum rate, $\lambda_{min}$, corresponding to 1 spike/second. Log-likelihoods in the table were also clipped so as not to drop below $\ln(10^{-6})$. These two choices regularize decoding such that a spurious spike from a neuron whose rate is close to 0 cannot dominate the decision of which neural state is most probable.

### Decoding R$^2$

Decoded behavioral variables were compared to ground truth behavioral variables over the evaluation epoch of all test trials. Performance was reported as the coefficient of determination ($R^2$) for the decoded behavioral variables, averaged across behavioral variables within a behavioral group. For example, $R^2$ for x-velocity and y-velocity were averaged to yield 'velocity $R^2$'. When computing $R^2$ for phase, circular distances and circular means were substituted for traditional distances and means in the $R^2$ definition. For the neural networks, the training/testing procedure was repeated 10 times with different random seeds. For most behavioral variables, there was very little variability in performance across repetitions. However, there were a few outliers for which variability was larger. Reported performance for each behavioral group is the average performance across the 10 repetitions to ensure results were not sensitive to any specific random initialization of each network. Decoding $R^2$ was always computed at 5 ms resolution (set to match the resolution used by the Neural Latents Benchmark).

### Decoder comparison

MINT was primarily compared to four other decode algorithms: Kalman filter, Wiener filter, feedforward neural network, and a recurrent neural network (GRU). An additional comparison to a Naive Bayes regression decoder was made for the MC_Maze dataset. All decoders were provided with the same training/testing data and used the same evaluation epochs. Complete details on their implementations are provided in the Appendix.

### Neuron dropping

The neuron dropping analyses in *Figure 8b* were performed on the MC_Maze-L dataset, which has 162 neurons. The analysis of decoding performance for known neuron loss consisted of training and testing the decoders with reduced sets of neurons. The analysis of performance for undetected neuron loss consisted of training and testing the decoder with all 162 neurons, but in the test set a subset of those neurons were artificially set to never spike. The analyses were performed for the following neuron counts: [1, 2, 3, 4, 5, 10, 15, ..., 150, 155, 160, 162]. At each neuron count, the known loss and undetected loss analyses were each repeated 50 times (with different neurons randomly dropped at each repetition). Linear interpolation between these neuron counts was then applied to generate $R^2$ values for every neuron count between 1 and 162.

### MINT variants

The analysis in *Figure 5—figure supplement 2a-c* required training and testing several variations on MINT for Area2_Bump, MC_Cycle, MC_Maze, and MC_RTT. These variations are described here.

The first variation determines how behavior is decoded from the neural state estimate. The 'Direct MINT Readout' utilizes the direct association between neural and behavioral states that is standard for MINT. The 'Linear MINT Readout' begins by estimating the neural state using MINT, but behavior is then decoded as a weighted sum of the rates comprising the neural state estimate. The weights are learned from training data using ridge regression, with the L2 regularization term in the regression optimized via a grid search with 5-fold cross validation. To improve the quality of the fit, a lag between neural and behavioral states was included (lags set to match the values used in *Pei et al., 2021*; MC_Cycle set to 100 ms).

The second variation considered whether interpolation was used. When interpolation was used, we employed the same methodology used for the analyses in *Figure 5* (6 candidate states for MC_RTT; 2 candidate states for other datasets). When interpolation was not used, the neural state was selected from the library of neural trajectories as the state that maximized the log-likelihood of the observed spikes.

The third variation determined whether decoding occurred causally or acausally. When decoding causally, the spiking observations all came prior to the decoded moment. When decoding acausally, the spiking observations were centered on the decoding moment. For the analysis in *Figure 5— figure supplement 2a-c*, the extent of spiking observations (window length) was optimized separately for causal vs. acausal decoding to give each variant the opportunity to perform as well as possible. The causal window lengths are reported in *Table 4*. The acausal window lengths were 560 ms for Area2_Bump, 400 ms for MC_Cycle, 580 ms for MC_Maze, and 660 ms for MC_RTT.

## SNR criteria for threshold crossings

When decoding based on threshold crossings in *Figure 5—figure supplement 2d*, a signal-to-noise ratio (SNR) was computed for each electrode channel. Then, only electrode channels with SNR > 2 were utilized for decoding. The SNR was computed as the ratio of the range of firing rates (determined from trial-averaged rates) and the standard deviation of the firing rate residuals. The firing rate residuals on each trial were computed as the difference between the filtered spikes and the corresponding trial-averaged rates. If the standard deviation was less than 5 spikes/second, it was set to 5 spikes/ second in the SNR computation. This ensured that channels with low variability in the residuals due to saturation at small or large firing rates were rejected. For example, an electrode channel that almost never spikes will have very low variability in the residuals simply because the firing rate is typically 0.

## Training non-MINT decoders with trial-averaged data

In the "Comparison to other decoders" section, it is reported that non-MINT decoders were trained with access to trial-averaged data and this did not improve decoding performance. This result comes from an analysis of the MC_Maze dataset. The training set was augmented to include two copies of each trial: the original trial, and an augmented trial in which spiking activity was replaced with trial-averaged rates for that trial's condition (appropriately normalized such that binned rates matched the scale of binned spike counts).

Thus, during training, each method was exposed both to the original trials (whose neural data match the statistics of neural data expected during testing) and the augmented trials that, in theory, might benefit decoding by exposing the decoder to denoised rates (i.e. providing each method with access to the same trial-averaged data that MINT leverages). In practice, decoding performance declined for each of the non-MINT decoders when trained this way, presumably due to the mismatch it created between the statistics of inputs during training and testing.

## Neural Latents Benchmark

All neural state estimation results in *Figure 7* came from the Neural Latents Benchmark (https:// neurallatents.github.io/), a neural latent variable modeling competition released through the 2021 NeurIPS Datasets & Benchmarks Track (*Pei et al., 2021*). For each of seven datasets, the benchmark provided training data containing simultaneous spiking activity and behavioral measurements. The neurons in the training data were divided into two sets: held-in neurons and held-out neurons. For the test data, the spiking activity of the held-in neurons was provided, but the spiking activity of the held-out neurons and the behavioral data was kept private by the benchmark curators. These splits are provided in *Table 5*. The evaluation epochs used are the same as those used in *Table 4*.

**Table 5.** Details for neural state estimation results.

Note that the training trial counts match the total number of trials reported in *Table 2*. This reflects that the Neural Latents Benchmark utilized an additional set of test trials not reflected in the *Table 2* trial counts. The test trials used for this analysis have ground truth behavior hidden by the benchmark creators and are therefore only suitable for this analysis.

| Dataset | Training Trials | Test Trials | Held-in Neurons | Held-out Neurons | Window Length (ms) |
|---|---|---|---|---|---|
| Area2_Bump | 364 | 98 | 49 | 16 | 500 |
| DMFC_RSG | 1006 | 283 | 40 | 14 | 1500 |
| MC_Maze | 2295 | 574 | 137 | 45 | 500 |
| MC_Maze-L | 500 | 100 | 122 | 40 | 500 |
| MC_Maze-M | 250 | 100 | 114 | 38 | 500 |
| MC_Maze-S | 100 | 100 | 107 | 35 | 500 |
| MC_RTT | 1080 | 271 | 98 | 32 | 500 |

(For DMFC_RSG, the evaluation epoch was the 1500 ms leading up to the 'go' time.) In the test data, the held-in spiking activity was only provided for time points within the evaluation epochs. For each dataset, submissions consisted of rate estimates for every trial in the training and test sets (held-in rates and held-out rates). The benchmark was fundamentally acausal: rate estimates at a given moment could leverage spiking activity from the entire trial. The data in *Figure 7* reflect all benchmark submissions up through Feb. 24th, 2023.

## MINT submissions

MINT first learned idealized neural trajectories during training as described in the "Learning idealized trajectories" section. Then, the neural trajectories were partitioned by neurons into two libraries: a held-in library of trajectories and a held-out library of trajectories. The held-in library contained neural states for the held-in neurons only. The held-out library similarly contained neural states for the held-out neurons only. MINT was run using the held-in library of trajectories to estimate a neural state in the held-in neural state space. A direct association between held-in neural states and held-out neural states (the same direct association typically used to map neural states to behavioral states) was then used to generate rate estimates for the held-out neurons. Rate estimates were saturated such that they could not be lower than 0.1 spikes/second. Interpolation was utilized across indices and conditions (with modifications for DMFC_RSG and MC_RTT as described in the "Other forms of interpolation" section).

The window length used for acausal neural state estimation on each dataset is provided in *Table 5*. Note that test set data was not provided outside of the evaluation epochs (an idiosyncracy of the benchmark). Thus, despite estimating rates acausally, the spiking observations often could not be centered on the estimated time. For example, when estimating the neural state for Area2_Bump, the window length was 500 ms, but the held-in spiking activity only spanned a 600 ms period. Thus, only neural states in the 250-350 ms portion of this 600 ms period could be estimated based on 500 ms of centered spiking activity. The neural states outside of this 250-350 ms zone were estimated by either propagating the earliest estimate backward or the latest estimate forward. This propagation is possible because each neural state estimate either occurs on a trajectory with a unique past and future or is an interpolation between trajectories that each have a unique past and future (i.e. the interpolation parameters are frozen and the states being interpolated are propagated backward or forward along their trajectories).

In addition to requiring test set rate estimates, the benchmark required training set rate estimates (these were needed for the velocity $R^2$ metric, see "Evaluation metrics" section). Each moment in each trial of the training data corresponded to a particular condition $c$ and a time within the execution of that condition $k$ (with the exception of MC_RTT, which will be discussed in a moment). Thus, training rate estimates were simply constructed by assigning each moment within each training trial a vector of rates $\bar{x}_k^c$ for the corresponding $c$ and $k$. For MC_RTT, the training rate estimates were simply the single-trial rates learned via AutoLFADS.

## Evaluation metrics

The evaluation metrics used in this paper are briefly described below. Detailed explanations can be found in *Pei et al., 2021*.

Bits per spike was used to assess how well the rate estimates for the held-out neurons on the test data matched the actual held-out spiking activity. The metric is computed by first computing the log-likelihood of the observed spikes, given rate estimates, across all neurons. Then, the log-likelihood that would have been obtained by letting the rate estimates be the neurons' mean rates is subtracted off. Finally, the metric is normalized. Positive values indicate that the held-out rate estimates are more predictive of held-out spiking activity than the mean rates.

PSTH $R^2$ was computed by collecting all rate estimates on the test set, sorting them by condition, and averaging rate estimates across trials of the same condition. Then the $R^2$ was computed between these rate-estimate-derived PSTHs and the empirical PSTHs (computed by smoothing spikes and averaging across trials within conditions). Empirical PSTHs were computed using trials from both the training and test sets. A separate $R^2$ value was generated for each neuron — the reported values are the average $R^2$ across all neurons. This metric could not be computed for MC_RTT due to lack of condition structure.

Velocity $R^2$ was computed by first regressing lagged x- and y-velocity of the hand against estimated rates in the training data (ridge regression). Then, the linear mapping learned via regression is applied to estimated rates in the test data to generate estimated x- and y-velocity of the hand. $R^2$ is computed between estimated velocities and actual velocities on the test data and averaged across the x- and y-components. This metric was not computed for DMFC_RSG because the evaluated portion of the trials did not involve motion of the monkey's arm.

## Hyperparameter optimization

MINT has very few hyperparameters, all of which can be readily set by hand. These hyperparameters typically relate straightforwardly to properties of the task or data, and performance is robust to their exact values (*Figure 5—figure supplement 1*). For example, we did not optimize bin size ($\Delta$). Rather, we let $\Delta = 20$ ms for all analyses, reasoning that this value would reduce computation time while remaining considerably shorter than the timescale over which rates change. MINT's only other relevant hyperparameters are window length (duration of spiking history considered at each moment) and the number of candidate states to use for interpolation. The number of candidate states was simply set to two for all datasets except MC_RTT. For MC_RTT, the nature of the training data argued for considering more candidate states during the interpolation stage. Thus, the number of states was optimized using a grid search with 10-fold cross validation on the training set (using velocity decoding $R^2$ as an objective function to maximize). Window length was also optimized via this grid search procedure, separately for each dataset. In some cases (e.g. the maze datasets), we still chose to standardize window length across datasets because optimization yielded similar values. Because neural state estimation was acausal for the Neural Latents Benchmark, whereas decoding analyses were typically causal (with one exception described in the "MINT variants" section), window length was also optimized separately across these analyses. For the simulated datasets, we chose not to optimize window length at all – the maze simulations and multitask spiking network were simply set to use the same window length as MC_Maze.

This approach to hyperparameter selection contrasts with our approach for the non-MINT decoders. As described in the Appendix, the non-MINT decoders utilized a more sophisticated technique to select hyperparameters: Bayesian optimization (*Snoek et al., 2012*). Additionally, the non-MINT decoders were allowed the flexibility of using a different window length for each behavioral group. The choice not to provide MINT with the same flexibility was not arbitrary. Rather, this choice emphasizes a particularly useful aspect of MINT: any relevant behavioral variable can be read out from the same neural state estimate. There is no need to use separate training or decoding procedures for different behavioral variables.

The methods we used for learning neural trajectories to use with MINT also contained hyperparameters (e.g. *Table 3*). For all datasets involved in the Neural Latents Benchmark (except MC_RTT), these hyperparameters were optimized using the grid search procedure (using bits per spike as an objective function to maximize). These hyperparameters were then re-used for decoding analyses with no additional optimization. For MC_RTT, we utilized AutoLFADS to learn neural trajectories, which contains

a built-in procedure for optimizing hyperparameters. MC_Cycle, MC_PacMan, the maze simulations, and the multitask network were not part of the Neural Latents Benchmark and therefore needed their trajectory-learning hyperparameters set in a different way. For MC_Cycle and MC_PacMan, these hyperparameters were optimized using the grid search procedure with velocity and force decoding $R^2$, respectively, as the objective functions. The maze simulations simply used the same hyperparameters as MC_Maze. For the multitask network, the trajectory learning hyperparameters were set very conservatively by hand to only perform very light temporal smoothing on spikes along with standard trial-averaging.

The example decoding results in *Figure 4* and *Figure 4—video 1* used the same hyperparameters that were used in generating the quantitative decoding results in *Figure 5*.

## Acknowledgements

We thank Felix Pei, Chethan Pandarinath, and the rest of the Neural Latents Benchmark team for curating many of the datasets used in this paper and releasing them publicly. We thank those who originally collected those datasets: Raeed Chowdhury (Area2_Bump), Hansem Sohn (DMFC_RSG), Matt Kaufman and Mark Churchland (MC_Maze, MC_Maze-L,M,S), and Joseph O'Doherty (MC_RTT). We additionally thank Felix and Chethan for providing AutoLFADS rates for the MC_RTT dataset. We thank Karen Schroeder for collecting and preprocessing the MC_Cycle dataset in collaboration with the first and last authors. We thank Najja Marshall, Eric Trautmann, and Andrew Zimnik for their central roles (with Elom Amematsro) in designing the PacMan task and collecting data using Neuropixels probes. We thank Yana Pavlova for expert animal care while collecting the MC_Cycle and MC_PacMan datasets. We thank Brian DePasquale for providing code for simulating the artificial multitask spiking network. We thank Andrew Zimnik and Eric Trautmann for helping the second author present this work at NCM 2022 when the first author had COVID. This work was supported by the Simons Foundation and the Grossman Center for the Statistics of Mind. Aspects of analysis (e.g. generalization performance during the PacMan task) and manuscript preparation were supported by NIH 1R01NS135240. Elom Amematsro was supported by an NSF graduate fellowship.

## Additional information

### Competing interests

Sean M Perkins, John Cunningham, Mark M Churchland: holds a patent pertaining to this work. The patent has been licensed to Blackrock Neurotech. The authors declare no additional competing interests. The other authors declare that no competing interests exist.

### Funding

| Funder | Grant reference number | Author |
|---|---|---|
| National Institutes of Health | 1R01NS135240 | Mark M Churchland |
| Simons Foundation | SCGB | John Cunningham<br>Mark M Churchland |
| Grossman Center for the Statistics of Mind | | Mark M Churchland |
| National Science Foundation | Graduate Fellowship | Elom A Amematsro |

The funders had no role in study design, data collection and interpretation, or the decision to submit the work for publication.

### Author contributions

Sean M Perkins, Conceptualization, Software, Formal analysis, Investigation, Methodology, Writing – original draft, Writing – review and editing; Elom A Amematsro, Data curation, Investigation; John Cunningham, Conceptualization, Methodology, Writing – original draft; Qi Wang, Supervision,

Investigation; Mark M Churchland, Conceptualization, Formal analysis, Supervision, Investigation, Methodology, Writing – original draft, Project administration, Writing – review and editing, Funding acquisition

**Author ORCIDs**
Sean M Perkins ⓘ https://orcid.org/0000-0001-9456-4648
Elom A Amematsro ⓘ https://orcid.org/0000-0003-4843-4513
Mark M Churchland ⓘ https://orcid.org/0000-0001-9123-6526

**Ethics**
For the MC_Cycle and MC_PacMan datasets, animal protocols were approved by the Columbia University Institutional Animal Care and Use Committee (Protocol number AC-AABE3550). For other datasets, protocols were approved by the relevant institutional committee for that institution.

Reviewer #1 (Public Review): https://doi.org/10.7554/eLife.89421.3.sa1
Reviewer #2 (Public Review): https://doi.org/10.7554/eLife.89421.3.sa2
Author response https://doi.org/10.7554/eLife.89421.3.sa3

---

# Additional files

**Supplementary files**
MDAR checklist

**Data availability**
MINT is available at GitHub (copy archived at *Perkins, 2025*) (MATLAB). Other decoders are available at GitHub (*Perkins, 2023*) (Python). Multitask spiking network was simulated with code from *DePasquale et al., 2023* that is available at GitHub (*DePasquale, 2022*) (MATLAB). Empirical datasets (except MC_Cycle and MC_PacMan) were curated by the Neural Latents Benchmark team and made publicly available on DANDI Archive. The MC_Cycle dataset is available at Zenodo. The MC_PacMan dataset is available at Zenodo. Useful functions for loading the DANDI datasets are available at GitHub (*Pei et al., 2024*).

The following datasets were generated:

| Author(s) | Year | Dataset title | Dataset URL | Database and Identifier |
|---|---|---|---|---|
| Perkins S, Schroeder K, Churchland M | 2025 | MC_Cycle: macaque motor cortex spiking activity during cycling | https://doi.org/10.5281/zenodo.14769761 | Zenodo, 10.5281/zenodo.14769761 |
| Amematsro E, Marshall N, Trautmann E, Churchland M | 2025 | MC_PacMan: macaque motor cortex spiking activity during isometric force production | https://doi.org/10.5281/zenodo.14769889 | Zenodo, 10.5281/zenodo.14769889 |

The following previously published datasets were used:

| Author(s) | Year | Dataset title | Dataset URL | Database and Identifier |
|---|---|---|---|---|
| Pei F, Chowdhury R, Miller L | 2022 | Area2_Bump: macaque somatosensory area 2 spiking activity during reaching with perturbations | https://doi.org/10.48324/dandi.000127/0.220113.0359 | DANDI Archive, 10.48324/dandi.000127/0.220113.0359 |
| Pei F, Sohn H, Jazayeri M | 2022 | DMFC_RSG: macaque dorsomedial frontal cortex spiking activity during time interval reproduction task | https://doi.org/10.48324/dandi.000130/0.220113.0407 | DANDI Archive, 10.48324/dandi.000130/0.220113.0407 |

*Continued on next page*

*Continued*

| Author(s) | Year | Dataset title | Dataset URL | Database and Identifier |
|---|---|---|---|---|
| Pei F, Churchland M, Kaufman M, Shenoy K | 2022 | MC_Maze: macaque primary motor and dorsal premotor cortex spiking activity during delayed reaching | https://doi.org/10.48324/dandi.000128/0.220113.0400 | DANDI Archive, 10.48324/dandi.000128/0.220113.0400 |
| Pei F, Churchland M, Kaufman M, Shenoy K | 2022 | MC_Maze_Large: macaque primary motor and dorsal premotor cortex spiking activity during delayed reaching | https://doi.org/10.48324/dandi.000138/0.220113.0407 | DANDI Archive, 10.48324/dandi.000138/0.220113.0407 |
| Pei F, Churchland M, Kaufman M, Shenoy K | 2022 | MC_Maze_Medium: macaque primary motor and dorsal premotor cortex spiking activity during delayed reaching | https://doi.org/10.48324/dandi.000139/0.220113.0408 | DANDI Archive, 10.48324/dandi.000139/0.220113.0408 |
| Pei F, Churchland M, Kaufman M, Shenoy K | 2022 | MC_Maze_Small: macaque primary motor and dorsal premotor cortex spiking activity during delayed reaching | https://doi.org/10.48324/dandi.000140/0.220113.0408 | DANDI Archive, 10.48324/dandi.000140/0.220113.0408 |
| Pei F, O'Doherty JE | 2024 | MC_RTT: macaque motor cortex spiking activity during self-paced reaching | https://doi.org/10.48324/dandi.000129/0.241017.1444 | DANDI Archive, 10.48324/dandi.000129/0.241017.1444 |

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

## Appendix 1

### Decoder implementations

All decoders described in this section were provided with the same training and testing data as MINT. When decoding, spiking activity was binned every 20 ms for each neuron (with the exception of the Naive Bayes regression decoder, which benefited from a larger bin size). Each decoder was given access to recent binned spike counts to use for decoding. The following sections will: (1) provide an overview of the notation and definitions needed to understand the subsequent equations (notation *will not* in general match the notation used for MINT), (2) describe the hyperparameter optimization technique used for all decoders in this appendix, and (3) describe each decoder's implementation in detail.

## Definitions

### Time Bins

Each decoder received binned spiking activity and decoded behavior at time-bin resolution. These time bins are indexed by $k$. When higher-resolution estimates were required (e.g. for evaluating decoding performance), the decoded variables were simply upsampled with a zero-order-hold.

### Observations

The vector of spike counts at time bin $k$ is denoted by the column vector $\mathbf{x}_k \in \mathbb{R}^N$ where $N$ is the number of neurons. When decoding, each decoder was given access to the current time bin of spike counts and $\kappa$ previous time bins of spike counts. $\kappa$ was not a fixed value and varied depending on the method, dataset, and behavioral group being decoded.

### Target Variables

The vector of behavioral variables at time bin $k$ is denoted by the column vector $\mathbf{y}_k \in \mathbb{R}^M$ where $M$ is the number of behavioral variables. This corresponds to the values of the behavioral variables at the end of the time bin.

### Decoded variables

The vector of decoded behavioral variables at time bin $k$ is denoted $\hat{\mathbf{y}}_k$.

### Training data

Observations and target variables at time bin $i$ of trial $j$ in the training data are denoted $\mathbf{x}_i^{(j)}$ and $\mathbf{y}_i^{(j)}$.

## Hyperparameter optimization

The hyperparameters for each decoder were set using Bayesian optimization (*Snoek et al., 2012*). The training set was split into a reduced training set (80% of the trials from the full training set) and a validation set (the remaining 20% of trials). The method is provided with a range of hyperparameter values to search. The method then seeks to learn a set of hyperparameters that, when trained on the reduced training set, will lead to maximal decoding $R^2$ on the validation set. The optimization occurs via an iterative process that involves exploring the space of hyperparameters and exploiting knowledge of how certain hyperparameter sets performed in previous iterations. In all cases, we set the method to initially perform 10 iterations of random exploration of hyperparameter space, followed by 10 iterations of Bayesian optimization. The optimization was performed separately for each behavioral group (e.g. position vs. velocity) within each dataset. In the decoder-specific sections below, the exact hyperparameter values learned for each dataset and behavioral group are reported along with the hyperparameter ranges that were searched.

## Wiener filter

The Wiener filter (*Ahmadi et al., 2021*) uses a model

$$\mathbf{y}_k = b + \sum_{i=0}^{\kappa} W_i \mathbf{x}_{k-i} + \boldsymbol{\epsilon}_k \tag{S1}$$

where $b \in \mathbb{R}^M$, $W_i \in \mathbb{R}^{M \times N}$, and the residuals are $\boldsymbol{\epsilon}_k \in \mathbb{R}^M$. Decoding occurs via the equation

$$\hat{\mathbf{y}}_k = W \begin{bmatrix} \mathbf{x}_k \\ \mathbf{x}_{k-1} \\ \vdots \\ \mathbf{x}_{k-\kappa} \\ 1 \end{bmatrix} \tag{S2}$$

for some matrix $W$ that minimizes the squared error of the residuals in the training data.

## Parameter fitting

For each trial $j$ in the training data, we construct matrices

$$\mathbf{X}^{(j)} = \begin{bmatrix} \mathbf{x}^{(j)}_{\kappa+1} & \mathbf{x}^{(j)}_{\kappa+2} & \cdots & \mathbf{x}^{(j)}_{T_j} \\ \mathbf{x}^{(j)}_{\kappa} & \mathbf{x}^{(j)}_{\kappa+1} & \cdots & \mathbf{x}^{(j)}_{T_j-1} \\ \vdots & \vdots & \ddots & \vdots \\ \mathbf{x}^{(j)}_1 & \mathbf{x}^{(j)}_2 & \cdots & \mathbf{x}^{(j)}_{T_j-\kappa} \\ 1 & 1 & \cdots & 1 \end{bmatrix} \tag{S3}$$

$$\mathbf{Y}^{(j)} = \begin{bmatrix} \mathbf{y}^{(j)}_{\kappa+1} & \mathbf{y}^{(j)}_{\kappa+2} & \cdots & \mathbf{y}^{(j)}_{T_j} \end{bmatrix} \tag{S4}$$

where $T_j$ is the number of time bins in trial $j$. Then, we concatenate across $J$ trials to get

$$\mathbf{X} = [\mathbf{X}^{(1)} \quad \mathbf{X}^{(2)} \quad \cdots \quad \mathbf{X}^{(J)}] \tag{S5}$$

$$\mathbf{Y} = [\mathbf{Y}^{(1)} \quad \mathbf{Y}^{(2)} \quad \cdots \quad \mathbf{Y}^{(J)}] \tag{S6}$$

can then be fit via a regularized regression

$$W = \mathbf{Y}\mathbf{X}^\top(\mathbf{X}\mathbf{X}^\top + \lambda I)^{-1} \tag{S7}$$

**Appendix 1—table 1.** Hyperparameters used for the Wiener filter.

The L2 regularization term $\lambda$ was optimized in the range [0, 2000], with the optimized values rounded to the closest multiple of 10. Window lengths were optimized (in 20 ms increments) in the range [200, 600] for Area2_Bump, [200, 1000] for MC_Cycle, [200, 1200] for MC_RTT, [200, 500] for MC_PacMan, and [200, 700] for MC_Maze, MC_Maze-L, MC_Maze-M, and MC_Maze-S. These ranges were determined by the structure of each dataset (e.g. Area2_Bump couldn't look back more than 600 ms from the beginning of the evaluation epoch without entering the previous trial). Window lengths are directly related to $\kappa$ via $\Delta$ (e.g. $\kappa = 14$ would correspond to a window length of $\Delta(\kappa + 1) = 20(14 + 1) = 300$ ms).

| Dataset | Behavioral Group | Window Length (ms) | L2 Regularization ($\lambda$) |
|---|---|---|---|
| | Position | 560 | 350 |
| | Velocity | 320 | 320 |
| | Force | 600 | 900 |
| | Joint Angles | 300 | 250 |
| | Joint Velocities | 400 | 1410 |
| | Muscle Lengths | 380 | 0 |
| Area2_Bump | Muscle Velocities | 460 | 940 |

*Appendix 1—table 1 Continued on next page*

*Appendix 1—table 1 Continued*

| Dataset | Behavioral Group | Window Length (ms) | L2 Regularization (λ) |
|---|---|---|---|
| | Phase | 1000 | 1730 |
| | Angular Velocity | 960 | 1470 |
| | Position | 920 | 1710 |
| | Velocity | 420 | 600 |
| MC_Cycle | EMG | 560 | 1990 |
| | Position | 660 | 530 |
| MC_Maze | Velocity | 540 | 210 |
| | Position | 540 | 510 |
| MC_Maze-L | Velocity | 540 | 440 |
| | Position | 660 | 440 |
| MC_Maze-M | Velocity | 480 | 310 |
| | Position | 680 | 130 |
| MC_Maze-S | Velocity | 420 | 270 |
| MC_RTT | Velocity | 1180 | 1610 |
| MC_PacMan | Force | 480 | 470 |

**Appendix 1—table 2.** Hyperparameters used for the Kalman filter.

The lag (in increments of 20 ms time bins) between neural activity and behavior was optimized in the range [2, 8], corresponding to 40–160 ms, for all datasets except Area2_Bump. For Area2_Bump the lag was not optimized and was simply set to 0 due to the fact that, in a sensory area, movement precedes sensory feedback. Given that $\mathbf{x}_k$ aggregates spikes across the whole time bin, but $\mathbf{y}_k$ corresponds to the behavioral variables at the end of the time bin, the effective lag is actually half a bin (10 ms) longer — i.e. the effective range of lags considered for the non-sensory datasets was 50–170 ms.

| Dataset | Behavioral Group | Lag (bins) |
|---|---|---|
| Area2_Bump | Position & Velocity | 0 |
| MC_Cycle | Position & Velocity | 2 |
| MC_Maze | Position & Velocity | 4 |
| MC_Maze-L | Position & Velocity | 4 |
| MC_Maze-M | Position & Velocity | 4 |
| MC_Maze-S | Position & Velocity | 4 |
| MC_RTT | Position & Velocity | 2 |

## Kalman filter

The Kalman filter (*Kalman, 1960*) uses a model

$$\mathbf{y}_k = A\mathbf{y}_{k-1} + \mathbf{q}_k \tag{S8}$$

$$\mathbf{x}_k = C\mathbf{y}_k + \mathbf{r}_k \tag{S9}$$

where $\mathbf{q}_k \in \mathcal{N}(0, Q)$ and $\mathbf{r}_k \in \mathcal{N}(0, R)$. Decoding occurs via the standard update equations:

$$\hat{\mathbf{y}}_k^- = A\hat{\mathbf{y}}_{k-1} \tag{S10}$$

$$\hat{\mathbf{y}}_k = \hat{\mathbf{y}}_k^- + K_k(\mathbf{x}_k - C\hat{\mathbf{y}}_k^-) \tag{S11}$$

where

$$P_k^- = AP_{k-1}A^\top + Q \tag{S12}$$

$$K_k = P_k^- C^\top (CP_k^- C^\top + R)^{-1} \tag{S13}$$

$$P_k = P_k^- - K_k C P_k^- \tag{S14}$$

The state vector $\mathbf{y}_k$ includes 7 elements: position, velocity, and acceleration (x- and y-components of each) as well as a constant 1. The 1 was included to allow firing rates to have a constant offset. Acceleration was included as in *Wu et al., 2003* to improve the position and velocity estimates.

## Parameter fitting

For each trial in the training set, we construct matrices

$$\mathbf{X}^{(j)} = \begin{bmatrix} \mathbf{x}_1^{(j)} & \mathbf{x}_2^{(j)} & \dots & \mathbf{x}_{T_{j-lag}}^{(j)} \end{bmatrix} \tag{S15}$$

$$\mathbf{Y}^{(j)} = \begin{bmatrix} \mathbf{y}_{1+lag}^{(j)} & \mathbf{y}_{2+lag}^{(j)} \cdots \mathbf{y}_{T_j}^{(j)} \end{bmatrix} \tag{S16}$$

$$\mathbf{Y}_1^{(j)} = \begin{bmatrix} \mathbf{y}_{1+lag}^{(j)} & \mathbf{y}_{2+lag}^{(j)} & \dots & \mathbf{y}_{T_j-1}^{(j)} \end{bmatrix} \tag{S17}$$

$$\mathbf{Y}_2^{(j)} = \begin{bmatrix} \mathbf{y}_{2+lag}^{(j)} & \mathbf{y}_{3+lag}^{(j)} & \dots & \mathbf{y}_{T_j}^{(j)} \end{bmatrix} \tag{S18}$$

where $T_j$ is the number of time bins in trial $j$. Then, we concatenate across $J$ trials to get

$$\mathbf{X} = \begin{bmatrix} \mathbf{X}^{(1)} & \mathbf{X}^{(2)} & \dots & \mathbf{X}^{(J)} \end{bmatrix} \tag{S19}$$

$$\mathbf{Y} = \begin{bmatrix} \mathbf{Y}^{(1)} & \mathbf{Y}^{(2)} & \dots & \mathbf{Y}^{(J)} \end{bmatrix} \tag{S20}$$

$$\mathbf{Y}_1 = \begin{bmatrix} \mathbf{Y}_1^{(1)} & \mathbf{Y}_1^{(2)} & \dots & \mathbf{Y}_1^{(J)} \end{bmatrix} \tag{S21}$$

$$\mathbf{Y}_2 = \begin{bmatrix} \mathbf{Y}_2^{(1)} & \mathbf{Y}_2^{(2)} & \dots & \mathbf{Y}_2^{(J)} \end{bmatrix} \tag{S22}$$

The parameters can then be fit as follows:

$$A = \mathbf{Y}_2 \mathbf{Y}_1^\top (\mathbf{Y}_1 \mathbf{Y}_1^\top)^{-1} \tag{S23}$$

$$C = \mathbf{X}\mathbf{Y}^\top (\mathbf{Y}\mathbf{Y}^\top)^{-1} \tag{S24}$$

$$Q = \frac{(\mathbf{Y}_2 - A\mathbf{Y}_1)(\mathbf{Y}_2 - A\mathbf{Y}_1)^\top}{T_1} \tag{S25}$$

$$R = \frac{(\mathbf{X} - C\mathbf{Y})(\mathbf{X} - C\mathbf{Y})^\top}{T} \tag{S26}$$

$$\hat{\mathbf{y}}_1^- = \frac{1}{J} \sum_{j=1}^{J} \mathbf{y}_{1+lag}^{(j)} \tag{S27}$$

$$P_1^- = \frac{1}{J} \sum_{j=1}^{J} \left( \mathbf{y}_{1+lag}^{(j)} - \hat{\mathbf{y}}_1^- \right)(\mathbf{y}_{1+lag}^{(j)} - \hat{\mathbf{y}}_1^-)^\top \tag{S28}$$

where $T_1$ and are the number of columns in $\mathbf{Y_1}$ and $\mathbf{Y}$, respectively

## Feedforward Neural Network

The feedforward neural network (**Rosenblatt, 1961**) we use takes spike count observations (current and previous time bins) and flattens them into a long vector,

$$\mathbf{f}_k = \begin{bmatrix} \mathbf{x}_k^\top & \mathbf{x}_{k-1}^\top & \dots & \mathbf{x}_{k-\kappa}^\top \end{bmatrix}^\top \tag{S29}$$

centers and normalizes that vector,

$$\tilde{\mathbf{f}}_k = (\mathbf{f}_k - \mu) \oslash \sigma \tag{S30}$$

and then feeds it through multiple hidden network layers

$$\mathbf{h}_k^1 = \mathrm{ReLU}(W_1 \tilde{\mathbf{f}}_k + b_1) \tag{S31}$$

$$\mathbf{h}_k^l = \mathrm{ReLU}(W_l \mathbf{h}_k^{l-1} + b_l),\ l \in \{2,\dots,L\} \tag{S32}$$

before finally reading out behavior

$$\hat{\mathbf{y}}_k = W_{out}\mathbf{h}_k^L + b_{out} \tag{S33}$$

where $\mathbf{h}_k^l \in \mathbb{R}^D$, $D$ is the number of units per hidden layer, and $L$ is the number of hidden layers.

## Parameter fitting

For each trial $j$ in the training set, we create flattened observations

$$\mathbf{f}_i^{(j)} = \begin{bmatrix} (\mathbf{x}_i^{(j)})^\top & (\mathbf{x}_{i-1}^{(j)})^\top & \dots & (\mathbf{x}_{i-\kappa}^{(j)})^\top \end{bmatrix}^\top \tag{S34}$$

for all $i > \kappa$ (sufficient spike count history doesn't exist for $i \leq \kappa$). We then let $\mu$ and $\sigma$ be the element-wise mean and standard deviation of $\mathbf{f}_i^{(j)}$ across all observations in the training set (i.e. across all time bins in all $J$ trials for which $i > \kappa$). The parameters $W_{out}$, $b_{out}$, $W_l$, and $b_l$ for $l \in \{1,\dots,L\}$ are then learned by training the network with the Adam optimization routine (**Kingma and Ba, 2014**) and a mean-squared error loss function. Training utilized dropout on the outputs of each hidden layer.

**Appendix 1—table 3.** Hyperparameters used for the feedforward neural network.
The number of hidden layers ($L$) was optimized in the range [1, 15]. The number of units per hidden layer ($D$) was optimized in the range [50, 1000], with the optimized values rounded to the closest multiple of 10. The dropout rate was optimized in the range [0,0.5] and the number of training epochs was optimized in the range [2, 100]. Window lengths were optimized (in 20 ms increments) in the range [200, 600] for Area2_Bump, [200, 1000] for MC_Cycle, [200, 1200] for MC_RTT, and [200, 700] for MC_Maze, MC_Maze-L, MC_Maze-M, and MC_Maze-S.

| Dataset | Behavioral Group | Window Length (ms) | Hidden Layers (*L*) | Units/Layer (*D*) | Dropout Rate | Epochs |
|---|---|---|---|---|---|---|
| | Position | 440 | 4 | 690 | 0.20 | 58 |
| | Velocity | 520 | 5 | 260 | 0 | 62 |
| | Force | 240 | 4 | 560 | 0.01 | 54 |
| | Joint Angles | 320 | 7 | 480 | 0.15 | 59 |
| | Joint Velocities | 460 | 5 | 660 | 0.12 | 36 |
| | Muscle Lengths | 240 | 10 | 840 | 0.19 | 81 |
| Area2_Bump | Muscle Velocities | 500 | 6 | 480 | 0.02 | 24 |

*Appendix 1—table 3 Continued on next page*

*Appendix 1—table 3 Continued*

| Dataset | Behavioral Group | Window Length (ms) | Hidden Layers (L) | Units/Layer (D) | Dropout Rate | Epochs |
|---|---|---|---|---|---|---|
| | Phase | 740 | 8 | 180 | 0.11 | 24 |
| | Angular Velocity | 700 | 8 | 710 | 0.01 | 43 |
| | Position | 360 | 9 | 470 | 0 | 45 |
| | Velocity | 460 | 5 | 330 | 0.25 | 55 |
| MC_Cycle | EMG | 840 | 5 | 970 | 0.07 | 43 |
| | Position | 680 | 7 | 340 | 0.15 | 65 |
| MC_Maze | Velocity | 700 | 14 | 760 | 0 | 89 |
| | Position | 380 | 6 | 160 | 0.05 | 44 |
| MC_Maze-L | Velocity | 600 | 8 | 380 | 0.13 | 43 |
| | Position | 520 | 8 | 860 | 0.26 | 79 |
| MC_Maze-M | Velocity | 420 | 2 | 360 | 0.09 | 59 |
| | Position | 580 | 8 | 340 | 0.02 | 74 |
| MC_Maze-S | Velocity | 520 | 4 | 210 | 0.07 | 94 |
| MC_RTT | Velocity | 1040 | 2 | 700 | 0.15 | 17 |

## GRU

The GRU neural network (*Cho et al., 2014*) we use takes spike count observations (current and previous time bins), centers and normalizes those observations,

$$\tilde{\mathbf{x}}_i = (\mathbf{x}_i - \mu) \oslash \sigma, \ i \in \{k - \kappa, \ldots, k\} \tag{S35}$$

and then feeds the observations sequentially into the network (initializing with $\mathbf{h}_{k-\kappa-1} = \mathbf{0}$)

$$\mathbf{z}_i = \text{sigmoid}(W_z \tilde{\mathbf{x}}_i + U_z \mathbf{h}_{i-1} + b_z) \tag{S36}$$

$$\mathbf{r}_i = \text{sigmoid}(W_r \tilde{\mathbf{x}}_i + U_r \mathbf{h}_{i-1} + b_r) \tag{S37}$$

$$\hat{\mathbf{h}}_i = \tanh(W_h \tilde{\mathbf{x}}_i + U_h(\mathbf{r}_i \odot \mathbf{h}_{i-1}) + b_h) \tag{S38}$$

$$\mathbf{h}_i = \mathbf{z}_i \odot \mathbf{h}_{i-1} + (1 - \mathbf{z}_i) \odot \hat{\mathbf{h}}_i \tag{S39}$$

ultimately reading out behavior from the final state

$$\hat{\mathbf{y}}_k = W_{out} \mathbf{h}_k + b_{out} \tag{S40}$$

where $\mathbf{z}_i \in \mathbb{R}^D$, $\mathbf{r}_i \in \mathbb{R}^D$, $\hat{\mathbf{h}}_i \in \mathbb{R}^D$, $\mathbf{h}_i \in \mathbb{R}^D$, and $D$ is the number of GRU units. The GRU hidden states do not persist from one decoding time bin to the next. Rather, at each decoding time bin, the GRU is re-initialized with $\mathbf{h}_{k-\kappa-1} = \mathbf{0}$ and run sequentially over recent history to generate an estimate of the behavior at the current time bin.

## Parameter fitting

$\mu$ and $\sigma$ (both column vectors of length $N$) are computed as the mean and standard deviation, respectively, of the observed spike counts for each neuron in the training set. The parameters $W_z$, $U_z$, $b_z$, $W_r$, $U_r$, $b_r$, $W_h$, $U_h$, $b_h$, $W_{out}$, and $b_{out}$ are then learned by training the network with the RMSProp optimization routine (*Tieleman, 2012*) and a mean-squared error loss function. Training utilized dropout both on the linear transformation of inputs and on the linear transformation of the recurrent state.

**Appendix 1—table 4.** Hyperparameters used for the GRU network.
The number of units ($D$) was optimized in the range [500, 1000], with the optimized values rounded to the closest multiple of 10. The dropout rate was optimized in the range [0,0.5] and the number

of training epochs was optimized in the range [2, 50]. Window lengths were optimized (in 20 ms increments) in the range [200, 600] for Area2_Bump, [200, 1000] for MC_Cycle, [200, 1200] for MC_RTT, and [200, 700] for MC_Maze, MC_Maze-L, MC_Maze-M, and MC_Maze-S.

| Dataset | Behavioral Group | Window Length (ms) | Units ($D$) | Dropout Rate | Epochs |
|---|---|---|---|---|---|
| | Position | 560 | 700 | 0.06 | 12 |
| | Velocity | 340 | 820 | 0.32 | 3 |
| | Force | 380 | 380 | 0.27 | 26 |
| | Joint Angles | 480 | 630 | 0.31 | 8 |
| | Joint Velocities | 260 | 390 | 0.27 | 9 |
| | Muscle Lengths | 460 | 990 | 0.34 | 45 |
| Area2_Bump | Muscle Velocities | 420 | 870 | 0.19 | 47 |
| | Phase | 460 | 580 | 0.18 | 49 |
| | Angular Velocity | 480 | 390 | 0.12 | 27 |
| | Position | 920 | 760 | 0.06 | 42 |
| | Velocity | 520 | 170 | 0.43 | 45 |
| MC_Cycle | EMG | 840 | 250 | 0.40 | 36 |
| | Position | 640 | 740 | 0.27 | 4 |
| MC_Maze | Velocity | 520 | 800 | 0.34 | 11 |
| | Position | 620 | 640 | 0.40 | 49 |
| MC_Maze-L | Velocity | 580 | 420 | 0.36 | 31 |
| | Position | 620 | 820 | 0.41 | 48 |
| MC_Maze-M | Velocity | 660 | 840 | 0.29 | 30 |
| | Position | 460 | 310 | 0.39 | 27 |
| MC_Maze-S | Velocity | 680 | 500 | 0.44 | 45 |
| MC_RTT | Velocity | 540 | 890 | 0.40 | 8 |

## Naive Bayes regression

The Naive Bayes regression decoder (*Zhang et al., 1998*, *Glaser et al., 2020*) is fully described with code provided in *Glaser et al., 2020*. The hyperparameters for this decoder include the density of the kinematic grid, bin size, and the number of previous bins available to the decoder. This decoder was only evaluated for the MC_Maze dataset. The best performance was achieved for a 50 x 50 grid of kinematics. For both position and velocity, hyperparameter optimization indicated use of only a single bin of spike counts (220 ms and 320 ms bin sizes for position and velocity, respectively).

