## [Editor Report · eLife Assessment]

This paper presents a new method called MINT that is effective at BCI-style decoding tasks. The authors show **convincing** evidence to support their claims regarding how MINT is a new method that produces excellent decoding performance relative to the state-of-the-art. This work is **important** and will be of broad interest to neuroscientists and neuroengineers.

---

## [Referee Report · Reviewer #1 (Public Review)]

Summary:

This paper presents an innovative decoding approach for brain-computer interfaces (BCIs), introducing a new method named MINT. The authors develop a trajectory-centric approach to decode behaviors across several different datasets, including eight empirical datasets from the Neural Latents Benchmark. Overall, the paper is well written and their method shows impressive performance compared to more traditional decoding approaches that use a simpler approach. While there are some concerns (see below), the paper's strengths, particularly its emphasis on a trajectory-centric approach and the simplicity of MINT, provide a compelling contribution to the field.

Strengths:

The adoption of a trajectory-centric approach that utilizes statistical constraints presents a substantial shift in methodology, potentially revolutionizing the way BCIs interpret and predict neural behaviour. This is one of the strongest aspects of the paper.

The thorough evaluation of the method across various datasets serves as an assurance that the superior performance of MINT is not a result of overfitting. The comparative simplicity of the method in contrast to many neural network approaches is refreshing and should facilitate broader applicability.

Weaknesses:

Scope: Despite the impressive performance of MINT across multiple datasets, it seems predominantly applicable to M1/S1 data. Only one of the eight empirical datasets comes from an area outside the motor/somatosensory cortex. It would be beneficial if the authors could expand further on how the method might perform with other brain regions that do not exhibit low tangling or do not have a clear trial structure (e.g. decoding of position or head direction from hippocampus)

When comparing methods, the neural trajectories of MINT are based on averaged trials, while the comparison methods are trained on single trials. An additional analysis might help in disentangling the effect of the trial averaging. For this, the authors could average the input across trials for all decoders, establishing a baseline for averaged trials. Note that inference should still be done on single trials. Performance can then be visualized across different values of N, which denotes the number of averaged trials used for training.

Comments on revisions:

I have looked at the responses and they are thorough and answer all of my questions.

---

## [Referee Report · Reviewer #2 (Public Review)]

Summary:

The goal of this paper is to present a new method, termed MINT, for decoding behavioral states from neural spiking data. MINT is a statistical method which, in addition to outputting a decoded behavioral state, also provides soft information regarding the likelihood of that behavioral state based on the neural data. The innovation in this approach is neural states are assumed to come from sparsely distributed neural trajectories with low tangling, meaning that neural trajectories (time sequences of neural states) are sparse in the high-dimensional space of neural spiking activity and that two dissimilar neural trajectories tend to correspond to dissimilar behavioral trajectories. The authors support these assumptions through analysis of previously collected data, and then validate the performance of their method by comparing it to a suite of alternative approaches. The authors attribute the typically improved decoding performance by MINT to its assumptions being more faithfully aligned to the properties of neural spiking data relative to assumptions made by the alternatives.

Strengths:

The paper did an excellent job critically evaluating common assumptions made by neural analytical methods, such as neural state being low-dimensional relative to the number of recorded neurons. The authors made strong arguments, supported by evidence and literature, for potentially high-dimensional neural states and thus the need for approaches that do not rely on an assumption of low dimensionality.

The paper was thorough in considering multiple datasets across a variety of behaviors, as well as existing decoding methods, to benchmark the MINT approach. This provided a valuable comparison to validate the method. The authors also provided nice intuition regarding why MINT may offer performance improvement in some cases and in which instances MINT may not perform as well.

In addition to providing a philosophical discussion as to the advantages of MINT and benchmarking against alternatives, the authors also provided a detailed description of practical considerations. This included training time, amount of training data, robustness to data loss or changes in the data, and interpretability. These considerations not only provided objective evaluation of practical aspects but also provided insights to the flexibility and robustness of the method as they relate back to the underlying assumptions and construction of the approach.

Impact:

This work is motivated by brain-computer interfaces applications, which it will surely impact in terms of neural decoder design. However, this work is also broadly impactful for neuroscientific analysis to relate neural spiking activity to observable behavioral features. Thus, MINT will likely impact neuroscience research generally. The methods are made publicly available, and the datasets used are all in public repositories, which facilitates adoption and validation of this method within the greater scientific community.

---

## [Author Response]

The following is the authors’ response to the original reviews.

Summary of reviewers’ comments and our revisions:

We thank the reviewers for their thoughtful feedback. This feedback has motivated multiple revisions and additions that, in our view, have greatly improved the manuscript. This is especially true with regard to a major goal of this study: clearly defining existing scientific perspectives and delineating their decoding implications. In addition to building on this conceptual goal, we have expanded existing analyses and have added a new analysis of generalization using a newly collected dataset. We expect the manuscript will be of very broad interest, both to those interested in BCI development and to those interested in fundamental properties of neural population activity and its relationship with behavior.

Importantly, all reviewers were convinced that MINT provided excellent performance, when benchmarked against existing methods, across a broad range of standard tasks:

“their method shows impressive performance compared to more traditional decoding approaches” (R1)

“The paper was thorough in considering multiple datasets across a variety of behaviors, as well as existing decoding methods, to benchmark the MINT approach. This provided a valuable comparison to validate the method.” (R2)

“The fact that performance on stereotyped tasks is high is interesting and informative…” (R3)

This is important. It is challenging to design a decoder that performs consistently across multiple domains and across multiple situations (including both decoding and neural state estimation). MINT does so. MINT consistently outperformed existing lightweight ‘interpretable’ decoders, despite being a lightweight interpretable decoder itself. MINT was very competitive with expressive machine-learning methods, yet has advantages in flexibility and simplicity that more ‘brute force’ methods do not. We made a great many comparisons, and MINT was consistently a strong performer. Of the many comparisons we made, there was only one where MINT was at a modest disadvantage, and it was for a dataset where all methods performed poorly. No other method we tested was as consistent. For example, although the GRU and the feedforward network were often competitive with MINT (and better than MINT in the one case mentioned above), there were multiple other situations where they performed less well and a few situations where they performed poorly. Moreover, no other existing decoder naturally estimates the neural state while also readily decoding, without retraining, a broad range of behavioral variables.

R1 and R2 were very positive about the broader impacts of the study. They stressed its impact both on decoder design, and on how our field thinks, scientifically, about the population response in motor areas:

“This paper presents an innovative decoding approach for brain-computer interfaces” (R1)

“presents a substantial shift in methodology, potentially revolutionizing the way BCIs interpret and predict neural behaviour” (R1)

“the paper's strengths, particularly its emphasis on a trajectory-centric approach and the simplicity of MINT, provide a compelling contribution to the field” (R1)

“The authors made strong arguments, supported by evidence and literature, for potentially high-dimensional neural states and thus the need for approaches that do not rely on an assumption of low dimensionality” (R2)

“This work is motivated by brain-computer interfaces applications, which it will surely impact in terms of neural decoder design.” (R2)

“this work is also broadly impactful for neuroscientific analysis... Thus, MINT will likely impact neuroscience research generally.” (R2)

We agree with these assessments, and have made multiple revisions to further play into these strengths. As one example, the addition of Figure 1b (and 6b) makes this the first study, to our knowledge, to fully and concretely illustrate this emerging scientific perspective and its decoding implications. This is important, because multiple observations convince us that the field is likely to move away from the traditional perspective in Figure 1a, and towards that in Figure 1b. We also agree with the handful of weaknesses R1 and R2 noted. The manuscript has been revised accordingly. The major weakness noted by R1 was the need to be explicit regarding when we suspect MINT would (and wouldn’t) work well in other brain areas. In non-motor areas, the structure of the data may be poorly matched with MINT’s assumptions. We agree that this is likely to be true, and thus agree with the importance of clarifying this topic for the reader. The revision now does so. R1 also wished to know whether existing methods might benefit from including trial-averaged data during training, something we now explore and document (see detailed responses below). R2 noted two weaknesses: (1) The need to better support (with expanded analysis) the statement that neural and behavioral trajectories are non-isometric, and (2) The need to more rigorously define the ‘mesh’. We agree entirely with both suggestions, and the revision has been strengthened by following them (see detailed responses below).

R3 also saw strengths to the work, stating that:

“This paper is well-structured and its main idea is clear.”

“The fact that performance on stereotyped tasks is high is interesting and informative, showing that these stereotyped tasks create stereotyped neural trajectories.”

“The task-specific comparisons include various measures and a variety of common decoding approaches, which is a strength.”

However, R3 also expressed two sizable concerns. The first is that MINT might have onerous memory requirements. The manuscript now clarifies that MINT has modest memory requirements. These do not scale unfavorably as the reviewer was concerned they might. The second concern is that MINT is:

“essentially a table-lookup rather than a model.”

Although we don’t agree, the concern makes sense and may be shared by many readers, especially those who take a particular scientific perspective. Pondering this concern thus gave us the opportunity to modify the manuscript in ways that support its broader impact. Our revisions had two goals: (1) clarify the ways in which MINT is far more flexible than a lookup-table, and (2) better describe the dominant scientific perspectives and their decoding implications.

The heart of R3’s concern is the opinion that MINT is an effective but unprincipled hack suitable for situations where movements are reasonably stereotyped. Of course, many tasks involve stereotyped movements (e.g. handwriting characters), so MINT would still be useful. Nevertheless, if MINT is not principled, other decode methods would often be preferable because they could (unlike MINT in R3’s opinion) gain flexibility by leveraging an accurate model. Most of R3’s comments flow from this fundamental concern:

“This is again due to MINT being a lookup table with a library of stereotyped trajectories rather than a model.”

“MINT models task-dependent neural trajectories, so the trained decoder is very task-dependent and cannot generalize to other tasks.”

“Unlike MINT, these works can achieve generalization because they model the neural subspace and its association to movement.”

“given that MINT tabulates task-specific trajectories, it will not generalize to tasks that are not seen in the training data even when these tasks cover the exact same space (e.g., the same 2D computer screen and associated neural space).”

“For proper training, the training data should explore the whole movement space and the associated neural space, but this does not mean all kinds of tasks performed in that space must be included in the training set (something MINT likely needs while modeling-based approaches do not).”

The manuscript has been revised to clarify that MINT is considerably more flexible than a lookup table, even though a lookup table is used as a first step. Yet, on its own, this does not fully address R3’s concern. The quotes above highlight that R3 is making a standard assumption in our field: that there exists a “movement space and associated neural space”. Under this perspective, one should, as R3 argues fully explore the movement space. This would perforce fully explore the associated neural subspace. One can then “model the neural subspace and its association to movement”. MINT does not use a model of this type, and thus (from R3’s perspective) does not appear to use a model at all. A major goal of our study is to question this traditional perspective. We have thus added a new figure to highlight the contrast between the traditional (Figure 1a) and new (Figure 1b) scientific perspectives, and to clarify their decoding implications.

While we favor the new perspective (Figure 1b), we concede that R3 may not share our view. This is fine. Part of the reason we believe this study is timely, and will be broadly read, is that it raises a topic of emerging interest where there is definitely room for debate. If we are misguided – i.e. if Figure 1a is the correct perspective – then many of R3’s concerns would be on target: MINT could still be useful, but traditional methods that make the traditional assumptions in Figure 1a would often be preferable. However, if the emerging perspective in Figure 1b is more accurate, then MINT’s assumptions would be better aligned with the data than those of traditional methods, making it a more (not less) principled choice.

Our study provides new evidence in support of Figure 1b, while also synthesizing existing evidence from other recent studies. In addition to Figure 2, the new analysis of generalization further supports Figure 1b. Also supporting Figure 1b is the analysis in which MINT’s decoding advantage, over a traditional decoder, disappears when simulated data approximate the traditional perspective in Figure 1a.

That said, we agree that the present study cannot fully resolve whether Figure 1a or 1b is more accurate. Doing so will take multiple studies with different approaches (indeed we are currently preparing other manuscripts on this topic). Yet we still have an informed scientific opinion, derived from past, present and yet-to-be-published observations. Our opinion is that Figure 1b is the more accurate perspective. This possibility makes it reasonable to explore the potential virtues of a decoding method whose assumptions are well-aligned with that perspective. MINT is such a method. As expected under Figure 1b, MINT outperforms traditional interpretable decoders in every single case we studied.

As noted above, we have added a new generalization-focused analysis (Figure 6) based on a newly collected dataset. We did so because R3’s comments highlight a deep point: which scientific perspective one takes has strong implications regarding decoder generalization. These implications are now illustrated in the new Figure 6a and 6b. Under Figure 6a, it is possible, as R3 suggests, to explore “the whole movement space and associated neural space” during training. However, under Figure 6b, expectations are very different. Generalization will be ‘easy’ when new trajectories are near the training-set trajectories. In this case, MINT should generalize well as should other methods. In contrast, generalization will be ‘hard’ when new neural trajectories have novel shapes and occupy previously unseen regions / dimensions. In this case, all current methods, including MINT, are likely to fail. R3 points out that traditional decoders have sometimes generalized well to new tasks (e.g. from center-out to ‘pinball’) when cursor movements occur in the same physical workspace. These findings could be taken to support Figure 6a, but are equally consistent with ‘easy’ generalization in Figure 6b. To explore this topic, the new analysis in Figure 6c-g considers conditions that are intended to span the range from easy to hard. Results are consistent with the predictions of Figure 6b.

We believe the manuscript has been significantly improved by these additions. The revisions help the manuscript achieve its twin goals: (1) introduce a novel class of decoder that performs very well despite being very simple, and (2) describe properties of motor-cortex activity that will matter for decoders of all varieties.

**Reviewer #1:**
Summary:This paper presents an innovative decoding approach for brain-computer interfaces (BCIs), introducing a new method named MINT. The authors develop a trajectory-centric approach to decode behaviors across several different datasets, including eight empirical datasets from the Neural Latents Benchmark. Overall, the paper is well written and their method shows impressive performance compared to more traditional decoding approaches that use a simpler approach. While there are some concerns (see below), the paper's strengths, particularly its emphasis on a trajectory-centric approach and the simplicity of MINT, provide a compelling contribution to the field.

We thank the reviewer for these comments. We share their enthusiasm for the trajectory-centric approach, and we are in complete agreement that this perspective has both scientific and decoding implications. The revision expands upon these strengths.

Strengths:The adoption of a trajectory-centric approach that utilizes statistical constraints presents a substantial shift in methodology, potentially revolutionizing the way BCIs interpret and predict neural behaviour. This is one of the strongest aspects of the paper.

Again, thank you. We also expect the trajectory-centric perspective to have a broad impact, given its relevance to both decoding and to thinking about manifolds.

The thorough evaluation of the method across various datasets serves as an assurance that the superior performance of MINT is not a result of overfitting. The comparative simplicity of the method in contrast to many neural network approaches is refreshing and should facilitate broader applicability.

Thank you. We were similarly pleased to see such a simple method perform so well. We also agree that, while neural-network approaches will always be important, it is desirable to also possess simple ‘interpretable’ alternatives.

Weaknesses:Comment (1) Scope: Despite the impressive performance of MINT across multiple datasets, it seems predominantly applicable to M1/S1 data. Only one of the eight empirical datasets comes from an area outside the motor/somatosensory cortex. It would be beneficial if the authors could expand further on how the method might perform with other brain regions that do not exhibit low tangling or do not have a clear trial structure (e.g. decoding of position or head direction from hippocampus)

We agree entirely. Population activity in many brain areas (especially outside the motor system) presumably will often not have the properties upon which MINT’s assumptions are built. This doesn’t necessarily mean that MINT would perform badly. Using simulated data, we have found that MINT can perform surprisingly well even when some of its assumptions are violated. Yet at the same time, when MINT’s assumptions don’t apply, one would likely prefer to use other methods. This is, after all, one of the broader themes of the present study: it is beneficial to match decoding assumptions to empirical properties. We have thus added a section on this topic early in the Discussion:

“In contrast, MINT and the Kalman filter performed comparably on simulated data that better approximated the assumptions in Figure 1a. Thus, MINT is not a ‘better’ algorithm – simply better aligned with the empirical properties of motor cortex data. This highlights an important caveat. Although MINT performs well when decoding from motor areas, its assumptions may be a poor match in other areas (e.g. the hippocampus). MINT performed well on two non-motor-cortex datasets – Area2_Bump (S1) and DMFC_RSG (dorsomedial frontal cortex) – yet there will presumably be other brain areas and/or contexts where one would prefer a different method that makes assumptions appropriate for that area.”

Comment (2) When comparing methods, the neural trajectories of MINT are based on averaged trials, while the comparison methods are trained on single trials. An additional analysis might help in disentangling the effect of the trial averaging. For this, the authors could average the input across trials for all decoders, establishing a baseline for averaged trials. Note that inference should still be done on single trials. Performance can then be visualized across different values of N, which denotes the number of averaged trials used for training.

We explored this question and found that the non-MINT decoders are harmed, not helped, by the inclusion of trial-averaged responses in the training set. This is presumably because the statistics of trialaveraged responses don’t resemble what will be observed during decoding. This statistical mismatch, between training and decoding, hurts most methods. It doesn’t hurt MINT, because MINT doesn’t ‘train’ in the normal way. It simply needs to know rates, and trial-averaging is a natural way to obtain them. To describe the new analysis, we have added the following to the text.

“We also investigated the possibility that MINT gained its performance advantage simply by having access to trial-averaged neural trajectories during training, while all other methods were trained on single-trial data. This difference arises from the fundamental requirements of the decoder architectures: MINT needs to estimate typical trajectories while other methods don’t. Yet it might still be the case that other methods would benefit from including trial-averaged data in the training set, in addition to single-trial data. Alternatively, this might harm performance by creating a mismatch, between training and decoding, in the statistics of decoder inputs. We found that the latter was indeed the case: all non-MINT methods performed better when trained purely on single-trial data.”

**Reviewer #2**:Summary:The goal of this paper is to present a new method, termed MINT, for decoding behavioral states from neural spiking data. MINT is a statistical method which, in addition to outputting a decoded behavioral state, also provides soft information regarding the likelihood of that behavioral state based on the neural data. The innovation in this approach is neural states are assumed to come from sparsely distributed neural trajectories with low tangling, meaning that neural trajectories (time sequences of neural states) are sparse in the high-dimensional space of neural spiking activity and that two dissimilar neural trajectories tend to correspond to dissimilar behavioral trajectories. The authors support these assumptions through analysis of previously collected data, and then validate the performance of their method by comparing it to a suite of alternative approaches. The authors attribute the typically improved decoding performance by MINT to its assumptions being more faithfully aligned to the properties of neural spiking data relative to assumptions made by the alternatives.

We thank the reviewer for this accurate summary, and for highlighting the subtle but important fact that MINT provides information regarding likelihoods. The revision includes a new analysis (Figure 6e) illustrating one potential way to leverage knowledge of likelihoods.

Strengths:The paper did an excellent job critically evaluating common assumptions made by neural analytical methods, such as neural state being low-dimensional relative to the number of recorded neurons. The authors made strong arguments, supported by evidence and literature, for potentially high-dimensional neural states and thus the need for approaches that do not rely on an assumption of low dimensionality.

Thank you. We also hope that the shift in perspective is the most important contribution of the study. This shift matters both scientifically and for decoder design. The revision expands on this strength. The scientific alternatives are now more clearly and concretely illustrated (especially see Figure 1a,b and Figure 6a,b). We also further explore their decoding implications with new data (Figure 6c-g).

The paper was thorough in considering multiple datasets across a variety of behaviors, as well as existing decoding methods, to benchmark the MINT approach. This provided a valuable comparison to validate the method. The authors also provided nice intuition regarding why MINT may offer performance improvement in some cases and in which instances MINT may not perform as well.

Thank you. We were pleased to be able to provide comparisons across so many datasets (we are grateful to the Neural Latents Benchmark for making this possible).

In addition to providing a philosophical discussion as to the advantages of MINT and benchmarking against alternatives, the authors also provided a detailed description of practical considerations. This included training time, amount of training data, robustness to data loss or changes in the data, and interpretability. These considerations not only provided objective evaluation of practical aspects but also provided insights to the flexibility and robustness of the method as they relate back to the underlying assumptions and construction of the approach.

Thank you. We are glad that these sections were appreciated. MINT’s simplicity and interpretability are indeed helpful in multiple ways, and afford opportunities for interesting future extensions. One potential benefit of interpretability is now explored in the newly added Figure 6e.

Impact:This work is motivated by brain-computer interfaces applications, which it will surely impact in terms of neural decoder design. However, this work is also broadly impactful for neuroscientific analysis to relate neural spiking activity to observable behavioral features. Thus, MINT will likely impact neuroscience research generally. The methods are made publicly available, and the datasets used are all in public repositories, which facilitates adoption and validation of this method within the greater scientific community.

Again, thank you. We have similar hopes for this study.

Weaknesses (1 & 2 are related, and we have switched their order in addressing them):

Comment (2) With regards to the idea of neural and behavioral trajectories having different geometries, this is dependent on what behavioral variables are selected. In the example for Fig 2a, the behavior is reach position. The geometry of the behavioral trajectory of interest would look different if instead the behavior of interest was reach velocity. The paper would be strengthened by acknowledgement that geometries of trajectories are shaped by extrinsic choices rather than (or as much as they are) intrinsic properties of the data.

We agree. Indeed, we almost added a section to the original manuscript on this exact topic. We have now done so:

“A potential concern regarding the analyses in Figure 2c,d is that they require explicit choices of behavioral variables: muscle population activity in Figure 2c and angular phase and velocity in Figure 2d. Perhaps these choices were misguided. Might neural and behavioral geometries become similar if one chooses ‘the right’ set of behavioral variables? This concern relates to the venerable search for movement parameters that are reliably encoded by motor cortex activity [69, 92–95]. If one chooses the wrong set of parameters (e.g. chooses muscle activity when one should have chosen joint angles) then of course neural and behavioral geometries will appear non-isometric. There are two reasons why this ‘wrong parameter choice’ explanation is unlikely to account for the results in Figure 2c,d. First, consider the implications of the left-hand side of Figure 2d. A small kinematic distance implies that angular position and velocity are nearly identical for the two moments being compared. Yet the corresponding pair of neural states can be quite distant. Under the concern above, this distance would be due to other encoded behavioral variables – perhaps joint angle and joint velocity – differing between those two moments. However, there are not enough degrees of freedom in this task to make this plausible. The shoulder remains at a fixed position (because the head is fixed) and the wrist has limited mobility due to the pedal design [60]. Thus, shoulder and elbow angles are almost completely determined by cycle phase. More generally, ‘external variables’ (positions, angles, and their derivatives) are unlikely to differ more than slightly when phase and angular velocity are matched. Muscle activity could be different because many muscles act on each joint, creating redundancy. However, as illustrated in Figure 2c, the key effect is just as clear when analyzing muscle activity. Thus, the above concern seems unlikely even if it can’t be ruled out entirely. A broader reason to doubt the ‘wrong parameter choice’ proposition is that it provides a vague explanation for a phenomenon that already has a straightforward explanation. A lack of isometry between the neural population response and behavior is expected when neural-trajectory tangling is low and output-null factors are plentiful [55, 60]. For example, in networks that generate muscle activity, neural and muscle-activity trajectories are far from isometric [52, 58, 60]. Given this straightforward explanation, and given repeated failures over decades to find the ‘correct’ parameters (muscle activity, movement direction, etc.) that create neural-behavior isometry, it seems reasonable to conclude that no such isometry exists.”

Comment (1) The authors posit that neural and behavioral trajectories are non-isometric. To support this point, they look at distances between neural states and distances between the corresponding behavioral states, in order to demonstrate that there are differences in these distances in each respective space. This supports the idea that neural states and behavioral states are non-isometric but does not directly address their point. In order to say the trajectories are non-isometric, it would be better to look at pairs of distances between corresponding trajectories in each space.

We like this idea and have added such an analysis. To be clear, we like the original analysis too: isometry predicts that neural and behavioral distances (for corresponding pairs of points) should be strongly correlated, and that small behavioral distances should not be associated with large neural distances. These predictions are not true, providing a strong argument against isometry. However, we also like the reviewer’s suggestion, and have added such an analysis. It makes the same larger point, and also reveals some additional facts (e.g. it reveals that muscle-geometry is more related to neural-geometry than is kinematic-geometry). The new analysis is described in the following section:

“We further explored the topic of isometry by considering pairs of distances. To do so, we chose two random neural states and computed their distance, yielding dneural1. We repeated this process, yielding dneural2. We then computed the corresponding pair of distances in muscle space (dmuscle1 and dmuscle2) and kinematic space (dkin1 and dkin2). We considered cases where dneural1 was meaningfully larger than (or smaller than) dneural2, and asked whether the behavioral variables had the same relationship; e.g. was dmuscle1 also larger than dmuscle2? For kinematics, this relationship was weak: across 100,000 comparisons, the sign of dkin1 − dkin2 agreed with dneural1 − dneural2 only 67.3% of the time (with 50% being chance). The relationship was much stronger for muscles: the sign of dmuscle1 − dmuscle2 agreed with dneural1 − dneural2 79.2% of the time, which is far more than expected by chance yet also far from what is expected given isometry (e.g. the sign agrees 99.7% of the time for the truly isometric control data in Figure 2e). Indeed there were multiple moments during this task when dneural1 was much larger than dneural2, yet dmuscle1 was smaller than dmuscle2. These observations are consistent with the proposal that neural trajectories resemble muscle trajectories in some dimensions, but with additional output-null dimensions that break the isometry [60].”

Comment (3) The approach is built up on the idea of creating a "mesh" structure of possible states. In the body of the paper the definition of the mesh was not entirely clear and I could not find in the methods a more rigorous explicit definition. Since the mesh is integral to the approach, the paper would be improved with more description of this component.

This is a fair criticism. Although MINTs actual operations were well-documented, how those operations mapped onto the term ‘mesh’ was, we agree, a bit vague. The definition of the mesh is a bit subtle because it only emerges during decoding rather than being precomputed. This is part of what gives MINT much more flexibility than a lookup table. We have added the following to the manuscript.

“We use the term ‘mesh’ to describe the scaffolding created by the training-set trajectories and the interpolated states that arise at runtime. The term mesh is apt because, if MINT’s assumptions are correct, interpolation will almost always be local. If so, the set of decodable states will resemble a mesh, created by line segments connecting nearby training-set trajectories. However, this mesh-like structure is not enforced by MINT’s operations.

Interpolation could, in principle, create state-distributions that depart from the assumption of a sparse manifold. For example, interpolation could fill in the center of the green tube in Figure 1b, resulting in a solid manifold rather than a mesh around its outer surface. However, this would occur only if spiking observations argued for it. As will be documented below, we find that essentially all interpolation is local”

We have also added Figure 4d. This new analysis documents the fact that decoded states are near trainingset trajectories, which is why the term ‘mesh’ is appropriate.

**Reviewer #3:**
Summary:This manuscript develops a new method termed MINT for decoding of behavior. The method is essentially a table-lookup rather than a model. Within a given stereotyped task, MINT tabulates averaged firing rate trajectories of neurons (neural states) and corresponding averaged behavioral trajectories as stereotypes to construct a library. For a test trial with a realized neural trajectory, it then finds the closest neural trajectory to it in the table and declares the associated behavior trajectory in the table as the decoded behavior. The method can also interpolate between these tabulated trajectories. The authors mention that the method is based on three key assumptions: (1) Neural states may not be embedded in a lowdimensional subspace, but rather in a high-dimensional space. (2) Neural trajectories are sparsely distributed under different behavioral conditions. (3) These neural states traverse trajectories in a stereotyped order.The authors conducted multiple analyses to validate MINT, demonstrating its decoding of behavioral trajectories in simulations and datasets (Figures 3, 4). The main behavior decoding comparison is shown in Figure 4. In stereotyped tasks, decoding performance is comparable (M_Cycle, MC_Maze) or better (Area 2_Bump) than other linear/nonlinear algorithms(Figure 4). However, MINT underperforms for the MC_RTT task, which is less stereotyped (Figure 4).This paper is well-structured and its main idea is clear. The fact that performance on stereotyped tasks is high is interesting and informative, showing that these stereotyped tasks create stereotyped neural trajectories. The task-specific comparisons include various measures and a variety of common decoding approaches, which is a strength. However, I have several major concerns. I believe several of the conclusions in the paper, which are also emphasized in the abstract, are not accurate or supported, especially about generalization, computational scalability, and utility for BCIs. MINT is essentially a table-lookup algorithm based on stereotyped task-dependent trajectories and involves the tabulation of extensive data to build a vast library without modeling. These aspects will limit MINT's utility for real-world BCIs and tasks. These properties will also limit MINT's generalizability from task to task, which is important for BCIs and thus is commonly demonstrated in BCI experiments with other decoders without any retraining. Furthermore, MINT's computational and memory requirements can be prohibitive it seems. Finally, as MINT is based on tabulating data without learning models of data, I am unclear how it will be useful in basic investigations of neural computations. I expand on these concerns below.

We thank the reviewer for pointing out weaknesses in our framing and presentation. The comments above made us realize that we needed to (1) better document the ways in which MINT is far more flexible than a lookup-table, and (2) better explain the competing scientific perspectives at play. R3’s comments also motivated us to add an additional analysis of generalization. In our view the manuscript is greatly improved by these additions. Specifically, these additions directly support the broader impact that we hope the study will have.

For simplicity and readability, we first group and summarize R3’s main concerns in order to better address them. (These main concerns are all raised above, in addition to recurring in the specific comments below. Responses to each individual specific comment are provided after these summaries.)

(1) R3 raises concerns about ‘computational scalability.’ The concern is that “MINT's computational and memory requirements can be prohibitive.” This point was expanded upon in a specific comment, reproduced below:

I also find the statement in the abstract and paper that "computations are simple, scalable" to be inaccurate. The authors state that MINT's computational cost is O(NC) only, but it seems this is achieved at a high memory cost as well as computational cost in training. The process is described in section "Lookup table of log-likelihoods" on line [978-990]. The idea is to precompute the log-likelihoods for any combination of all neurons with discretization x all delay/history segments x all conditions and to build a large lookup table for decoding. Basically, the computational cost of precomputing this table is O(V^{Nτ} x TC) and the table requires a memory of O(V^{Nτ}), where V is the number of discretization points for the neural firing rates, N is the number of neurons, τ is the history length, T is the trial length, and C is the number of conditions. This is a very large burden, especially the V^{Nτ} term. This cost is currently not mentioned in the manuscript and should be clarified in the main text. Accordingly, computation claims should be modified including in the abstract.

The revised manuscript clarifies that our statement (that computations are simple and scalable) is absolutely accurate. There is no need to compute, or store, a massive lookup table. There are three tables: two of modest size and one that is tiny. This is now better explained:

“Thus, the log-likelihood of s¯t′−τ′:t′, for a particular current neural state, is simply the sum of many individual log-likelihoods (one per neuron and time-bin). Each individual log-likelihood depends on only two numbers: the firing rate at that moment and the spike count in that bin. To simplify online computation, one can precompute the log-likelihood, under a Poisson model, for every plausible combination of rate and spike-count. For example, a lookup table of size 2001 × 21 is sufficient when considering rates that span 0-200 spikes/s in increments of 0.1 spikes/s, and considering 20 ms bins that contain at most 20 spikes (only one lookup table is ever needed, so long as its firing-rate range exceeds that of the most-active neuron at the most active moment in Ω). Now suppose we are observing a population of 200 neurons, with a 200 ms history divided into ten 20 ms bins. For each library state, the log-likelihood of the observed spike-counts is simply the sum of 200 × 10 = 2000 individual loglikelihoods, each retrieved from the lookup table. In practice, computation is even simpler because many terms can be reused from the last time bin using a recursive solution (Methods). This procedure is lightweight and amenable to real-time applications.”

In summary, the first table simply needs to contain the firing rate of each neuron, for each condition, and each time in that condition. This table consumes relatively little memory. Assuming 100 one-second-long conditions (rates sampled every 20 ms) and 200 neurons, the table would contain 100 x 50 x 200 = 1,000,000 numbers. These numbers are typically stored as 16-bit integers (because rates are quantized), which amounts to about 2 MB. This is modest, given that most computers have (at least) tens of GB of RAM. A second table would contain the values for each behavioral variable, for each condition, and each time in that condition. This table might contain behavioral variables at a finer resolution (e.g. every millisecond) to enable decoding to update in between 20 ms bins (1 ms granularity is not needed for most BCI applications, but is the resolution used in this study). The number of behavioral variables of interest for a particular BCI application is likely to be small, often 1-2, but let’s assume for this example it is 10 (e.g. x-, y-, and z-position, velocity, and acceleration of a limb, plus one other variable). This table would thus contain 100 x 1000 x 10 = 1,000,000 floating point numbers, i.e. an 8 MB table. The third table is used to store the probability of s spikes being observed given a particular quantized firing rate (e.g. it may contain probabilities associated with firing rates ranging from 0 – 200 spikes/s in 0.1 spikes/s increments). This table is not necessary, but saves some computation time by precomputing numbers that will be used repeatedly. This is a very small table (typically ~2000 x 20, i.e. 320 KB). It does not need to be repeated for different neurons or conditions, because Poisson probabilities depend on only rate and count.

(2) R3 raises a concern that MINT “is essentially a table-lookup rather than a model.’ R3 states that MINT

“is essentially a table-lookup algorithm based on stereotyped task-dependent trajectories and involves the tabulation of extensive data to build a vast library without modeling.”

and that,

“as MINT is based on tabulating data without learning models of data, I am unclear how it will be useful in basic investigations of neural computations.”

This concern is central to most subsequent concerns. The manuscript has been heavily revised to address it. The revisions clarify that MINT is much more flexible than a lookup table, even though MINT uses a lookup table as its first step. Because R3’s concern is intertwined with one’s scientific assumptions, we have also added the new Figure 1 to explicitly illustrate the two key scientific perspectives and their decoding implications.

Under the perspective in Figure 1a, R3 would be correct in saying that there exist traditional interpretable decoders (e.g. a Kalman filter) whose assumptions better model the data. Under this perspective, MINT might still be an excellent choice in many cases, but other methods would be expected to gain the advantage when situations demand more flexibility. This is R3’s central concern, and essentially all other concerns flow from it. It makes sense that R3 has this concern, because their comments repeatedly stress a foundational assumption of the perspective in Figure 1a: the assumption of a fixed lowdimensional neural subspace where activity has a reliable relationship to behavior that can be modeled and leveraged during decoding. The phrases below accord with that view:

“Unlike MINT, these works can achieve generalization because they model the neural subspace and its association to movement.”

“it will not generalize… even when these tasks cover the exact same space (e.g., the same 2D computer screen and associated neural space).”

“For proper training, the training data should explore the whole movement space and the associated neural space”

“I also believe the authors should clarify the logic behind developing MINT better. From a scientific standpoint, we seek to gain insights into neural computations by making various assumptions and building models that parsimoniously describe the vast amount of neural data rather than simply tabulating the data. For instance, low-dimensional assumptions have led to the development of numerous dimensionality reduction algorithms and these models have led to important interpretations about the underlying dynamics”

Thus, R3 prefers a model that (1) assumes a low-dimensional subspace that is fixed across tasks and (2) assumes a consistent ‘association’ between neural activity and kinematics. Because R3 believes this is the correct model of the data, they believe that decoders should leverage it. Traditional interpretable method do, and MINT doesn’t, which is why they find MINT to be unprincipled. This is a reasonable view, but it is not our view. We have heavily revised the manuscript to clarify that a major goal of our study is to explore the implications of a different, less-traditional scientific perspective.

The new Figure 1a illustrates the traditional perspective. Under this perspective, one would agree with R3’s claim that other methods have the opportunity to model the data better. For example, suppose there exists a consistent neural subspace – conserved across tasks – where three neural dimensions encode 3D hand position and three additional neural dimensions encode 3D hand velocity. A traditional method such as a Kalman filter would be a very appropriate choice to model these aspects of the data.

Figure 1b illustrates the alternative scientific perspective. This perspective arises from recent, present, and to-be-published observations. MINT’s assumptions are well-aligned with this perspective. In contrast, the assumptions of traditional methods e.g. the Kalman filter are not well-aligned with the properties of the data under this perspective. This does not mean traditional methods are not useful. Yet under Figure 1b, it is traditional methods, such as the Kalman filter, that lack an accurate model of the data. Of course, the reviewer may disagree with our scientific perspective. We would certainly concede that there is room for debate. However, we find the evidence for Figure 1b to be sufficiently strong that it is worth exploring the utility of methods that align with this scientific perspective. MINT is such a method. As we document, it performs very well.

Thus, in our view, MINT is quite principled because its assumptions are well aligned with the data. It is true that the features of the data that MINT models are a bit different from those that are traditionally modeled. For example, R3 is quite correct that MINT does not attempt to use a biomimetic model of the true transformation from neural activity, to muscle activity, and thence to kinematics. We see this as a strength, and the manuscript has been revised accordingly (see paragraph beginning with “We leveraged this simulated data to compare MINT with a biomimetic decoder”).

(3) R3 raises concerns that MINT cannot generalize. This was a major concern of R3 and is intimately related to concern #2 above. The concern is that, if MINT is “essentially a lookup table” that simply selects pre-defined trajectories, then MINT will not be able to generalize. R3 is quite correct that MINT generalizes rather differently than existing methods. Whether this is good or bad depends on one’s scientific perspective. Under Figure 1a, MINT’s generalization would indeed be limiting because other methods could achieve greater flexibility. Under Figure 1b, all methods will have serious limits regarding generalization. Thus, MINT’s method for generalizing may approximate the best one can presently do. To address this concern, we have made three major changes, numbered i-iii below:

i) Large sections of the manuscript have been restructured to underscore the ways in which MINT can generalize. A major goal was to counter the impression, stated by R3 above, that:

“for a test trial with a realized neural trajectory, [MINT] then finds the closest neural trajectory to it in the table and declares the associated behavior trajectory in the table as the decoded behavior”.

This description is a reasonable way to initially understand how MINT works, and we concede that we may have over-used this intuition. Unfortunately, it can leave the misimpression that MINT decodes by selecting whole trajectories, each corresponding to ‘a behavior’. This can happen, but it needn’t and typically doesn’t. As an example, consider the cycling task. Suppose that the library consists of stereotyped trajectories, each four cycles long, at five fixed speeds from 0.5-2.5 Hz. If the spiking observations argued for it, MINT could decode something close to one of these five stereotyped trajectories. Yet it needn’t. Decoded trajectories will typically resemble library trajectories locally, but may be very different globally. For example, a decoded trajectory could be thirty cycles long (or two, or five hundred) perhaps speeding up and slowing down multiple times across those cycles.

Thus, the library of trajectories shouldn’t be thought of as specifying a limited set of whole movements that can be ‘selected from’. Rather, trajectories define a scaffolding that outlines where the neural state is likely to live and how it is likely to be changing over time. When we introduce the idea of library trajectories, we are now careful to stress that they don’t function as a set from which one trajectory is ‘declared’ to be the right one:

“We thus designed MINT to approximate that manifold using the trajectories themselves, rather than their covariance matrix or corresponding subspace. Unlike a covariance matrix, neural trajectories indicate not only which states are likely, but also which state-derivatives are likely. If a neural state is near previously observed states, it should be moving in a similar direction. MINT leverages this directionality.

Training-set trajectories can take various forms, depending on what is convenient to collect. Most simply, training data might include one trajectory per condition, with each condition corresponding to a discrete movement. Alternatively, one might instead employ one long trajectory spanning many movements. Another option is to employ many sub-trajectories, each briefer than a whole movement. The goal is simply for training-set trajectories to act as a scaffolding, outlining the manifold that might be occupied during decoding and the directions in which decoded trajectories are likely to be traveling.”

Later in that same section we stress that decoded trajectories can move along the ‘mesh’ in nonstereotyped ways:

“Although the mesh is formed of stereotyped trajectories, decoded trajectories can move along the mesh in non-stereotyped ways as long as they generally obey the flow-field implied by the training data. This flexibility supports many types of generalization, including generalization that is compositional in nature. Other types of generalization – e.g. from the green trajectories to the orange trajectories in Figure 1b – are unavailable when using MINT and are expected to be challenging for any method (as will be documented in a later section).”

The section “Training and decoding using MINT” has been revised to clarify the ways in which interpolation is flexible, allowing decoded movements to be globally very different from any library trajectory.

“To decode stereotyped trajectories, one could simply obtain the maximum-likelihood neural state from the library, then render a behavioral decode based on the behavioral state with the same values of c and k. This would be appropriate for applications in which conditions are categorical, such as typing or handwriting. Yet in most cases we wish for the trajectory library to serve not as an exhaustive set of possible states, but as a scaffolding for the mesh of possible states. MINT’s operations are thus designed to estimate any neural trajectory – and any corresponding behavioral trajectory – that moves along the mesh in a manner generally consistent with the trajectories in Ω.”

“…interpolation allows considerable flexibility. Not only is one not ‘stuck’ on a trajectory from Φ, one is also not stuck on trajectories created by weighted averaging of trajectories in Φ. For example, if cycling speed increases, the decoded neural state could move steadily up a scaffolding like that illustrated in Figure 1b (green). In such cases, the decoded trajectory might be very different in duration from any of the library trajectories. Thus, one should not think of the library as a set of possible trajectories that are selected from, but rather as providing a mesh-like scaffolding that defines where future neural states are likely to live and the likely direction of their local motion. The decoded trajectory may differ considerably from any trajectory within Ω.”

This flexibility is indeed used during movement. One empirical example is described in detail:

“During movement… angular phase was decoded with effectively no net drift over time. This is noteworthy because angular velocity on test trials never perfectly matched any of the trajectories in Φ. Thus, if decoding were restricted to a library trajectory, one would expect growing phase discrepancies. Yet decoded trajectories only need to locally (and approximately) follow the flow-field defined by the library trajectories. Based on incoming spiking observations, decoded trajectories speed up or slow down (within limits).

This decoding flexibility presumably relates to the fact that the decoded neural state is allowed to differ from the nearest state in Ω. To explore… [the text goes on to describe the new analysis in Figure 4d, which shows that the decoded state is typically not on any trajectory, though it is typically close to a trajectory].”

Thus, MINT’s operations allow considerable flexibility, including generalization that is compositional in nature. Yet R3 is still correct that there are other forms of generalization that are unavailable to MINT. This is now stressed at multiple points in the revision. However, under the perspective in Figure 1b, these forms of generalization are unavailable to any current method. Hence we made a second major change in response to this concern… (ii) We explicitly illustrate how the structure of the data determines when generalization is or isn’t possible. The new Figure 1a,b introduces the two perspectives, and the new Figure 6a,b lays out their implications for generalization. Under the perspective in Figure 6a, the reviewer is quite right: other methods can generalize in ways that MINT cannot. Under the perspective in Figure 6b, expectations are very different. Those expectations make testable predictions. Hence the third major change… (iii) We have added an analysis of generalization, using a newly collected dataset. This dataset was collected using Neuropixels Probes during our Pac-Man force-tracking task. This dataset was chosen because it is unusually well-suited to distinguishing the predictions in Figure 6a versus Figure 6b. Finding a dataset that can do so is not simple. Consider R3’s point that training data should “explore the whole movement space and the associated neural space”. The physical simplicity of the Pac-Man task makes it unusually easy to confirm that the behavioral workspace has been fully explored. Importantly, under Figure 6b, this does not mean that the neural workspace has been fully explored, which is exactly what we wish to test when testing generalization. We do so, and compare MINT with a Wiener filter. A Wiener filter is an ideal comparison because it is simple, performs very well on this task, and should be able to generalize well under Figure 1a. Additionally, the Wiener filter (unlike the Kalman Filter) doesn’t leverage the assumption that neural activity reflects the derivative of force. This matters because we find that neural activity does not reflect dforce/dt in this task. The Wiener filter is thus the most natural choice of the interpretable methods whose assumptions match Figure 1a.

The new analysis is described in Figure 6c-g and accompanying text. Results are consistent with the predictions of Figure 6b. We are pleased to have been motivated to add this analysis for two reasons. First, it provides an additional way of evaluating the predictions of the two competing scientific perspectives that are at the heart of our study. Second, this analysis illustrates an underappreciated way in which generalization is likely to be challenging for any decode method. It can be tempting to think that the main challenge regarding generalization is to fully explore the relevant behavioral space. This makes sense if a behavioral space has “an associated neural space”. However, we are increasingly of the opinion that it doesn’t. Different tasks often involve different neural subspaces, even when behavioral subspaces overlap. We have even seen situations where motor output is identical but neural subspaces are quite different. These facts are relevant to any decoder, something highlighted in the revised Introduction:

“MINT’s performance confirms that there are gains to be made by building decoders whose assumptions match a different, possibly more accurate view of population activity. At the same time, our results suggest fundamental limits on decoder generalization. Under the assumptions in Figure 1b, it will sometimes be difficult or impossible for decoders to generalize to not-yet-seen tasks. We found that this was true regardless of whether one uses MINT or a more traditional method. This finding has implications regarding when and how generalization should be attempted.”

We have also added an analysis (Figure 6e) illustrating how MINT’s ability to compute likelihoods can be useful in detecting situations that may strain generalization (for any method). MINT is unusual in being able to compute and use likelihoods in this way.

Detailed responses to R3: we reproduce each of R3’s specific concerns below, but concentrate our responses on issues not already covered above.

Main comments:Comment 1. MINT does not generalize to different tasks, which is a main limitation for BCI utility compared with prior BCI decoders that have shown this generalizability as I review below. Specifically, given that MINT tabulates task-specific trajectories, it will not generalize to tasks that are not seen in the training data even when these tasks cover the exact same space (e.g., the same 2D computer screen and associated neural space).

First, the authors provide a section on generalization, which is inaccurate because it mixes up two fundamentally different concepts: (1) collecting informative training data and (2) generalizing from task to task. The former is critical for any algorithm, but it does not imply the latter. For example, removing one direction of cycling from the training set as the authors do here is an example of generating poor training data because the two behavioral (and neural) directions are non-overlapping and/or orthogonal while being in the same space. As such, it is fully expected that all methods will fail. For proper training, the training data should explore the whole movement space and the associated neural space, but this does not mean all kinds of tasks performed in that space must be included in the training set (something MINT likely needs while modeling-based approaches do not). Many BCI studies have indeed shown this generalization ability using a model. For example, in Weiss et al. 2019, center-out reaching tasks are used for training and then the same trained decoder is used for typing on a keyboard or drawing on the 2D screen. In Gilja et al. 2012, training is on a center-out task but the same trained decoder generalizes to a completely different pinball task (hit four consecutive targets) and tasks requiring the avoidance of obstacles and curved movements. There are many more BCI studies, such as Jarosiewicz et al. 2015 that also show generalization to complex realworld tasks not included in the training set. Unlike MINT, these works can achieve generalization because they model the neural subspace and its association to movement. On the contrary, MINT models task-dependent neural trajectories, so the trained decoder is very task-dependent and cannot generalize to other tasks. So, unlike these prior BCIs methods, MINT will likely actually need to include every task in its library, which is not practical.

I suggest the authors remove claims of generalization and modify their arguments throughout the text and abstract. The generalization section needs to be substantially edited to clarify the above points. Please also provide the BCI citations and discuss the above limitation of MINT for BCIs.

As discussed above, R3’s concerns are accurate under the view in Figure 1a (and the corresponding Figure 6a). Under this view, a method such as that in Gilja et al. or Jarosiewicz et al. can find the correct subspace, model the correct neuron-behavior correlations, and generalize to any task that uses “the same 2D computer screen and associated neural space”, just as the reviewer argues. Under Figure 1b things are quite different.

This topic – and the changes we have made to address it – is covered at length above. Here we simply want to highlight an empirical finding: sometimes two tasks use the same neural subspace and sometimes they don’t. We have seen both in recent data, and it is can be very non-obvious which will occur based just on behavior. It does not simply relate to whether one is using the same physical workspace. We have even seen situations where the patterns of muscle activity in two tasks are nearly identical, but the neural subspaces are fairly different. When a new task uses a new subspace, neither of the methods noted above (Gilja nor Jarosiewicz) will generalize (nor will MINT). Generalizing to a new subspace is basically impossible without some yet-to-be-invented approach. On the other hand, there are many other pairs of tasks (center-out-reaching versus some other 2D cursor control) where subspaces are likely to be similar, especially if the frequency content of the behavior is similar (in our recent experience this is often critical). When subspaces are shared, most methods will generalize, and that is presumably why generalization worked well in the studies noted above.

Although MINT can also generalize in such circumstances, R3 is correct that, under the perspective in Figure 1a, MINT will be more limited than other methods. This is now carefully illustrated in Figure 6a. In this traditional perspective, MINT will fail to generalize in cases where new trajectories are near previously observed states, yet move in very different ways from library trajectories. The reason we don’t view this is a shortcoming is that we expect it to occur rarely (else tangling would be high). We thus anticipate the scenario in Figure 6b.

This is worth stressing because R3 states that our discussion of generalization “is inaccurate because it mixes up two fundamentally different concepts: (1) collecting informative training data and (2) generalizing from task to task.” We have heavily revised this section and improved it. However, it was never inaccurate. Under Figure 6b, these two concepts absolutely are mixed up. If different tasks use different neural subspaces, then this requires collecting different “informative training data” for each. One cannot simply count on having explored the physical workspace.

Comment 2. MINT is shown to achieve competitive/high performance in highly stereotyped datasets with structured trials, but worse performance on MC_RTT, which is not based on repeated trials and is less stereotyped. This shows that MINT is valuable for decoding in repetitive stereotyped use-cases. However, it also highlights a limitation of MINT for BCIs, which is that MINT may not work well for real-world and/or less-constrained setups such as typing, moving a robotic arm in 3D space, etc. This is again due to MINT being a lookup table with a library of stereotyped trajectories rather than a model. Indeed, the authors acknowledge that the lower performance on MC_RTT (Figure 4) may be caused by the lack of repeated trials of the same type. However, real-world BCI decoding scenarios will also not have such stereotyped trial structure and will be less/un-constrained, in which MINT underperforms. Thus, the claim in the abstract or lines 480-481 that MINT is an "excellent" candidate for clinical BCI applications is not accurate and needs to be qualified. The authors should revise their statements according and discuss this issue. They should also make the use-case of MINT on BCI decoding clearer and more convincing.

We discussed, above, multiple changes and additions to the revision that were made to address these concerns. Here we briefly expand on the comment that MINT achieves “worse performance on MC_RTT, which is not based on repeated trials and is less stereotyped”. All decoders performed poorly on this task. MINT still outperformed the two traditional methods, but this was the only dataset where MINT did not also perform better (overall) than the expressive GRU and feedforward network. There are probably multiple reasons why. We agree with R3 that one likely reason is that this dataset is straining generalization, and MINT may have felt this strain more than the two machine-learning-based methods. Another potential reason is the structure of the training data, which made it more challenging to obtain library trajectories in the first place. Importantly, these observations do not support the view in Figure 1a. MINT still outperformed the Kalman and Wiener filters (whose assumptions align with Fig. 1a). To make these points we have added the following:

“Decoding was acceptable, but noticeably worse, for the MC_RTT dataset… As will be discussed below, every decode method achieved its worst estimates of velocity for the MC_RTT dataset. In addition to the impact of slower reaches, MINT was likely impacted by training data that made it challenging to accurate estimate library trajectories. Due to the lack of repeated trials, MINT used AutoLFADS to estimate the neural state during training. In principle this should work well. In practice AutoLFADS may have been limited by having only 10 minutes of training data. Because the random-target task involved more variable reaches, it may also have stressed the ability of all methods to generalize, perhaps for the reasons illustrated in Figure 1b.

The only dataset where MINT did not perform the best overall was the MC_RTT dataset, where it was outperformed by the feedforward network and GRU. As noted above, this may relate to the need for MINT to learn neural trajectories from training data that lacked repeated trials of the same movement (a design choice one might wish to avoid). Alternatively, the less-structured MC_RTT dataset may strain the capacity to generalize; all methods experienced a drop in velocity-decoding R2 for this dataset compared to the others. MINT generalizes somewhat differently than other methods, and may have been at a modest disadvantage for this dataset. A strong version of this possibility is that perhaps the perspective in Figure 1a is correct, in which case MINT might struggle because it cannot use forms of generalization that are available to other methods (e.g. generalization based on neuron-velocity correlations). This strong version seems unlikely; MINT continued to significantly outperform the Wiener and Kalman filters, which make assumptions aligned with Figure 1a.”

Comment 3. Related to 2, it may also be that MINT achieves competitive performance in offline and trial-based stereotyped decoding by overfitting to the trial structure in a given task, and thus may not generalize well to online performance due to overfitting. For example, a recent work showed that offline decoding performance may be overfitted to the task structure and may not represent online performance (Deo et al. 2023). Please discuss.

We agree that a limitation of our study is that we do not test online performance. There are sensible reasons for this decision:

“By necessity and desire, all comparisons were made offline, enabling benchmarked performance across a variety of tasks and decoded variables, where each decoder had access to the exact same data and recording conditions.”

We recently reported excellent online performance in the cycling task with a different algorithm

(Schroeder et al. 2022). In the course of that study, we consistently found that improvements in our offline decoding translated to improvements in our online decoding. We thus believe that MINT (which improves on the offline performance of our older algorithm) is a good candidate to work very well online. Yet we agree this still remains to be seen. We have added the following to the Discussion:

“With that goal in mind, there exist three important practical considerations. First, some decode algorithms experience a performance drop when used online. One presumed reason is that, when decoding is imperfect, the participant alters their strategy which in turn alters the neural responses upon which decoding is based. Because MINT produces particularly accurate decoding, this effect may be minimized, but this cannot be known in advance. If a performance drop does indeed occur, one could adapt the known solution of retraining using data collected during online decoding [13]. Another presumed reason (for a gap between offline and online decoding) is that offline decoders can overfit the temporal structure in training data [107]. This concern is somewhat mitigated by MINT’s use of a short spike-count history, but MINT may nevertheless benefit from data augmentation strategies such as including timedilated versions of learned trajectories in the libraries”

Comment 4. Related to 2, since MINT requires firing rates to generate the library and simple averaging does not work for this purpose in the MC_RTT dataset (that does not have repeated trials), the authors needed to use AutoLFADS to infer the underlying firing rates. The fact that MINT requires the usage of another model to be constructed first and that this model can be computationally complex, will also be a limiting factor and should be clarified.

This concern relates to the computational complexity of computing firing-rate trajectories during training. Usually, rates are estimated via trial-averaging, which makes MINT very fast to train. This was quite noticeable during the Neural Latents Benchmark competition. As one example, for the “MC_Scaling 5 ms Phase”, MINT took 28 seconds to train while GPFA took 30 minutes, the transformer baseline (NDT) took 3.5 hours, and the switching nonlinear dynamical system took 4.5 hours.

However, the reviewer is quite correct that MINT’s efficiency depends on the method used to construct the library of trajectories. As we note, “MINT is a method for leveraging a trajectory library, not a method for constructing it”. One can use trial-averaging, which is very fast. One can also use fancier, slower methods to compute the trajectories. We don’t view this as a negative – it simply provides options. Usually one would choose trial-averaging, but one does not have to. In the case of MC_RTT, one has a choice between LFADS and grouping into pseudo-conditions and averaging (which is fast). LFADS produces higher performance at the cost of being slower. The operator can choose which they prefer. This is discussed in the following section:

“For MINT, ‘training’ simply means computation of standard quantities (e.g. firing rates) rather than parameter optimization. MINT is thus typically very fast to train (Table 1), on the order of seconds using generic hardware (no GPUs). This speed reflects the simple operations involved in constructing the library of neural-state trajectories: filtering of spikes and averaging across trials. At the same time we stress that MINT is a method for leveraging a trajectory library, not a method for constructing it. One may sometimes wish to use alternatives to trial-averaging, either of necessity or because they improve trajectory estimates. For example, for the MC_RTT task we used AutoLFADS to infer the library. Training was consequently much slower (hours rather than seconds) because of the time taken to estimate rates. Training time could be reduced back to seconds using a different approach – grouping into pseudo-conditions and averaging – but performance was reduced. Thus, training will typically be very fast, but one may choose time-consuming methods when appropriate.”

Comment 5. I also find the statement in the abstract and paper that "computations are simple, scalable" to be inaccurate. The authors state that MINT's computational cost is O(NC) only, but it seems this is achieved at a high memory cost as well as computational cost in training. The process is described in section "Lookup table of log-likelihoods" on line [978-990]. The idea is to precompute the log-likelihoods for any combination of all neurons with discretization x all delay/history segments x all conditions and to build a large lookup table for decoding. Basically, the computational cost of precomputing this table is O(V^{Nτ} x TC) and the table requires a memory of O(V^{Nτ}), where V is the number of discretization points for the neural firing rates, N is the number of neurons, τ is the history length, T is the trial length, and C is the number of conditions. This is a very large burden, especially the V^{Nτ} term. This cost is currently not mentioned in the manuscript and should be clarified in the main text. Accordingly, computation claims should be modified including in the abstract.

As discussed above, the manuscript has been revised to clarify that our statement was accurate.

Comment 6. In addition to the above technical concerns, I also believe the authors should clarify the logic behind developing MINT better. From a scientific standpoint, we seek to gain insights into neural computations by making various assumptions and building models that parsimoniously describe the vast amount of neural data rather than simply tabulating the data. For instance, low-dimensional assumptions have led to the development of numerous dimensionality reduction algorithms and these models have led to important interpretations about the underlying dynamics (e.g., fixed points/limit cycles). While it is of course valid and even insightful to propose different assumptions from existing models as the authors do here, they do not actually translate these assumptions into a new model. Without a model and by just tabulating the data, I don't believe we can provide interpretation or advance the understanding of the fundamentals behind neural computations. As such, I am not clear as to how this library building approach can advance neuroscience or how these assumptions are useful. I think the authors should clarify and discuss this point.

As requested, a major goal of the revision has been to clarify the scientific motivations underlying MINT’s design. In addition to many textual changes, we have added figures (Figures 1a,b and 6a,b) to outline the two competing scientific perspectives that presently exist. This topic is also addressed by extensions of existing analyses and by new analyses (e.g. Figure 6c-g).

In our view these additions have dramatically improved the manuscript. This is especially true because we think R3’s concerns, expressed above, are reasonable. If the perspective in Figure 1a is correct, then R3 is right and MINT is essentially a hack that fails to model the data. MINT would still be effective in many circumstances (as we show), but it would be unprincipled. This would create limitations, just as the reviewer argues. On the other hand, if the perspective in Figure 1b is correct, then MINT is quite principled relative to traditional approaches. Traditional approaches make assumptions (a fixed subspace, consistent neuron-kinematic correlations) that are not correct under Figure 1b.

We don’t expect R3 to agree with our scientific perspective at this time (though we hope to eventually convince them). To us, the key is that we agree with R3 that the manuscript needs to lay out the different perspectives and their implications, so that readers have a good sense of the possibilities they should be considering. The revised manuscript is greatly improved in this regard.

Comment 7. Related to 6, there seems to be a logical inconsistency between the operations of MINT and one of its three assumptions, namely, sparsity. The authors state that neural states are sparsely distributed in some neural dimensions (Figure 1a, bottom). If this is the case, then why does MINT extend its decoding scope by interpolating known neural states (and behavior) in the training library? This interpolation suggests that the neural states are dense on the manifold rather than sparse, thus being contradictory to the assumption made. If interpolation-based dense meshes/manifolds underlie the data, then why not model the neural states through the subspace or manifold representations? I think the authors should address this logical inconsistency in MINT, especially since this sparsity assumption also questions the low-dimensional subspace/manifold assumption that is commonly made.

We agree this is an important issue, and have added an analysis on this topic (Figure 4d). The key question is simple and empirical: during decoding, does interpolation cause MINT to violate the assumption of sparsity? R3 is quite right that in principle it could. If spiking observations argue for it, MINT’s interpolation could create a dense manifold during decoding rather than a sparse one. The short answer is that empirically this does not happen, in agreement with expectations under Figure 1b. Rather than interpolating between distant states and filling in large ‘voids’, interpolation is consistently local. This is a feature of the data, not of the decoder (MINT doesn’t insist upon sparsity, even though it is designed to work best in situations where the manifold is sparse).

In addition to adding Figure 4d, we added the following (in an earlier section):

“The term mesh is apt because, if MINT’s assumptions are correct, interpolation will almost always be local. If so, the set of decodable states will resemble a mesh, created by line segments connecting nearby training-set trajectories. However, this mesh-like structure is not enforced by MINT’s operations. Interpolation could, in principle, create state-distributions that depart from the assumption of a sparse manifold. For example, interpolation could fill in the center of the green tube in Figure 1b, resulting in a solid manifold rather than a mesh around its outer surface. However, this would occur only if spiking observations argued for it. As will be documented below, we find that essentially all interpolation is local.”

**Recommendations for the authors**:
**Reviewer #1 (Recommendations For The Authors):**
I appreciate the detailed methods section, however, more specifics should be integrated into the main text. For example on Line 238, it should additionally be stated how many minutes were used for training and metrics like the MAE which is used later should be reported here.

Thank you for this suggestion. We now report the duration of training data in the main text:

“Decoding R^2 was .968 over ~7.1 minutes of test trials based on ~4.4 minutes of training data.”

We have also added similar specifics throughout the manuscript, e.g. in the Fig. 5 legend:

“Results are based on the following numbers of training / test trials: MC\_Cycle (174 train, 99 test), MC\_Maze (1721 train, 574 test), Area2\_Bump (272 train, 92 test), MC\_RTT (810 train, 268 test).”

Similar additions were made to the legends for Fig. 6 and 8. Regarding the request to add MAE for the multitask network, we did not do so for the simple reason that the decoded variable (muscle activity) has arbitrary units. The raw MAE is thus not meaningful. We could of course have normalized, but at this point the MAE is largely redundant with the correlation. In contrast, the MAE is useful when comparing across the MC_Maze, Area2_Bump, and MC_RTT datasets, because they all involve the same scale (cm/s).

Regarding the MC_RTT task, AutoLFADS was used to obtain robust spike rates, as reported in the methods. However, the rationale for splitting the neural trajectories after AutoLFADS is unclear. If the trajectories were split based on random recording gaps, this might lead to suboptimal performance? It might be advantageous to split them based on a common behavioural state?

When learning neural trajectories via AutoLFADS, spiking data is broken into short (but overlapping) segments, rates are estimated for each segment via AutoLFADs, and these rates are then stitched together across segments into long neural trajectories. If there had been no recording gaps, these rates could have been stitched into a single neural trajectory for this dataset. However, the presence of recording gaps left us no choice but to stitch together these rates into more than one trajectory. Fortunately, recording gaps were rare: for the decoding analysis of MC_RTT there were only two recording gaps and therefore three neural trajectories, each ~2.7 minutes in duration.

We agree that in general it is desirable to learn neural trajectories that begin and end at behaviorallyrelevant moments (e.g. in between movements). However, having these trajectories potentially end midmovement is not an issue in and of itself. During decoding, MINT is never stuck on a trajectory. Thus, if MINT were decoding states near the end of a trajectory that was cut short due to a training gap, it would simply begin decoding states from other trajectories or elsewhere along the same trajectory in subsequent moments. We could have further trimmed the three neural trajectories to begin and end at behaviorallyrelevant moments, but chose not to as this would have only removed a handful of potentially useful states from the library.

We now describe this in the Methods:

“Although one might prefer trajectory boundaries to begin and end at behaviorally relevant moments (e.g. a stationary state), rather than at recording gaps, the exact boundary points are unlikely to be consequential for trajectories of this length that span multiple movements. If MINT estimates a state near the end of a long trajectory, its estimate will simply jump to another likely state on a different trajectory (or earlier along the same trajectory) in subsequent moments. Clipping the end of each trajectory to an earlier behaviorally-relevant moment would only remove potentially useful states from the libraries.”

Are the training and execution times in Table 1 based on pure Matlab functions or Mex files? If it's Mex files as suggested by the code, it would be good to mention this in the Table caption.

They are based on a combination of MATLAB and MEX files. This is now clarified in the table caption:

“Timing measurements taken on a Macbook Pro (on CPU) with 32GB RAM and a 2.3 GHz 8-Core Intel Core i9 processor. Training and execution code used for measurements was written in MATLAB (with the core recursion implemented as a MEX file).”

As the method most closely resembles a Bayesian decoder it would be good to compare performance against a Naive Bayes decoder.

We agree and have now done so. The following has been added to the text:

“A natural question is thus whether a simpler Bayesian decoder would have yielded similar results. We explored this possibility by testing a Naïve Bayes regression decoder [85] using the MC_Maze dataset. This decoder performed poorly, especially when decoding velocity (R2 = .688 and .093 for hand position and velocity, respectively), indicating that the specific modeling assumptions that differentiate MINT from a naive Bayesian decoder are important drivers of MINT’s performance.”

Line 199 Typo: The assumption of stereotypy trajectory also enables neural states (and decoded behaviors) to be updated in between time bins.

Fixed

Table 3: It's unclear why the Gaussian binning varies significantly across different datasets. Could the authors explain why this is the case and what its implications might be?

We have added the following description in the “Filtering, extracting, and warping data on each trial” subsection of the Methods to discuss how 𝜎 may vary due to the number of trials available for training and how noisy the neural data for those trials is:

“First, spiking activity for each neuron on each trial was temporally filtered with a Gaussian to yield single-trial rates. Table 3 reports the Gaussian standard deviations σ (in milliseconds) used for each dataset. Larger values of σ utilize broader windows of spiking activity when estimating rates and therefore reduce variability in those rate estimates. However, large σ values also yield neural trajectories with less fine-grained temporal structure. Thus, the optimal σ for a dataset depends on how variable the rate estimates otherwise are.”

An implementation of the method in an open-source programming language could further enhance the widespread use of the tool.

We agree this would be useful, but have yet not implemented the method in any other programming languages. Implementation in Python is still a future goal.

**Reviewer #2 (Recommendations For The Authors):**
- Figures 4 and 5 should show the error bars on the horizontal axis rather than portraying them vertically.

[Note that these are now Figures 5 and 6]

The figure legend of Figure 5 now clarifies that the vertical ticks are simply to aid visibility when symbols have very similar means and thus overlap visually. We don’t include error bars (for this analysis) because they are very small and would mostly be smaller than the symbol sizes. Instead, to indicate certainty regarding MINT’s performance measurements, the revised text now gives error ranges for the correlations and MAE values in the context of Figure 4c. These error ranges were computed as the standard deviation of the sampling distribution (computed via resampling of trials) and are thus equivalent to SEMs. The error ranges are all very small; e.g. for the MC_Maze dataset the MAE for x-velocity is 4.5 +/- 0.1 cm/s. (error bars on the correlations are smaller still).

Thus, for a given dataset, we can be quite certain of how well MINT performs (within ~2% in the above case). This is reassuring, but we also don’t want to overemphasize this accuracy. The main sources of variability one should be concerned about are: (1) different methods can perform differentially well for different brain areas and tasks, (2) methods can decode some behavioral variables better than others, and (3) performance depends on factors like neuron-count and the number of training trials, in ways that can differ across decode methods. For this reason, the study examines multiple datasets, across tasks and brain areas, and measures performance for a range of decoded variables. We also examine the impact of training-set-size (Figure 8a) and population size (solid traces in Fig. 8b, see R2’s next comment below).

There is one other source of variance one might be concerned about, but it is specific to the neuralnetwork approaches: different weight initializations might result in different performance. For this reason, each neural-network approach was trained ten times, with the average performance computed. The variability around this average was very small, and this is now stated in the Methods.

“For the neural networks, the training/testing procedure was repeated 10 times with different random seeds. For most behavioral variables, there was very little variability in performance across repetitions. However, there were a few outliers for which variability was larger. Reported performance for each behavioral group is the average performance across the 10 repetitions to ensure results were not sensitive to any specific random initialization of each network.”

- For Figure 6, it is unclear whether the neuron-dropping process was repeated multiple times. If not, it should be since the results will be sensitive to which particular subsets of neurons were "dropped". In this case, the results presented in Figure 6 should include error bars to describe the variability in the model performance for each decoder considered.

A good point. The results in Figure 8 (previously Figure 6) were computed by averaging over the removal of different random subsets of neurons (50 subsets per neuron count), just as the reviewer requests. The figure has been modified to include the standard deviation of performance across these 50 subsets. The legend clarifies how this was done.

**Reviewer #3 (Recommendations For The Authors):**
Other comments:(1) [Line 185-188] The authors argue that in a 100-dimensional space with 10 possible discretized values, 10^100 potential neural states need to be computed. But I am not clear on this. This argument seems to hold only in the absence of a model (as in MINT). For a model, e.g., Kalman filter or AutoLFADS, information is encoded in the latent state. For example, a simple Kalman filter for a linear model can be used for efficient inference. This 10^100 computation isn't a general problem but seems MINT-specific, please clarify.

We agree this section was potentially confusing. It has been rewritten. We were simply attempting to illustrate why maximum likelihood computations are challenging without constraints. MINT simplifies this problem by adding constraints, which is why it can readily provide data likelihoods (and can do so using a Poisson model). The rewritten section is below:

“Even with 1000 samples for each of the neural trajectories in Figure 3, there are only 4000 possible neural states for which log-likelihoods must be computed (in practice it is fewer still, see Methods). This is far fewer than if one were to naively consider all possible neural states in a typical rate- or factor-based subspace. It thus becomes tractable to compute log-likelihoods using a Poisson observation model. A Poisson observation model is usually considered desirable, yet can pose tractability challenges for methods that utilize a continuous model of neural states. For example, when using a Kalman filter, one is often restricted to assuming a Gaussian observation model to maintain computational tractability “

(2) [Figure 6b] Why do the authors set the dropped neurons to zero in the "zeroed" results of the robustness analysis? Why not disregard the dropped neurons during the decoding process?

We agree the terminology we had used in this section was confusing. We have altered the figure and rewritten the text. The following, now at the beginning of that section, addresses the reviewer’s query:

“It is desirable for a decoder to be robust to the unexpected loss of the ability to detect spikes from some neurons. Such loss might occur while decoding, without being immediately detected. Additionally, one desires robustness to a known loss of neurons / recording channels. For example, there may have been channels that were active one morning but are no longer active that afternoon. At least in principle, MINT makes it very easy to handle this second situation: there is no need to retrain the decoder, one simply ignores the lost neurons when computing likelihoods. This is in contrast to nearly all other methods, which require retraining because the loss of one neuron alters the optimal parameters associated with every other neuron.”

The figure has been relabeled accordingly; instead of the label ‘zeroed’, we use the label ‘undetected neuron loss’.

(3) Authors should provide statistical significance on their results, which they already did for Fig. S3a,b,c but missing on some other figures/places.

We have added error bars in some key places, including in the text when quantifying MINT’s performance in the context of Figure 4. Importantly, error bars are only as meaningful as the source of error they assess, and there are reasons to be careful given this. The standard method for putting error bars on performance is to resample trials, which is indeed what we now report. These error bars are very small. For example, when decoding horizontal velocity for the MC_Maze dataset, the correlation between MINT’s decode and the true velocity had a mean and SD of the sampling distribution of 0.963 +/- 0.001. This means that, for a given dataset and target variable, we have enough trials/data that we can be quite certain of how well MINT performs. However, we want to be careful not to overstate this certainty. What one really wants to know is how well MINT performs across a variety of datasets, brain areas, target variables, neuron counts, etc. It is for this reason that we make multiple such comparisons, which provides a more valuable view of performance variability.

For Figure 7, error bars are unavailable. Because this was a benchmark, there was exactly one test-set that was never seen before. This is thus not something that could be resampled many times (that would have revealed the test data and thus invalidated the benchmark, not to mention that some of these methods take days to train). We could, in principle, have added resampling to Figure 5. In our view it would not be helpful and could be misleading for the reasons noted above. If we computed standard errors using different train/test partitions, they would be very tight (mostly smaller than the symbol sizes), which would give the impression that one can be quite certain of a given R^2 value. Yet variability in the train/test partition is not the variability one is concerned about in practice. In practice, one is concerned about whether one would get a similar R^2 for a different dataset, or brain area, or task, or choice of decoded variable. Our analysis thus concentrated on showing results across a broad range of situations. In our view this is a far more relevant way of illustrating the degree of meaningful variability (which is quite large) than resampling, which produces reassuringly small but (mostly) irrelevant standard errors.

Error bars are supplied in Figure 8b. These error bars give a sense of variability across re-samplings of the neural population. While this is not typically the source of variability one is most concerned about, for this analysis it becomes appropriate to show resampling-based standard errors because a natural concern is that results may depend on which neurons were dropped. So here it is both straightforward, and desirable, to compute standard errors. (The fact that MINT and the Wiener filter can be retrained many times swiftly was also key – this isn’t true of the more expressive methods). Figure S1 also uses resampling-based confidence intervals for similar reasons.

(4) [Line 431-437] Authors state that MINT outperforms other methods with the PSTH R^2 metric (trial-averaged smoothed spikes for each condition). However, I think this measure may not provide a fair comparison and is confounded because MINT's library is built using PSTH (i.e., averaged firing rate) but other methods do not use the PSTH. The author should clarify this.

The PSTH R^2 metric was not created by us; it was part of the Neural Latents Benchmark. They chose it because it ensures that a method cannot ‘cheat’ (on the Bits/Spike measure) by reproducing fine features of spiking while estimating rates badly. We agree with the reviewer’s point: MINT’s design does give it a potential advantage in this particular performance metric. This isn’t a confound though, just a feature. Importantly, MINT will score well on this metric only if MINT’s neural state estimate is accurate (including accuracy in time). Without accurate estimation of the neural state at each time, it wouldn’t matter that the library trajectory is based on PSTHs. This is now explicitly stated:

“This is in some ways unsurprising: MINT estimates neural states that tend to resemble (at least locally) trajectories ‘built’ from training-set-derived rates, which presumably resemble test-set rates. Yet strong performance is not a trivial consequence of MINT’s design. MINT does not ‘select’ whole library trajectories; PSTH R2 will be high only if condition (c), index (k), and the interpolation parameter (α) are accurately estimated for most moments.”